# Aging affects reprogramming of pulmonary capillary endothelial cells after lung injury in male mice

Marin Truchi[1,2,13], Marine Gautier-Isola [1,2,13], Grégoire Savary[3,13], Célia Scribe [1,2], Arun Lingampally[4,5], Hugo Cadis[1,2], Alberto Baeri[1], Virginie Magnone [1,2], Cédric Girard-Riboulleau[1], Marie-Jeanne Arguel [1,2], Clémentine de Schutter[3], Julien Fassy[1,2], Nihad Boukrout[3], Romain Larrue[3], Nathalie Martin [3], Roger Rezzonico [1,2], Olivier Pluquet[3], Michael Perrais[3], Véronique Hofman[2,6,7], Charles-Hugo Marquette [2,7,8], Paul Hofman [2,6,7], Andreas Günther [4,5,9,10], Nicolas Ricard [11], Pascal Barbry [1,2,12], Sylvie Leroy[1,2,8], Kevin Lebrigand [1,2], Saverio Bellusci[4,5], Christelle Cauffiez[3], Georges Vassaux [1,2], Nicolas Pottier [3,13] & Bernard Mari [1,2,13] ✉

Aging increases the risk of developing fibrotic diseases by hampering tissue regeneration after injury. Using longitudinal single-cell RNA-seq and spatial transcriptomics, here we compare the transcriptome of bleomycin (BLM) - induced fibrotic lungs of young and aged male mice, at 3 time points corresponding to the peak of fibrosis, regeneration, and resolution. We find that lung injury shifts the transcriptomic profiles of three pulmonary capillary endothelial cells (PCEC) subpopulations. The associated signatures are linked to pro-angiogenic signaling with strong *Lrg1* expression and do not progress similarly throughout the resolution process between young and old animals. Moreover, part of this set of resolution-associated markers is also detected in PCEC from samples of patients with idiopathic pulmonary fibrosis. Finally, we find that aging also alters the transcriptome of PCEC, which displays typical pro-fibrotic and pro-inflammatory features. We propose that age-associated alterations in specific PCEC subpopulations may interfere with the process of lung progenitor differentiation, thus contributing to the persistent fibrotic process typical of human pathology.

Gas exchanges between the external environment and the cardiovascular system take place in the lung alveoli through the thin membrane separating alveolar epithelial cells (AEC) and the capillary plexus formed by pulmonary capillary endothelial cells (PCEC), supported by mesenchymal cells. This alveolar niche, whose homeostasis is constantly threatened by external aggressions, deploys high regeneration abilities through various mechanisms[1]. Inadequate responses can lead to the development of chronic diseases such as idiopathic pulmonary fibrosis (IPF), an age-related pathology characterized by the

accumulation of collagen-producing mesenchymal cells (myofibroblasts) resulting in the remodeling of the alveolar environment and accumulation of extracellular matrix (ECM), leading to progressive destruction of the parenchyma[2]. IPF is the prototypical progressive interstitial lung disease (ILD) associated with progressive decline in lung function and a median survival time of less than five years after diagnosis. Although two treatments are available to manage the disease (i.e.,: nintedanib and perfenidone) and several therapeutic options are under investigation, there is no curative treatment for IPF

to date[3]. The pathogenesis of IPF is complex and largely unknown. Current hypotheses suggest that repetitive microinjuries of unknown origin to AEC in the aging lung results in ineffective repair with excessive wound healing, chronic inflammation, apoptosis of AEC and endothelial cells (EC), activation of fibrogenic effector cells, formation of myofibroblasts foci, and finally excessive deposition of ECM resulting in the destruction of the lung architecture and the loss of lung functions[4]. Mechanisms linking aging to the development of pulmonary fibrosis are not fully understood and involve numerous cell-intrinsic and cell-extrinsic alterations. Among them, it has been proposed that genetic and epigenetic changes, loss of proteostasis, cellular senescence, metabolic dysfunction, as well as stem cell exhaustion, may contribute to the initiation and progression of fibrosis[5–9].

Following lung injury, the restoration of parenchymal tissue integrity depends in particular on the proliferative capacities of PCEC[10]. Recent studies showed that, in physiological conditions, PCEC are functionally heterogeneous, with two distinct cell types conserved in multiple mammal species[11]. Aerocytes (aCap), are specialized in gas exchanges with alveolar type 1 (AT1) cells, whereas general capillaries (gCap) act as progenitor-like cells able to self-renew and to differentiate into aCap to maintain the homeostasis of the microvascular endothelium. Multiple studies also report various EC subpopulations emerging after lung injury in mouse models[12–14] or in the context of IPF[15]. In particular, a population of peri-bronchial venous EC expressing COL15A1, and referred as "systemic venous" (SV EC), has been shown to expand in areas of bronchiolization and fibrosis in IPF patients, while this EC subtype was restricted to the bronchial vasculature in healthy lungs[15,16]. A recent study[17] further characterized these COL15A1[pos] systemic EC in lungs from lethal COVID-19 and IPF patients and identified a second subpopulation lacking expression of venous marker genes while expressing markers commonly detected in gCap, referred to as systemic capillary EC (sCap). Nevertheless, the precise contribution of the physiological state-associated cell types in the formation of these pathological subpopulations is not fully known. Moreover, as the regenerative functions of EC decline with aging[18,19], the impact of aging on the dynamics of EC subpopulations associated with alveolar niche regeneration after injury-induced fibrosis remains largely unaddressed[20].

The murine single intratracheal bleomycin (BLM) instillation model is considered as the best-characterized animal model available for the preclinical assessment of potential therapies for pulmonary fibrosis[21]. After the acute inflammatory phase has subsided, BLM-instilled mice display AEC remodeling and fibrosis, but unlike patients with IPF, fibrosis progressively resolves, allowing to study the cellular and molecular mechanisms associated with alveolar repair and regeneration. Furthermore, studies assessing the response of older mice to BLM have revealed a more pronounced fibrosis, with mechanisms more reminiscent of IPF[22–24]. It has been notably suggested that fibrosis resolution or tissue regeneration is impaired in aged mice[25–28] indicating that age-associated processes contribute to the persistent fibrosis that is observed in IPF.

In this context, the aim of the present study was to compare the fate of different cell populations present in the lungs after BLM-induced injury in young and aged animals, from the fibrotic response peak (14 days) until partial (28 days) and complete (60 days) resolution.

## Results
### Aging is associated with a delay in the resolution of lung fibrosis
Lungs of young (7 weeks) and old (18 months) mice were collected 14, 28 or 60 days after BLM challenge (Fig. 1A). We noticed a shift of fibrosis resolution between young and aged mice, as visualized by total hydroxyproline assay (Fig. 1B), histological analyses (Fig. 1C) and Sirius red assay (Supplementary Fig. S1A)

with a peak of fibrosis at day 14 in young mice followed by a progressive reduction of fibrosis at day 28 while a strong fibrotic pattern was still detected at day 28 in old mice. In both young and old animals, fibrosis resolution was almost complete at day 60, with only a few limited fibrotic areas. We then performed a spatial transcriptomics experiment with the 10x Genomics Visium platform on lung sections from young or old mice taken 14 or 28 days after BLM instillation to compare the fibrotic and resolution stages between the two age groups (Fig. 1D). After filtering the genes and the low-quality spots, we processed the data, corresponding to a corpus of 2479 genes x 10,813 spots count matrix and the associated spatial coordinates, using a deconvolution tool leveraging only on spatial gene expression profiles[29]. We identified 14 unique topics, i.e. modules of covarying genes proportionally represented in each spot, and annotated them according to their DEGs (Supplementary Fig. S1B, C and Supplementary Table S1). Canonical pathway enrichment (Fig. 1E) and correlation analysis (Supplementary Fig. S1D) revealed similarity between topics, with topics related to fibrotic tissue (topics 1, 13–14), regenerative or normal alveoli (4–6) or immune cells extravasation, immunoglobin and antigen presentation signaling (8 and 12). We then compared the average proportion of each topic in all spots of each lung section, indicating at day 14 large areas of fibrosis, enriched in fibroblast and macrophage markers (topics 1, 13-14) in lungs from both young and aged mice (Fig. 1F). At day 28, lungs from young mice and, to a lesser extent, from aged mice, showed a reduced representation of topics related to fibrotic tissue compared to day 14 while the areas corresponding to regenerative or normal alveoli (4–6) showed an opposite trend. Lung sections of old mice were also specifically enriched for the topic 8 (antigen presentation signaling), suggesting that infiltration of immune cells such as plasma cells, known to form aggregates in both IPF and the BLM model[30,31], is exacerbated by aging during fibrosis resolution. Overall, using various approaches, our data indicated that fibrosis resolution is delayed in lungs from aged mice.

### Time-resolved scRNA-seq captures cellular heterogeneity of BLM-injured lungs from young and aged mice
We reproduced the same experiment and performed scRNA-seq (10x Genomics Chromium platform) on whole lungs of young (7 weeks) and aged (18 months) mice (n = 3) collected 14, 28 or 60 days after BLM challenge or PBS (Fig. 2A). The samples of both conditions from each time point were multiplexed using cell hashing[32] to mitigate batch effects. The count matrices from the 36 sequenced samples were integrated to obtain a single dataset of 45,311 cells (Supplementary Fig. S2A, B). Based on their transcriptomic profile, cells were clustered and manually annotated based on the expression of specific gene markers (Fig. 2B, Supplementary Fig. S2C, and Supplementary Table S2). We identified 41 populations recapitulating the cellular heterogeneity of the mouse lung, of which 29 were immune cells from lymphoid or myeloid lineage, such as alveolar macrophages (AMs). We also detected mesenchymal cells, including fibroblasts, pericytes, mesothelial cells as well as epithelial populations such as AT1, AT2, and bronchial multiciliated cells. We finally identified 6 EC populations: proliferating EC (Prolif. EC), gCap, aCap, venous EC, arterial EC and lymphatic EC. As expected, we observed a shift in several populations following BLM injury when performing systemic differential expression analysis between cells from BLM- and PBS-treated samples (Fig. 2C and Supplementary Table S3). This was particularly the case for macrophages, as previously described[33–35], with a rapid influx of recruited monocyte-derived macrophages AM3 populations along with a drop of the tissue-resident AM1 (Supplementary Fig. S2D). We evaluated the impact of age on the frequency of each AMs subpopulation relatively to the total number of cells analyzed in each mouse treated with PBS or

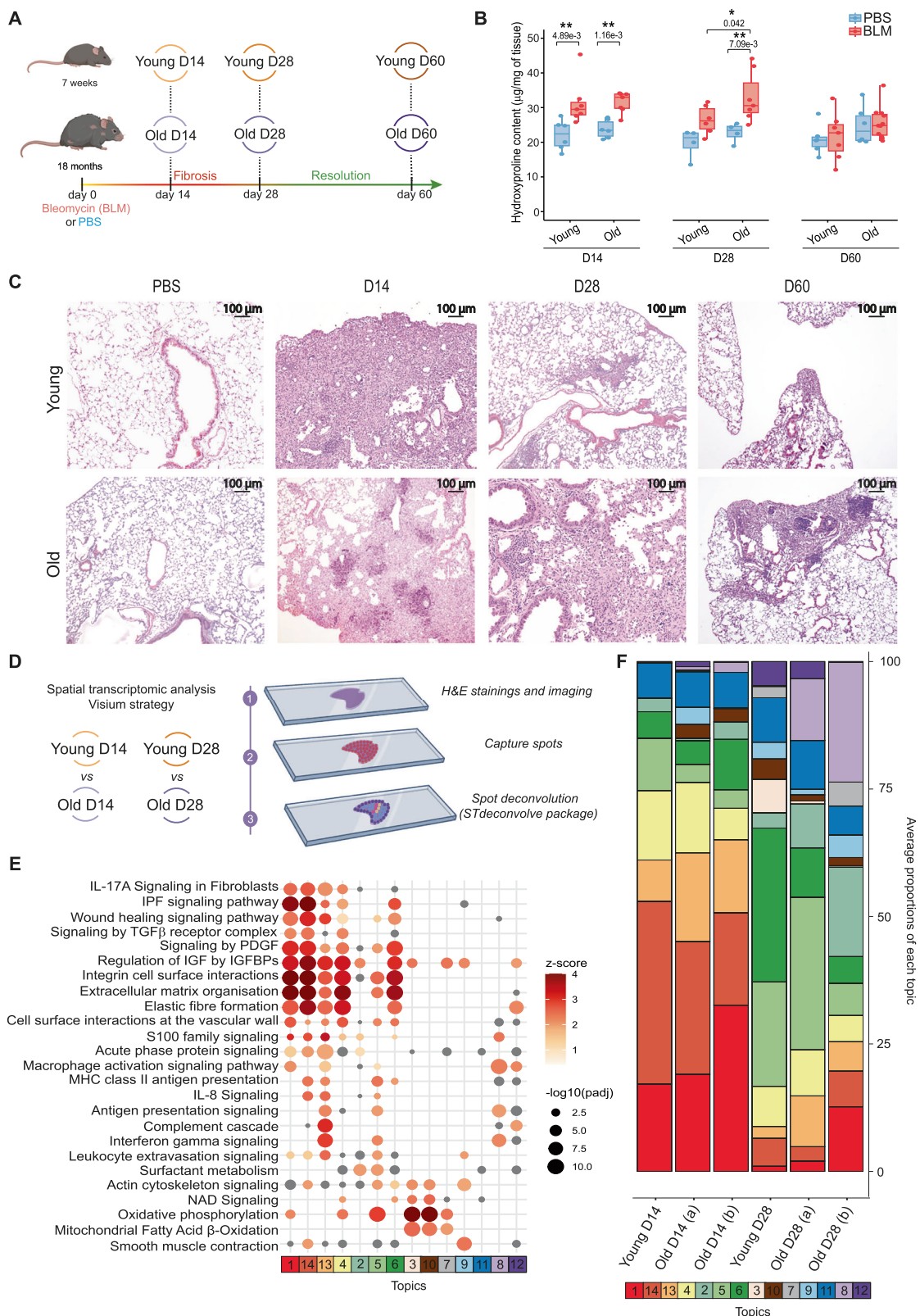

BLM (Supplementary Fig. S2E, F). Overall, BLM treatment induced a very similar dynamics of macrophage subsets in young and old animals from the peak of fibrosis to complete resolution indicating that none of these specific subpopulations are associated with the delayed resolution in aged animals. Remarkably, BLM injury also induced major alterations in pulmonary EC types, especially gCap, venous EC and

aCap (Fig. 2C), suggesting underlying cellular heterogeneity in BLM-treated mice.

## Lung injury shifts PCEC towards Lrg1[pos] subpopulations

By first examining more closely the cluster of venous EC, we were able to discriminate pulmonary-venous EC (PV EC) from systemic-venous

**Fig. 1 | Analyses of fibrotic parameters in BLM-injured lungs from young and old mice. A** Experimental design. Young (7 weeks) or old (18 months) mice were treated either with BLM or PBS. Their lungs were collected at the BLM-induced fibrotic peak at day 14 and during lung regeneration (day 28) and fibrosis resolution (day 60). **B** Hydroxyproline quantification on BLM ($n \geq 6$) or PBS ($n = 4$)-treated young or old mouse samples at indicated time points. Source data are provided as a Source Data file. Boxplot are represented with the median in the center, the whiskers correspond to the interquartile ranges, and the bounds correspond to the minimum and maximum values. * $P < 0.05$. **$P < 0.01$. P values were calculated by two-way ANOVA test followed by a multiple comparisons test with Holm-Šídák correction. **C** Histological sections of control and fibrotic lungs from young and old mice using H&E at indicated time points (representative images, $n = 3$). **D** Experimental design of spatial transcriptomics data from histological sections (H&E) of lungs from young ($n = 1$) and old ($n = 2$) mice challenged with BLM and collected at fibrotic peak (day 14, D14) and during regeneration (day 28, D28). Spots captured corresponding to tissue area were deconvoluted through ST deconvolve package in topics ($n = 14$). **E** Functional annotations of topics by IPA. Enrichment $p$-values obtained with IPA are calculated by right-tailed Fisher's Exact Test and Benjamini-Hochberg correction. **F** Average proportion of each topic across deconvoluted spots in each slice. **A, D** created in BioRender. MARI, B. (2025) https://BioRender.com/b0jq8np.

EC (SV EC), marked by the expression of Col15a1 and Ackr1 and which are rising up following lung injury (Fig. 2D–G and Table 1). Then, the subclustering of PCEC revealed the emergence of three additional capillary subpopulations in BLM-injured lungs. Among these sub-clusters, two subpopulations of gCap and aCap cells emerged in fibrotic lungs with recognizable signatures particularly characterized by the overexpression of Leucine-rich α−2 glycoprotein 1 (*Lrg1*) and mostly found at day 14 and day 28 following BLM treatment (Fig. 2D–F, Supplementary Fig. S3A and Table 1). We referred the third subcluster as sCap, which corresponds to a population of systemic capillary EC that has been recently characterized in fibrotic areas of COVID and IPF patients[17]. Indeed, this population expresses capillary markers, such as Aplnr, Rgcc, Kit or Sox11, as well as systemic markers, such as Col15a1, *Hspg2, Gja1, Vwa1, Lamb1, Spry1, Filip1, Ndrg1, Nr5a2, Galnt15* and *Meox1*, but lacks venous-specific markers (Fig. 2D–G and Tables 1, 2). RNA Fluorescence in situ hybridization (FISH) confirmed the co-expression of Col15a1 and Aplnr to clearly demonstrate the emergence of sCap in mouse lungs following BLM-induced injury (Fig. 2H). We hence considered these Col15a1[pos] sCap as a distinct PCEC sub-population for the rest of the study. In order to establish a link between the endothelial remodeling induced after lung injury in the mouse model and human IPF, we re-analyzed the pulmonary endothelial subset from the Habermann dataset[36] (IPF Cell atlas). We confirmed in this larger IPF dataset increase of both *COL15A1*[pos] sCap and SV EC in lungs from transplanted patients with IPF compared to healthy donors (Fig. 2I–K), which can also be observed using immunofluorescence (Fig. 2L). In contrast, we found only weak *LRG1* expression in these systemic EC and almost no expression in either gCap or aCap from IPF samples (Fig. 2J).

We then analyzed the profile of *Lrg1* expression in lung cell types in both the basal condition and after BLM treatment in young and aged mice. In physiological condition, *Lrg1* expression is restricted to the macrovascular *Col15a1*[pos] SV EC, PV EC, neu-trophils, large cavity macrophages (LCM), and to a lesser extent to AT2, gCap, and Lymphatic EC (Supplementary Fig. S3B). At the peak of fibrosis, *Lrg1* expression was dramatically increased in most EC subpopulations as well as in alveolar macrophages sub-sets (Supplementary Fig. S3B). We validated this increase of *Lrg1* in BLM-treated lungs by RNA FISH, confirming the emergence of Lrg1[pos]/Aplnr[pos] gCap and Lrg1[pos]/Ednrb[pos] aCap in lungs from BLM-treated mice (Supplementary Fig. S3C). We also validated the co-expression of Col15a1, Lrg1, and Aplnr by RNAscope™ FISH to clearly demonstrate the emergence of sCap in mouse lungs fol-lowing BLM-induced injury (Supplementary Fig. S3D). The kinet-ics of *Lrg1* expression in old versus young mice treated with BLM was similar in most subpopulations, apart for gCap, aCap and Arterial EC, for which *Lrg1* expression peak was time-shifted at day 28 (Supplementary Fig. S3E). Finally, overexpression of *Lrg1* in gCap and aCap was validated on an independent scRNA-seq dataset[37] in which lungs of adult mice treated with BLM were collected at a shorter time interval, indicating that *Lrg1* expres-sion is induced during the inflammatory phase following lung injury (Supplementary Fig. S3F).

## Lrg1[pos] PCEC signaling is associated with vascular remodeling and alveolar regeneration

*LRG1* is a gene coding for a pro-inflammatory glycoprotein, which can notably promote angiogenesis by switching TGF-β signaling from SMAD2/3 angiostatic signaling to SMAD1/5 angiogenic signaling[38]. To confirm this hypothesis, we overexpressed LRG1 in HMEC1 human endothelial cell line as well as in primary human pulmonary micro-vascular EC (HMVEC-L). Control and LRG1 vector transduced cells were stimulated or not with TGF-β1 at 2 different concentrations (3 or 10 ng/ml). While in control cells, TGF-β1 mostly induced phosphorylation of SMAD2/3, LRG1 expression preferentially promoted SMAD1/5 phos-phorylation at both low and high TGF-β1 concentrations in both models (Supplementary Fig. S4A, B). This switch was also accompanied by a significant increase of *VEGF* transcript (Supplementary Fig. S4C) and a significant increase in spheroid sprouting (Supplementary Fig. S4D–E), indicating that Lrg1[pos] PCEC signaling may participate in alveolar regeneration through induction of angiogenesis.

Ingenuity Pathway Analysis (IPA)™ performed on *Lrg1*[pos] PCEC markers indicated an activation of mTOR, TGF-β, integrin-linked kinase and wound healing canonical pathways in all 3 PCEC subpopulations (Fig. 3A). Enriched functions pointed to activation of angiogenesis or vasculogenesis, EC migration and proliferation in all 3 PCEC sub-populations, which is lacking or to a lesser extent in SV EC and PV EC (Fig. 3B). We also noted elevated expression of hypoxia-regulated genes with *Lrg1*[pos] aCap showing the strongest signature (Fig. 3C). Expression of markers associated with angiogenesis was shared by all three *Lrg1*[pos] PCEC subpopulations, in particular genes coding for ECM-associated proteins, transcription factors/regulators, cytokines/growth factors and receptors (Fig. 3D). Predicted upstream regulators analysis also pointed to the potential initiation role of hypoxia- (HIF1A, EPAS1/HIF2A) and inflammatory (NFkB, STAT6) -associated pathways in PCEC and also proposed additional candidates such as FoxM1, an important mediator of endothelial regeneration following chronic injury, whose expression was previously shown to be impaired in aged mice[39] (Supplementary Fig. S4F). RNAscope™ FISH confirmed the expression of HIF1α mRNA in *Lrg1*[pos] cells in fibrotic areas, as well as increased expression of *Vegfa* and *Sox17* in *Lrg1*[pos] gCaps at the peak of fibrosis (Fig. 3E), suggesting a central role for *Lrg1*[pos] PCEC in endo-thelial regeneration processes via intense signaling within the alveo-lar niche.

To follow up this hypothesis, we used NicheNet[40] to infer ligand-receptor interactions potentially associated with the expression of genes related to the pro-angiogenic signature of Lrg1[pos] PCEC. We selected the top ligands with the highest probability to induce these pro-angiogenic signatures in each Lrg1[pos] PCEC subpopulation. Most of the selected ligands were commonly prioritized in the 3 subpopula-tions and were known pro-angiogenic factors, such as *Fgf2, Vegfa, Il1b, Tgfb1, Apoe, Bmp4, Adam17* or *Cxcl12* (Supplementary Fig. S5A). We next looked at the expression of the selected ligands as well as the downstream genes most likely to be regulated by these ligands in the different lung cell populations. Notably, *Vegfa* and *Cxcl12* ligands were among the top predicted ligands as well as the top predicted downstream-regulated genes, suggesting autocrine signaling within

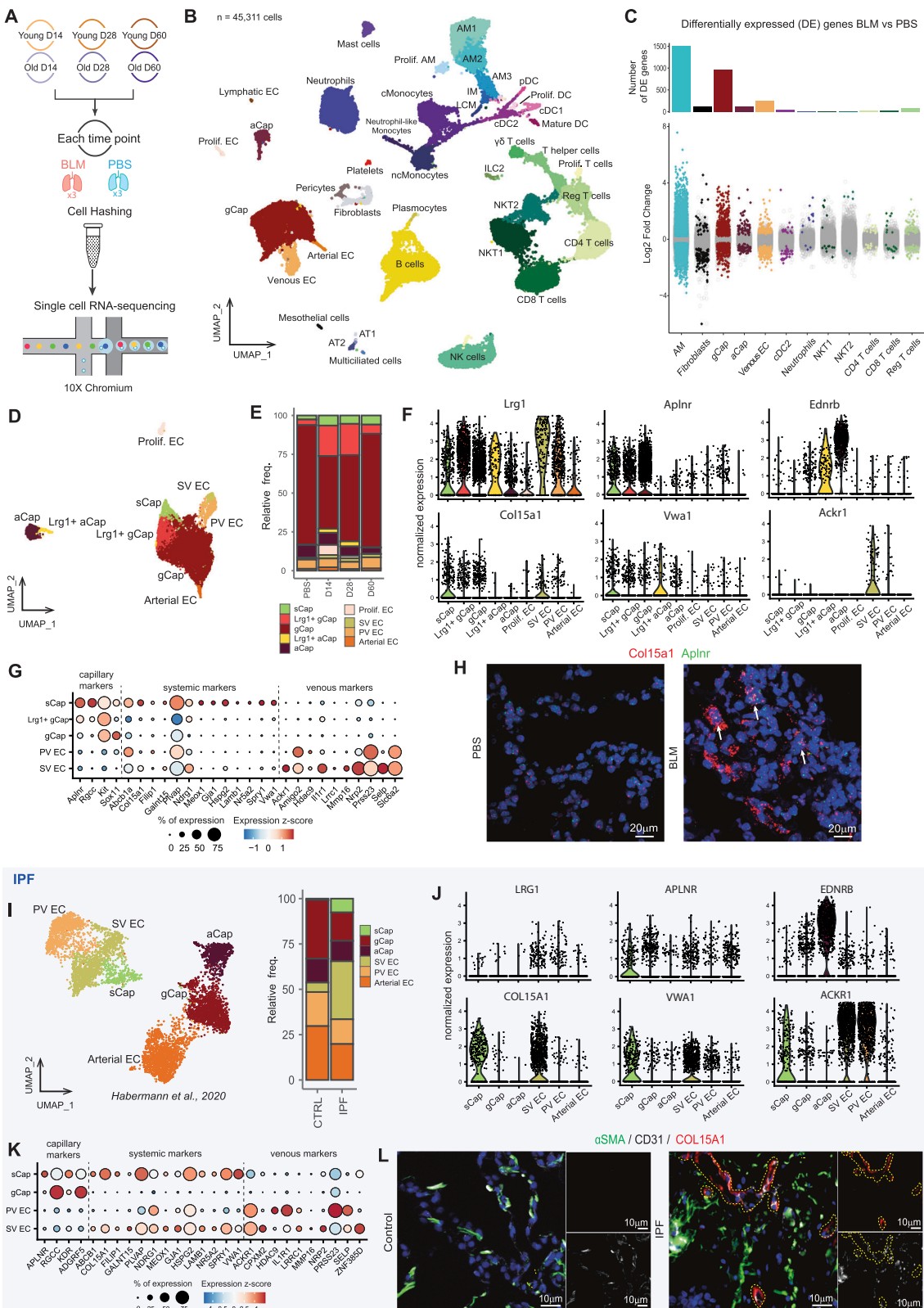

*Lrg1*[pos] PCEC (Supplementary Fig. S5A). We finally checked the expression of genes encoding bona fide receptors of the selected ligands in the 3 *Lrg1*[pos] PCEC subpopulations (Fig. 3F). Overall, their transcripts were equivalently detected in the 3 *Lrg1*[pos] EC subpopulations, with the exception of Fgf2 receptors Fgfr1 and Fgfr3, which are more specific to the *Lrg1*[pos] aCap. We have summarized these interaction inferences in a circos plot, linking the selected ligands grouped by

expression in specific cell populations to their top receptors expressed in *Lrg1*[pos] PCEC subpopulations (Fig. 3G). We represented potential paracrine and autocrine signaling involving ligands expressed by mast cells (*Hgf*), neutrophils (*Il1b*), macrophages (*Il1a*, *Adam17*, *Tnf*, and *Apoe*) as well as ligands expressed in multiple cell types such as *Tgfb1* and *Gpl1*. We also found potential interactions between ligands of neighboring cell types within the alveolar niche and *Lrg1*[pos] PCEC

**Fig. 2 | Time-resolved scRNA-seq captures PCEC subpopulations expressing Lrg1 in BLM-injured lungs from young and old mice. A** scRNA-seq experimental design (*n* = 3, for each experimental condition at each time point). **B** UMAP of the integrated dataset of the 45,311 sequenced cells. Cells are colored according to the annotated clusters. Prolif. AM = proliferating macrophages, AM = alveolar macrophages, IM = interstitial macrophages, LCM = large-cavity macrophages, Prolif. DC = proliferating dendritic cells, cDC = conventional dendritic cell, Mature DC = mature dendritic cells, pDC = plasmacytoid dendritic cells, cMonocytes/ncMonocytes = conventional/non-conventional monocytes, Prolif. T cells = proliferating T cells, Reg T cells = regulatory T cells, ILC2 = type 2 innate lymphoid cells, NKT = natural killer T cells, NK cells = natural killer, AT1/2 = Alveolar Type 1/2 cells, gCap = general capillary endothelial cells, aCap = aerocytes capillary endothelial cells, Prolif. EC = proliferating endothelial cells. **C** DEGs between BLM- and PBS-treated cells for indicated populations. The number of DEGs is limited to 1500 for the bar plot. **D** UMAP of pulmonary EC subpopulations. sCap = systemic capillary endothelial cells, SV EC = systemic venous endothelial cells, PV EC = pulmonary-venous endothelial cells. **E** Relative proportions of PCEC subpopulations across time points. (**F**) Normalized expression of *Lrg1, Aplnr, Ednrb, Col15a1, Vwa1* and *Ackr1* in mouse EC subpopulations. **G** Level and percentage of expression of capillary, systemic and venous markers in *Lrg1*[pos] PCEC subpopulations. **H** In situ hybridization of *Col15a1* and *Aplnr* mRNA in BLM or PBS conditions. Analysis of EC subpopulations in IPF (Habermann et al. dataset): UMAP and relative proportion of EC subpopulations (**I**); Normalized expression of *LRG1, APLNR, EDNRB, COL15A1, VWA1* and *ACKR1* in EC from IPF and control lungs (**J**); Level and percentage of expression of capillary, systemic and venous markers in human IPF EC subpopulations (**K**). **L** Expression of COL15A1 in lung sections from normal or IPF tissues by Immunofluorescence. Representative immunofluorescent images of COL15A1 (in red) with the pan EC marker CD31 (white) and αSMA (green). Nuclei are counterstained with DAPI. (*n* = 5).

receptors, such as the *Bmp4/Bmpr2* pair involving alveolar epithelial cells and the *Tgfb3/Tgfbr2* pair with mesenchymal cells. Based on the hypothesis that a receptor differentially expressed between physiological and fibrotic conditions likely indicates a modulation of the associated signaling pathway, we then compared receptor expression levels between gCap from BLM- versus PBS-treated mice. We observed that *Ackr3, Il1r1, Itgb1, Nrp2, Scarb1, Sdc4* and *Tgfbr2* were all upregulated while *Bmpr2* and *Tgfbr3* were downregulated in BLM-treated compared to control gCap (Supplementary Fig. S5B). These observations appear consistent in the context of capillary endothelial remodeling. For example, the autocrine signaling pathway involving *Cxcl12* and *Ackr3* in PCEC has been shown to drive AEC expansion and neoalveolarization by releasing Mmp14 following lung injury[41]. Our analysis indeed validated the upregulation of *Mmp14* in BLM-treated gCap at the single-cell transcriptomic level (Supplementary Fig. S5C). Finally, other genes driving key signal transduction proteins in EC, such as Eng, Id1, and several Smad family members, were also differentially expressed between both conditions in gCap (Supplementary Fig. S5C). In particular, we found that *Smad6* was significantly downregulated in BLM-treated gCap. Coupled with the repression of *Bmpr2* and a similar trend for other BMP9 signaling pathway-associated genes (*Acvrl1, Eng,* and *Id1*), we hypothesized that this pathway responsible for EC quiescence may be inhibited in gCap following lung injury to promote angiogenesis[42]. Taken together, these results converge towards proangiogenic signaling in *Lrg1*[pos] PCEC which, by releasing angiocrine factors, could thus contribute to the regeneration of the pulmonary alveolar niche.

## BLM-induced endothelial remodeling echoes pathological signatures found in human IPF

Using again the Habermann IPF dataset[36], we then systematically compared the mouse fibrotic signature induced by BLM with the IPF signature in systemic EC (SV EC and sCap), PV EC, gCap, and aCap (Fig. 3H and Supplementary table S6). The most conserved pathological signature in both models was found for sCap, with a common modulation of 505 genes that includes most of the pro-angiogenic signature enriched in mouse *Lrg1*[pos] PCEC (Table 2). Moreover, we observed a conserved upregulation of several signaling-associated genes, transcription factors, and ECM-associated genes. With regard to conserved downregulated genes, we also observed a common inhibition of multiple transcription factors, including FOXF1, SMAD6 and GATA2 as documented recently[43] as well as genes -associated with signaling, notably the BMP9 signaling genes (Table 2). These findings show that the pulmonary endothelial remodeling observed after BLM-induced lung injury in mice mimics at least in part the recruitment of systemic EC occurring in IPF, with a conserved signature associated with angiogenesis, ECM organization and cell adhesion.

## *Lrg1*[pos] PCEC dynamics are delayed in fibrotic aged mouse lungs

We then compared the dynamics of pulmonary EC after injury and during resolution between young and old mice (Fig. 4A), leveraging on their relative proportions found in each mouse of the Chromium scRNA-seq dataset. We observed a similar dynamics pattern for sCap, Lrg1[pos] gCap, Lrg1[pos] aCap, Prolif. EC and SV EC, characterized in young mice by a sharp increase of their abundances at D14 post BLM instillation, that progressively decreased during the resolution phase. In old mice, the abundance peak of these populations was systematically delayed at D28 and appeared weaker than their peak observed in young mice. We also observed an increase of PV EC and Arterial EC abundances in young mice at D14 but not in old. Moreover, the relative frequencies of both gCap and aCap remained the same than in physiological condition, suggesting that lung injury induces neovascularization of capillary EC, which is consistent with the detection of Prolif. EC. In young animals, this regenerative response appeared stronger, shorter and more broadly distributed to all endothelial vessel types.

We confirmed these observations at the gene-level by calculating a score based on the expression of upregulated genes following BLM-induced lung injury, which we compared between cells grouped by age and time point (Fig. 4B and Supplementary Fig. S6A-H).

We also validated this delayed dynamic in independent cohorts of mice using in situ hybridization assay. We first confirmed that the increase of Col15a1 in young mice was restricted to day 14, while in old mice we still observed its expression at day 28 and day 60 (Fig. 4C). Similarly, the percentage of *Lrg1*[pos]/*Aplnr*[pos] cells peaked at day 14 following BLM treatment and progressively decreased to reach their basal level at day 60, whereas a strong percentage was still maintained at day 28 and day 60 in old mice (Fig. 4D). A similar progression profile was observed for *Lrg1*[pos]/*Ednrb*[pos] cells, likely corresponding to *Lrg1*[pos] aCap.

Finally, we returned to the spatial transcriptomic data to look for evidence of physiological and pathological PCEC signatures in spatial topics potentially associated with Lrg1 expression. Topic 6, highly represented in the young lung section at day 28, was enriched in physiological gCap and aCap markers (*Sema3c, Hpgd, Thbd, Cldn5,* and *Car4*), as well as AT1 markers (*Ager, Hopx, Cldn18*), suggesting successful restoration of alveolar functions (Fig. 1F, Supplementary Fig. S1B, C). *Lrg1* was found in topics 1 and 4, where it was associated with fibroblast markers rather than endothelial markers. Interestingly, it was also enriched in topic 13, which displayed a clear inflammatory signature (Fig. 1E), notably carried by cytokines, complement, and MHC components. This topic persisted at day 28 in both lung sections from old mice (representing 6% and 10% of deconvoluted spots, respectively), whereas it tended to be cleared (2% of deconvoluted spots) in the section from young mice at the same time point.

Taken together, our results showed that the dynamics of Lrg1[pos] PCEC cells are delayed in lungs from aged mice, paralleling the retarded resolution of lung fibrosis observed in aged animals.

## Table 1 | Main markers of Lrg1pos PCEC and SV EC subpopulations

| Subpopulation | Marker | Description |
|---|---|---|
| Lrg1pos aCap | Adora2a | adenosine A2a receptor |
| | Bdkrb2 | bradykinin receptor, beta 2 |
| | Bnip3 | BCL2/adenovirus E1B interacting protein 3 |
| | Ccdc184 | coiled-coil domain containing 184 |
| | Dnah11 | dynein, axonemal, heavy chain 11 |
| | Fgfr1 | fibroblast growth factor receptor 1 |
| | Meox1 | mesenchyme homeobox 1 |
| | Ncam1 | neural cell adhesion molecule 1 |
| | Serpine1 | serine (or cysteine) peptidase inhibitor, clade E, member 1 |
| | Tnfrsf11b | tumor necrosis factor receptor superfamily, member 11b (osteoprotegerin) |
| Lrg1pos aCap Lrg1pos gCap | Ackr3 | atypical chemokine receptor 3 |
| | Adam15 | ADAM metallopeptidase domain 15 |
| | Ankrd37 | ankyrin repeat domain 37 |
| | Gadd45g | growth arrest and DNA-damage-inducible 45 gamma |
| | Ntrk2 | neurotrophic tyrosine kinase, receptor, type 2 |
| | Tgfbr2 | transforming growth factor, beta receptor II |
| Lrg1pos gCap | Adamts1 | ADAM metallopeptidase with thrombospondin type 1 motif 1 |
| | Cmah | cytidine monophospho-N-acetylneuraminic acid hydroxylase |
| | Cxcl12 | C-X-C motif chemokine ligand 12 |
| | Scn3b | sodium channel, voltage-gated, type III, beta |
| | Tiam1 | T cell lymphoma invasion and metastasis 1 |
| sCap | Abcb1a | ATP-binding cassette, sub-family B member 1 A |
| | Ccnd1 | cyclin D1 |
| | Fscn1 | fascin actin-bundling protein 1 |
| | Gja1 | gap junction protein, alpha 1 |
| | Hspg2 | perlecan (heparan sulfate proteoglycan 2) |
| | Ivns1abp | influenza virus NS1A binding protein |
| | Lamb1 | laminin B1 |
| | Nr5a2 | nuclear receptor subfamily 5, group A, member 2 |
| | Rgcc | regulator of cell cycle |
| | Spry1 | sprouty RTK signaling antagonist 1 |
| | Trp53i11 | transformation related protein 53 inducible protein 11 |
| sCap Lrg1pos aCap | F2r | coagulation factor II thrombin receptor |
| | Igfbp7 | insulin-like growth factor binding protein 7 |
| | Vwa1 | von Willebrand factor A domain containing 1 |
| sCap Lrg1pos aCap Lrg1pos gCap | Cd34 | CD34 antigen |
| | Col4a1 | collagen, type IV, alpha 1 |
| | Col4a2 | collagen, type IV, alpha 2 |
| | Emp1 | epithelial membrane protein 1 |
| | Hif1a | hypoxia inducible factor 1, alpha subunit |
| | Ndrg1 | N-myc downstream regulated gene 1 |
| | Scarb1 | scavenger receptor class B, member 1 |
| | Sparc | secreted acidic cysteine rich glycoprotein |
| | Sparcl1 | SPARC-like 1 |
| | Tmem176a | transmembrane protein 176 A |
| | Tmem176b | transmembrane protein 176B |
| sCap Lrg1pos aCap Lrg1pos gCap SVEC | Lrg1 | leucine-rich alpha-2-glycoprotein 1 |
| sCap Lrg1pos gCap SVEC | Atp8b1 | ATPase, class I, type 8B, member 1 |
| | Mctp1 | multiple C2 domains, transmembrane 1 |
| | Plac8 | placenta-specific 8 |

## Table 1 (continued) | Main markers of Lrg1pos PCEC and SV EC subpopulations

| Subpopulation | Marker | Description |
|---|---|---|
| sCap SVEC | Col15a1 | collagen, type XV, alpha 1 |
| | Nrp2 | neuropilin 2 |
| | Plxnd1 | plexin D1 |
| | Rflnb | refilin B |
| SVEC | Ackr1 | atypical chemokine receptor 1 (Duffy blood group) |
| | Amigo2 | adhesion molecule with Ig like domain 2 |
| | Il1r1 | interleukin 1 receptor, type I |
| | Mmp16 | matrix metallopeptidase 16 |
| | Selp | selectin, platelet |

## Table 2 | Commonly regulated genes in sCap from IPF samples and from BLM model

| Gene | Description | Expression in fibrosis |
|---|---|---|
| ACKR4 | atypical chemokine receptor 4 | downregulated |
| ADAMTS7 | ADAM metallopeptidase with thrombospondin type 1 motif 7 | upregulated |
| ADRB2 | adrenoceptor beta 2 | downregulated |
| ANKRD33B | ankyrin repeat domain 33B | downregulated |
| APLNR | apelin receptor | upregulated |
| APRT | adenine phosphoribosyltransferase | upregulated |
| BACE2 | beta-secretase 2 | upregulated |
| BDKRB2 | bradykinin receptor B2 | upregulated |
| CAMK1 | calcium/calmodulin dependent protein kinase I | downregulated |
| CD34 | CD34 molecule | upregulated |
| CDA | cytidine deaminase | upregulated |
| CDKN1B | cyclin dependent kinase inhibitor 1B | downregulated |
| CEBPB | CCAAT enhancer binding protein beta | downregulated |
| CLK4 | CDC like kinase 4 | downregulated |
| COL15A1 | collagen type XV alpha 1 chain | upregulated |
| COL18A1 | collagen type XVIII alpha 1 chain | upregulated |
| COL4A1 | collagen type IV alpha 1 chain | upregulated |
| COL4A2 | collagen type IV alpha 2 chain | upregulated |
| COTL1 | coactosin like F-actin binding protein 1 | upregulated |
| DBN1 | drebrin 1 | upregulated |
| DDHD2 | DDHD domain containing 2 | downregulated |
| DNAJB1 | DnaJ heat shock protein family (Hsp40) member B1 | downregulated |
| EEF1B2 | eukaryotic translation elongation factor 1 beta 2 | upregulated |
| ENPP4 | ectonucleotide pyrophosphatase/phosphodiesterase 4 | downregulated |
| EPAS1 | endothelial PAS domain protein 1 | downregulated |
| EPB41 | erythrocyte membrane protein band 4.1 | downregulated |
| EVL | Enah/Vasp-like | upregulated |
| FKBP10 | FKBP prolyl isomerase 10 | upregulated |
| FXN | frataxin | upregulated |
| GALNT1 | polypeptide N-acetylgalactosaminyltransferase 1 | upregulated |
| GATA2 | GATA binding protein 2 | downregulated |
| GFOD1 | Gfo/Idh/MocA-like oxidoreductase domain containing 1 | downregulated |
| GJA1 | gap junction protein alpha 1 | upregulated |
| GPATCH4 | | upregulated |

**Table 2 (continued) | Commonly regulated genes in sCap from IPF samples and from BLM model**

| Gene | Description | Expression in fibrosis |
|------|-------------|------------------------|
|  | G-patch domain containing 4 (gene/pseudogene) |  |
| GPIHBP1 | glycosylphosphatidylinositol anchored high density lipoprotein binding protein 1 | downregulated |
| HCLS1 | hematopoietic cell-specific Lyn substrate 1 | upregulated |
| HIF1A | hypoxia inducible factor 1 subunit alpha | upregulated |
| HIPK2 | homeodomain interacting protein kinase 2 | downregulated |
| HLA-E | major histocompatibility complex, class I, E | downregulated |
| HOXA5 | homeobox A5 | downregulated |
| IGFBP7 | insulin like growth factor binding protein 7 | upregulated |
| INHBB | inhibin subunit beta B | upregulated |
| JUN | Jun proto-oncogene, AP-1 transcription factor subunit | downregulated |
| KCNJ15 | potassium inwardly rectifying channel subfamily J member 15 | upregulated |
| KLF2 | KLF transcription factor 2 | downregulated |
| KLF3 | KLF transcription factor 3 | downregulated |
| KLF4 | KLF transcription factor 4 | downregulated |
| LAMA4 | laminin subunit alpha 4 | upregulated |
| LAMC1 | laminin subunit gamma 1 | upregulated |
| LIMS2 | LIM zinc finger domain containing 2 | downregulated |
| LMNB1 | lamin B1 | upregulated |
| MAP1B | microtubule associated protein 1B | upregulated |
| MCC | MCC regulator of WNT signaling pathway | downregulated |
| MEOX1 | mesenchyme homeobox 1 | upregulated |
| METTL1 | methyltransferase 1, tRNA methylguanosine | upregulated |
| MRTO4 | MRT4 homolog, ribosome maturation factor | upregulated |
| MTHFD1L | methylenetetrahydrofolate dehydrogenase (NADP+ dependent) 1 like | upregulated |
| MYO1B | myosin IB | upregulated |
| NCOA3 | nuclear receptor coactivator 3 | downregulated |
| NKTR | natural killer cell triggering receptor | downregulated |
| NME1 | NME/NM23 nucleoside diphosphate kinase 1 | upregulated |
| NOP16 | NOP16 nucleolar protein | upregulated |
| NPM3 | nucleophosmin/nucleoplasmin 3 | upregulated |
| NR5A2 | nuclear receptor subfamily 5 group A member 2 | upregulated |
| NREP | neuronal regeneration related protein | upregulated |
| NRGN | neurogranin | upregulated |
| P2RY6 | pyrimidinergic receptor P2Y6 | upregulated |
| PAFAH1B3 | platelet activating factor acetylhydrolase 1b catalytic subunit 3 | upregulated |
| PDLIM4 | PDZ and LIM domain 4 | upregulated |
| PLPP2 | phospholipid phosphatase 2 | upregulated |
| PPP1R14B | protein phosphatase 1 regulatory inhibitor subunit 14B | upregulated |
| PPP1R15A | protein phosphatase 1 regulatory subunit 15A | downregulated |
| PRR5L | proline rich 5 like | upregulated |
| PTPRB | protein tyrosine phosphatase receptor type B | downregulated |

**Table 2 (continued) | Commonly regulated genes in sCap from IPF samples and from BLM model**

| Gene | Description | Expression in fibrosis |
|------|-------------|------------------------|
| RAB13 | RAB13, member RAS oncogene family | upregulated |
| RAI14 | retinoic acid induced 14 | upregulated |
| RHOD | ras homolog family member D | upregulated |
| RPL10A | ribosomal protein L10a | upregulated |
| RPL22L1 | ribosomal protein L22 like 1 | upregulated |
| RPS18 | ribosomal protein S18 | upregulated |
| RPS3 | ribosomal protein S3 | upregulated |
| RPS6 | ribosomal protein S6 | upregulated |
| RUNX1T1 | RUNX1 partner transcriptional co-repressor 1 | downregulated |
| SECISBP2L | SECIS binding protein 2 like | downregulated |
| SEMA6B | semaphorin 6B | upregulated |
| SH2B2 | SH2B adaptor protein 2 | upregulated |
| SHE | Src homology 2 domain containing E | downregulated |
| SKIL | SKI like proto-oncogene | downregulated |
| SMAD3 | SMAD family member 3 | upregulated |
| SMAD7 | SMAD family member 7 | downregulated |
| SMOX | spermine oxidase | upregulated |
| SNX18 | sorting nexin 18 | downregulated |
| SPRY1 | sprouty RTK signaling antagonist 1 | upregulated |
| SRM | spermidine synthase | upregulated |
| ST6GALNAC3 | ST6 N-acetylgalactosaminide alpha-2,6-sialyltransferase 3 | downregulated |
| STK4 | serine/threonine kinase 4 | downregulated |
| SYNE1 | spectrin repeat containing nuclear envelope protein 1 | downregulated |
| SYNPO | synaptopodin | upregulated |
| TEK | TEK receptor tyrosine kinase | downregulated |
| THBS1 | thrombospondin 1 | upregulated |
| TLE4 | TLE family member 4, transcriptional corepressor | downregulated |
| TMEM50B | transmembrane protein 50B | downregulated |
| TMTC1 | transmembrane O-mannosyltransferase targeting cadherins 1 | downregulated |
| TRIB1 | tribbles pseudokinase 1 | upregulated |
| TSHZ2 | teashirt zinc finger homeobox 2 | upregulated |
| TSPAN5 | tetraspanin 5 | upregulated |
| TYMS | thymidylate synthetase | upregulated |
| UBTD1 | ubiquitin domain containing 1 | upregulated |
| UCK2 | uridine-cytidine kinase 2 | upregulated |
| VWA1 | von Willebrand factor A domain containing 1 | upregulated |
| ZMYND11 | zinc finger MYND-type containing 11 | downregulated |
| ZNF593 | zinc finger protein 593 | upregulated |

## Aged mouse lung gCap exhibit pro-inflammatory and injury-activated phenotype

We next performed differential expression analyses between old and young gCap in fibrotic or control conditions. In fibrotic conditions, we observed an enriched pro-inflammatory signature in old gCap, mainly characterized by the overexpression of genes encoding for MHC class II molecules, *Cd74*, *Cd52*, and the interferon-induced protein Gbp4 (Fig. 5A, B and Supplementary Table S4). The expression of *Cd74*, the receptor of Macrophage migration Inhibitory Factor (Mif), a critical upstream regulator of the immune response[44] was validated by RNAScope™ FISH, showing a stronger signal of *Cd74* in lungs from

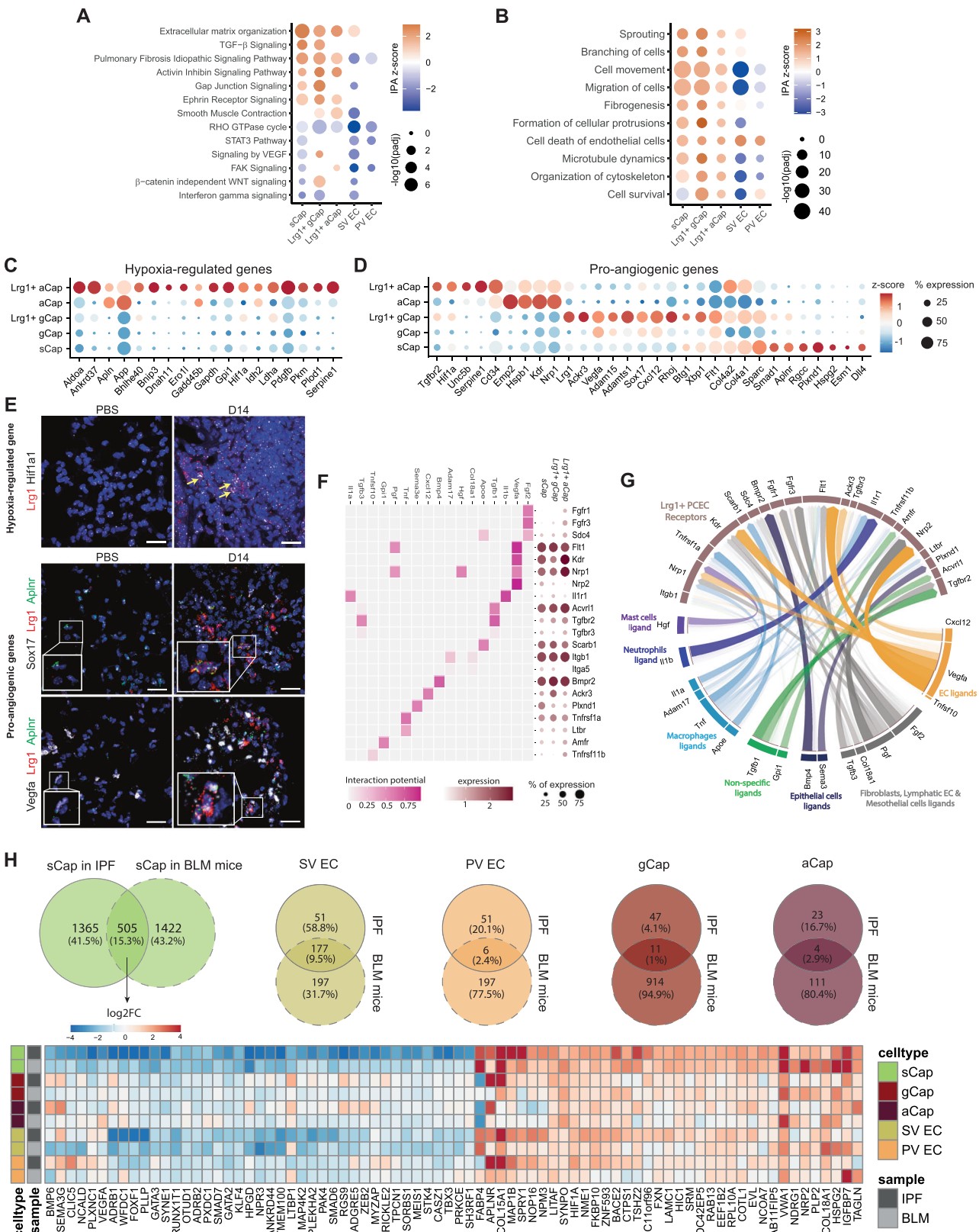

BLM-treated aged mice as well as partial co-localization of *Cd74* and *Lrg1* (Fig. 5C). Conversely, we found in young gCap an enrichment of transcription factors associated with vascular homeostasis and repair such as *Klf2, Klf10 and Peg3*[45–47] (Fig. 5B and Supplementary table S4). In PBS-treated mice, comparison between old and young gCap revealed even more gene modulations (Fig. 5D, E), which were

associated with multiple functions, including an increase of inflammatory and immune responses, but also ECM organization and IPF signaling pathway in aged mice (Fig. 5F and Supplementary Table S5). Indeed, this aging signature contained 21 genes significantly deregulated in gCap following BLM challenge, including the top *Lrg1*[pos] PCEC markers *Lrg1, Col15a1, Nrtk2, Ndrg1*, and *Ankrd37* (Fig. 5G), indicating

**Fig. 3 | Lung injury shift pulmonary endothelial cells towards *Lrg1*^pos sub-populations associated with lung alveolar niche regeneration. A** Selection of top activated canonical pathways in *Lrg1*^pos PCEC and venous EC subpopulations. **B** Selection of top activated functions in *Lrg1*^pos PCEC and venous EC subpopulations. Enrichment p-values obtained with IPA are calculated by right-tailed Fisher's Exact Test and Benjamini-Hochberg correction. Level and percentage of expression of hypoxia-regulated (**C**) and pro-angiogenic (**D**) genes in *Lrg1*^pos PCEC and venous EC subpopulations. **E** RNA FISH acquisition for *Hif1a1* (black, top), *Sox17* (black, middle), *Vegfa* (black, down), *Aplnr* (green) and *Lrg1* (red) mRNA in PBS or

D14 (BLM) in young mice. Scale bars = 20 μm, *n* = 3 independent mice and 5 field/mouse were captured. **F** Interaction potential between the prioritized ligands and their bona fide receptors (left); Level and percentage of expression of each receptor in *Lrg1*^pos PCEC subpopulations (right). **G** Circos plot of ligands-receptors interactions predicted by NicheNet analysis. Ligands are colored according to their specific expression. **H** Comparison of human and mouse pulmonary fibrosis signatures in sCap, SV EC, PV EC, gCap and aCap. Log2FC of a selection of genes that are similarly modulated by fibrosis in murine and human sCap.

that old gCap in physiological condition already expresses a pro-inflammatory and injury-activated signature. We also observed that old gCap overexpressed *Vwf, Slc6a2, Prss23 and Plat* (Fig. 5D, E), which are normally restricted to macrovascular EC (Supplementary Fig. S3A). These cells also show a decrease in *Aplnr* expression compared to young gCap, in line with a predicted decrease in the Apelin signaling pathway in old gCap (Fig. 5F), strongly suggesting a potential change in their intrinsic functions.

## Trajectory inference indicates a transition between *Col15a1*^pos sCap, *Lrg1*^pos gCap, and physiological gCap following alveolar injury

The parallel kinetic profile between the *Lrg1*^pos PCEC populations after BLM treatment, as well as their transcriptomic proximity and delayed appearance in aged animals, correlating with a delayed resolution, suggests that a dynamic process links these different PCEC subpopulations or cell states during alveolar niche regeneration. To model a potential trajectory, we used the generalized RNA velocity approach[48], leveraging on the splicing dynamics estimated in young BLM and PBS-treated mice samples at day 14. Latent time ordering indicated that *Col15a1*^pos sCap could initiate a differentiation process, transiting through *Lrg1*^pos gCap before ending up to physiological gCap (Fig. 6A). Of note, the same trajectory was proposed in the same configuration in the old mice 28 days after instillation (Fig. 6B). This inferred trajectory was supported by genes displaying a dynamic expression profile according to latent time, either progressively decreasing (*Smad1, Sparc, Col4a1, Ndrg1, Igfbp7*), transiently upregulated (*Ackr3, Emp1, Cmah, Vwf, Smad7*), or progressively increasing (*Npr3, Hpgd, Clec1a, Bmp6, Itga1*) (Fig. 6C,D and Supplementary Fig. S7A). These data suggest that in the BLM reversible fibrosis model, sCap are recruited to contribute to gCap replenishment and to the regeneration of the lung capillary endothelium integrity. Finally, we also tried to use this conserved gene signature to capture information on the molecular mechanisms associated with this differentiation process. The main predicted upstream regulators candidates correspond to hypoxia-induced genes such as Sox17, Cxcl8, Yap1, and Gata6 with pro-survival, pro-angiogenic, and/or pro-inflammatory activities (Supplementary Fig. S7B, C). Of note, the model also predicted the switch of the TGF-β signaling from SMAD2/3 angiostatic signaling to SMAD1/5 angiogenic signaling associated with Lrg1 function (Supplementary Fig. 4).

## Discussion

In this work, we used scRNA-seq combined with spatial transcriptomics to follow the transcriptomic modifications of lung cell populations during fibrosis formation and resolution in a mouse model of lung fibrosis. We specifically focused on the transcriptomic alterations correlated with a delayed fibrosis resolution in aged mice and identified several aging-associated changes in some PCEC subpopulations, including aCap and gCap, shifting towards a pro-angiogenic phenotype expressing *Lrg1* and associated with alveolar niche regeneration. We also found that aging altered the transcriptome of gCap in control mice, with a signature suggesting an inflammatory and injury-activated state. Because PCECs are intimately connected to alveolar epithelial cells and fibroblasts in the alveolus, age-associated defects in the PCEC differentiation may likely

participate in the dysfunction of other cells in the alveolar niche and impede alveolar regeneration.

Our study presents a whole picture of cell dynamic changes in lung from young and aged animals during fibrosis formation and resolution at 3 time points. Using several approaches, we confirmed previous works showing that aged mice exhibit impaired resolution after BLM injury[25–27] and further propose using spatial transcriptomics that infiltration of immune cells, such as plasma cells, known in IPF to form aggregates and bronchus-associated lymphoid tissue (BALT)[31], contributes to the delayed resolution observed in aged mice. We also confirmed by scRNA-seq that many cell populations are affected by BLM, including AEC, fibroblasts, macrophages, and PCEC[28,34,35,49], providing a useful dataset to analyze the transcriptomic dynamics of these populations and compared it to the kinetics of the resolution process measured with histology, collagen production and spatial transcriptomic.

One of the most affected cell types by the BLM challenge in our dataset corresponded to EC, with several main subpopulations that were only apparent in BLM-injured lungs. First, we identified homologous Col15a1-positive systemic EC, previously described in lungs from IPF patients[15]. Further analyses showed that these systemic EC corresponded to two distinct populations, one with venous macrovascular origin previously defined as bronchial venous EC and a second one lacking typical macrovascular and venous markers, likely corresponding to systemic capillary EC (sCap), in agreement with a recent study in COVID-19 and IPF patients[17]. Secondly, we found two subpopulations of gCap and aCap cells in fibrotic lungs sharing a partial common gene signature characterized by the overexpression of *Lrg1*. Interestingly, among these distinct populations, only *Lrg1*^pos gCap and *Lrg1*^pos aCap cells displayed a different kinetic profile following BLM challenge in young and aged mice. While they both peaked at day 14 in young mice lungs, their dynamic was strongly shifted in the lungs of aged mice, peaking at day 28. Overall, our findings demonstrate that BLM injury induces PCEC remodeling marked by the upregulation of *Lrg1* and the emergence of distinct capillary activated cell states that are delayed in aged mice.

*Lrg1* encodes for a secreted member of the family of leucine-rich repeat proteins, which often served as pattern recognition motifs for the innate immune system[50]. It is a multifunctional signaling molecule initially associated with pathological angiogenesis[38] that can notably modulate the TGF-β pathway in a highly context-dependent manner. The physiological function of *Lrg1* remains poorly understood, with knockout mice showing no obvious phenotypic defects. Several studies indicate that it promotes physiological wound healing and maintains tissue homeostasis[51,52], although abnormal expression levels have been found to disturb effective wound healing and contribute to fibrosis in several tissues including lung[53]. These data suggest that *Lrg1* expression must be finely regulated during the repair process and that abnormal levels may disrupt effective wound healing and contribute to fibrogenesis. One of the most compelling examples of LRG1 function in promoting diseased vessels is commonly described as the "TGF-β angiogenic switch", which switches TGF-β signaling to a proliferative pathway in a variety of pathological settings, including age-related macular degeneration and cancer[38]. While binding of TGF-β to endothelial TGF-β type II receptor (TGFβRII) normally initiates signaling

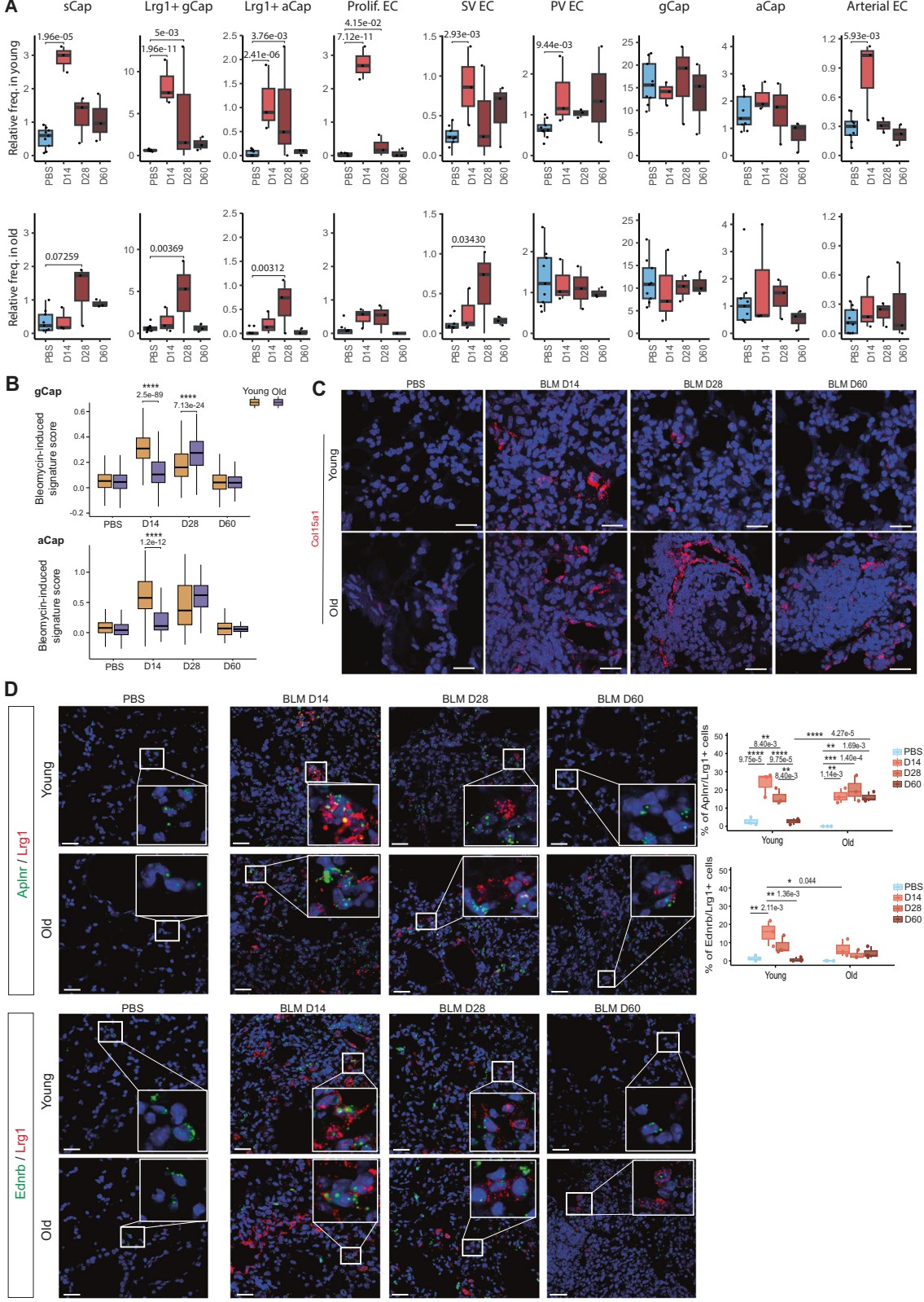

**Fig. 4 | *Lrg1*<sup>pos</sup> PCEC dynamics are delayed in old mice. A** Pulmonary endothelial subpopulations relative frequencies in young and old mice across time points. **B** BLM-induced signature score (see material and methods section) in gCap (top) and aCap (bottom) in young and old mice across time points. **C** In situ hybridization of *Col15a1* mRNA at indicated time points in young and old mice. Scale bars = 20 μm. **D** RNA FISH of *Lrg1* mRNA with gCap marker *Aplnr* mRNA or aCap marker *Ednrb* mRNA at indicated time points in young and old mice. (*n* = 3 independent mice, 3

counted microscopic fields/mouse, scale bars = 20 μm). Quantification was performed in three different fields for each mouse. *P < 0.05, **P < 0.01, ***P < 0.001, ****P < 0.0001. P-values were calculated by a Two-way ANOVA test followed by a multiple comparisons test with Holm-Sidak correction. Boxplot are represented with the median in the center, the whiskers correspond to the interquartile ranges, and the bounds correspond to the minimum and maximum values. Source data are provided as a Source Data file.

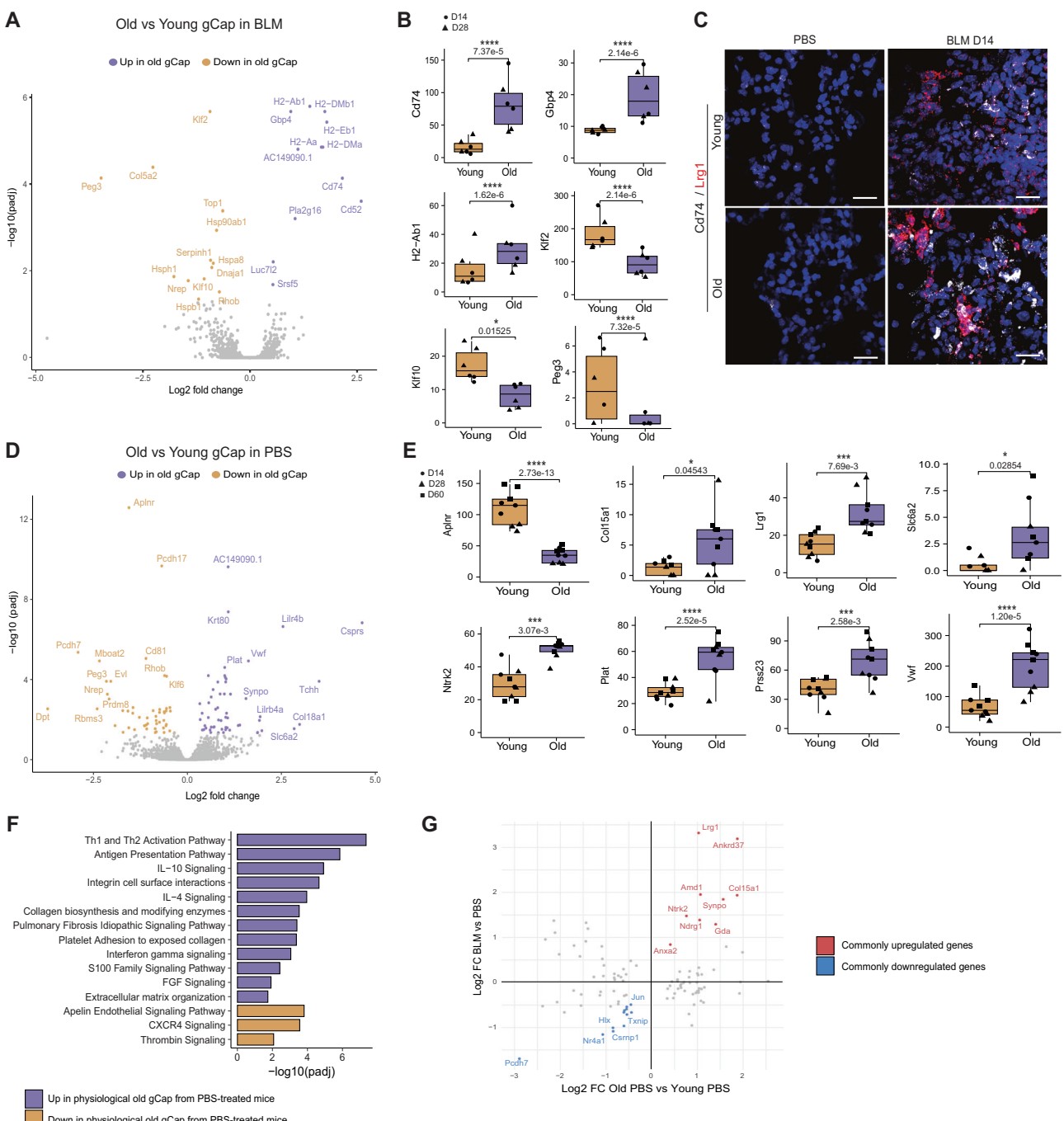

**Fig. 5 | Aged mouse lung gCap exhibits pro-inflammatory and injury-activated phenotype. A** Differentially expressed genes (DEGs) between old and young gCap in fibrotic condition. **B** Expression of several genes dysregulated in age-related in gCap under BLM conditions (*Cd74, Gbp4, H2-Ab1, Klf2, Klf10* and *Peg3*). Boxplot of relative expression, each point corresponds to one mouse (n = 3 by condition, ● for D14 ▲ for D28). **C** In situ hybridization of *Cd74* mRNA in BLM or Day 14 after BLM induction in young and old mice. Scale bars = 20 μm. **D** DEGs between old and young gCap in physiological condition. **E** Pseudobulk expression of several genes dysregulated by ageing in gCap under PBS conditions (*Aplnr, Col15a1, Lrg1, Slc6a2, Ntrk2, Plat, Prss23* and *Vwf*). Each point corresponds to the aggregated expression

in one mouse (n = 3 by condition, ● for D14, ▲ for D28 and ■ for D60). **F** Function enrichment analysis on DEGs between old and young physiological gCap. **G** Comparison of the log₂FC obtained by comparing physiological gCap of old and young mice, and the corresponding log₂FC between gCap of BLM and PBS-treated mice. Boxplot are represented with the median in the center, the whiskers correspond to the interquartile ranges, and the bounds correspond to the minimum and maximum values. Source data are provided as a Source Data file. Statistic: P-values were calculated by the Wald test and the Benjamini-Hochberg method for multiple tests correction.

through the tyrosine kinase receptor ALK5 coupled to SMAD2/3, preserving cell quiescence, high levels of LRG1 can redirect TGF-β to form a transduction complex with ALK1 and the accessory receptor endoglin (ENG), activating the pro-angiogenic SMAD1/5/8 pathway and promoting EC proliferation, migration, and tubulogenesis as well as the expression of several pro-angiogenic factors such as VEGFA. Our

in vitro experiments confirmed this molecular mechanism, which likely explains our data showing inhibition of endothelial quiescence in gCap *Lrg1*ᵖᵒˢ as well as strong induction of markers associated with EC proliferation and migration after lung injury to promote angiogenesis and sprouting. During the preparation of this manuscript, Zhao et al. indeed demonstrated that *Lrg1* was able to counterbalance the

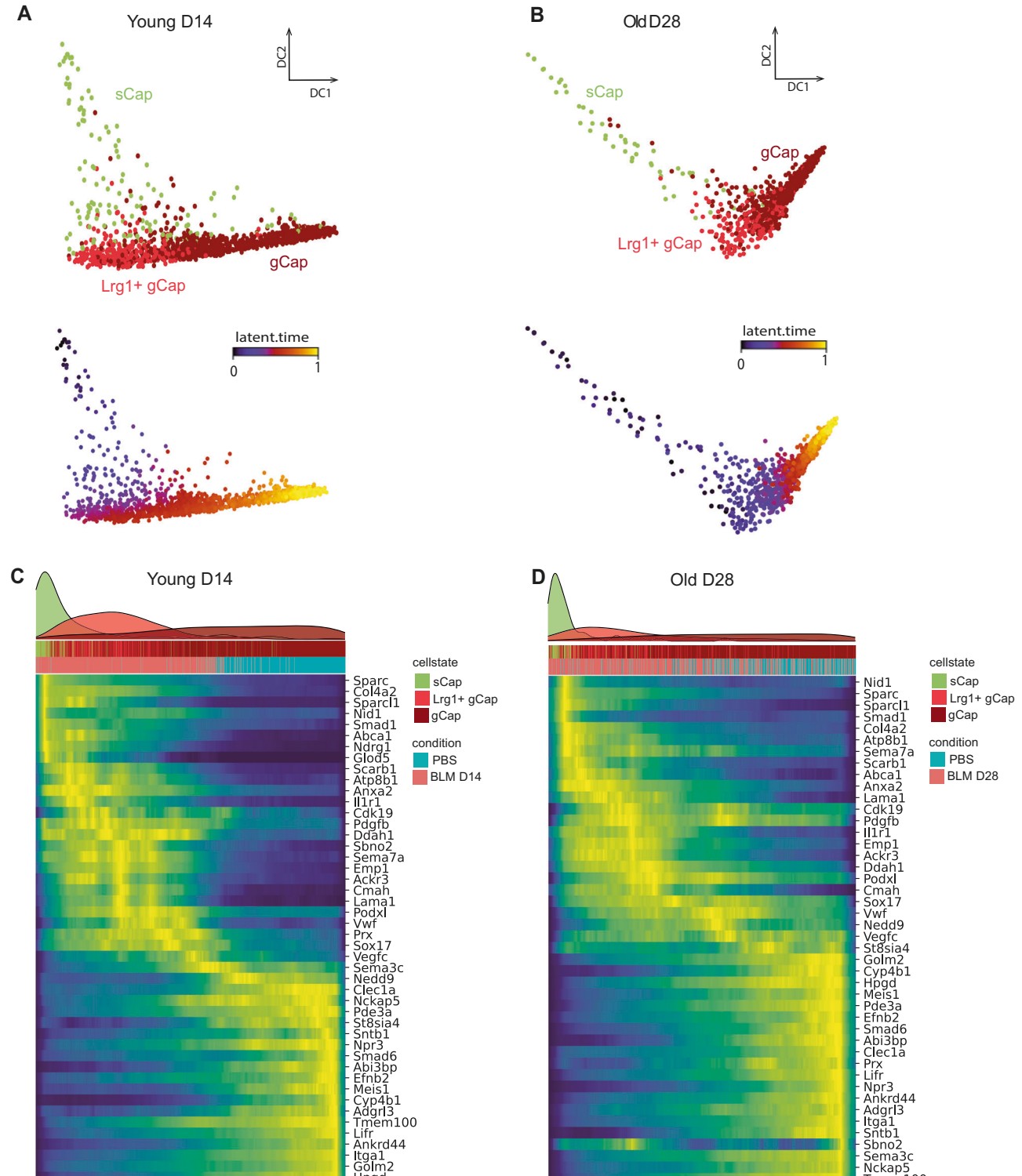

**Fig. 6 | RNA velocity analysis reveals transcriptional dynamics of PCEC during alveolar regeneration.** Diffusion map computed on Young D14 (**A**) or Old D28 (**B**) mouse spliced RNA data (3 BLM and 3 PBS-treated mice), with cells colored according to PCEC subpopulation (up) and RNA velocity latent time (bottom). Heatmap of the top contributing genes to the transcriptional dynamic, with cells ordered by latent time in young D14 (**C**) or old D28 (**D**) mice.

canonical angiostatic TGFβRII signaling to promote lung vascular repair following viral injury[54]. While this *Lrg1*-associated pro-angiogenic activity appears to be necessary to promote alveolar vascular repair, it is likely that delayed expression of *Lrg1* in aged mice could slow down the formation of a new mature capillary network thus causing a delay in the regeneration of the alveolar niche.

The *Lrg1*-associated gene signature found in our study also contains *Sox17*, a member of the Sry-related high mobility group domain family F (Sox F) transcription factors, a key developmental regulator of EC lineage. Previous work indicated that the induction of *Sox17* expression through the activation of HIF-1α in EC plays an essential role in regenerating EC following endotoxin-induced EC injury, notably

by transcribing Cyclin E1 and stimulating the proliferation of EC[55]. Of note, both *SOX17* and *HIF1A* were also detected in PCEC in lungs from IPF patients, indicative of a pro-angiogenic activity. Overexpression of these two genes was validated in fibrotic areas of mouse lungs, with co-localization with *Aplnr* or *Lrg1*, respectively. Activation of a hypoxic response was also particularly visible in *Lrg1*pos aCap and *Lrg1*pos gCap with several markers associated with glycolysis and hypoxia, in agreement with studies indicating that transient activation of hypoxia-mediated signaling pathways contributed to angiogenesis and vascular repair, while chronic hypoxia due to defective vascular remodeling leads to impaired tissue repair and induction of fibrosis[56–59]. In addition to hypoxia, our analysis revealed transcriptomic modulations of mTOR, TGF-β, IL-8, and integrin-linked kinase signaling and notably related to vasculogenesis, sprouting, cell-cell contact, cell migration, and inflammation. The identification of multiple cytokines/growth factors and receptors in activated PCEC suggested extensive heterotypic interactions within the alveolar niche. Our ligand-receptor interaction predictions propose a complex network of interactions involving ligands expressed by activated PCEC and neighboring cell types within the alveolar niche, including AT1, AT2, mesenchymal cells as well as mast cells, neutrophils and macrophages that likely influence alveolar repair. Remarkably, some of these predictions are in agreement with previous studies, highlighting in particular the importance of the signaling pathway involving *Cxcl12* (also known as *Sdf1*) and *Ackr3* (*Cxcr7*) in PCEC, leading to neo-alveolarization[41]. Further work is needed to analyze the functional importance of this predicted network and to characterize its central regulators with the goal to potentially targeting specific signaling pathways within the vascular niche for regenerative therapies.

We also found additional aging-related transcriptomic differences in PCEC subpopulations in physiological or pathological conditions. We focused on the gCap population that functions as a progenitor cell for aCap in capillary homeostasis and repair[11]. We found that in lungs from aged BLM-challenged mice, gCap expressed an enriched pro-inflammatory and immune signature, as well as decreased expression levels of several transcription factors associated with vascular homeostasis, repair, and the anti-inflammatory response (*Klf2, Peg3, Klf10*)[45–47], indicating that the BLM response was exacerbated in gCap from aged mice. In addition, we also found an aging-related signature in gCap from healthy control lungs. Part of this signature (21 genes), corresponding mostly to genes associated with angiogenesis, migration, and sprouting, was correlated with the BLM-induced signature, suggesting a pre-inflammatory and injury-activated cell state in old gCap.

During the revision of this manuscript, a study by Raslan et al. also investigated the BLM-induced lung injury response in young and aged mice at single-cell resolution[60]. Interestingly, they also isolated a sub-population of activated gCap EC marked by the expression of *Ntrk2/TrkB* (Tropomyosin Receptor Kinase B) that appeared in BLM-injured lungs. These cells exhibited a similar gene expression signature to that presented here, including the upregulation of *Lrg1*, associated with activation of HIF1 and YAP/TAZ signaling. Their data, derived in part from scRNA-seq combined with lineage tracing of Aplnrpos cells, indicated that this activation resolved in young mouse lungs but persisted at day 30 in aged animals, confirming that aging delays the dynamics of these endothelial populations, as we demonstrated using robust statistical frameworks. Combining these data with ours thus enables the following hypotheses to be proposed. Firstly, the transient emergence of Lrg1pos PCEC subpopulations during the early phase of injury would be beneficial to the regeneration of the alveolar niche (young mice). On the other hand, delayed appearance and late accumulation of these pro-angiogenic EC would conversely be associated with non-functional angiogenesis and failure to resolve. Further

functional investigations aimed at depleting these *Lrg1*pos sub-populations in vivo in young and aged mice will be required in order to clarify these two contrasting hypotheses.

Our study has several limitations, in particular because of some technical biases during the single-cell and spatial transcriptomic workflow. First, our dissociation protocol, combined with the cell Hashing protocol[32], requiring additional cell labeling and washes has led to a depletion of fibroblasts and epithelial cells, which did not allow us to perform a complete cell state analysis for these cell types. Secondly, while the Visium spatial technology skips the dissociation step, its resolution did not allow to perform a single-cell characterization of the multiple cell lineages composing the alveolar niche, despite the use of a deconvolution tool. Moreover, although this approach was intended to highlight some spatial information explaining the resolution delay in aged mice, it will need to be supplemented by a larger number of replicates to improve these observations. Finally, the BLM mouse model may not accurately recapitulate the vascular alterations observed in IPF patients. The comparison of our data with a public dataset from the IPF cell atlas[36] indicated that the aCap Lrg1pos and gCap Lrg1pos populations were not detected in IPF samples, questioning the relevance of this mouse model. The *LRG1* transcript was also expressed at a low level in *COL15A1*pos systemic EC (SV EC and sCap) compared with their homologous mouse populations, confirming that LRG1 is not a relevant vascular marker of IPF in scRNA-seq data. This can be first explained by a significantly different basal expression pattern in human PCEC compared to that observed in murine data, suggesting variable physiological regulation between the two species. In addition, the difference between the acute injury BLM model resolution of fibrosis and the persistent end-stage fibrosis found in human samples is another plausible explanation. The transient dynamics of Lrg1 induction in response to injury, returning to a basal state after resolution, support this notion.

Nevertheless, the comparative analysis of the mouse and human fibrotic lung PCEC populations has allowed to draw several conclusions. In particular, we found a large conserved pathological signature of more than 500 genes between the *Col15a1*pos sCap from fibrotic mouse and IPF lungs, confirming their close functional state as well as a common pro-angiogenic signature between PCEC from mouse fibrotic lungs and human IPF systemic EC, suggesting that the pulmonary endothelial remodeling observed after BLM-induced lung injury in mice mimics the recruitment of systemic EC occurring in IPF. The characterization of EC heterogeneity in IPF is still in its infancy. In particular, it is not known whether the ectopic presence of *COL15A1*pos sCap in the IPF-associated distal parenchyma, plays a role in the pathogenesis of IPF. In order to gain information on their potential function, we used our scRNA-seq dataset on young PCEC at day 14 to investigate the potential relationship between sCap and gCap using the RNA velocity approach[48]. Our results indicate that following lung injury, sCap are recruited and may function as progenitor cells that can initiate a differentiation process, transiting through Lrg1pos gCap before reaching the physiological gCap cell state. The same differentiation scheme was also found in the old mice but with a delayed kinetic (28 days). Overall, these data suggest that in the BLM acute injury reversible fibrosis model, sCap are recruited and can act as progenitor cells to reconstitute the gCap pool and regenerate the integrity of the alveolar endothelial capillaries while aging appears to delay this resolution process. It is therefore tempting to hypothesize that in the context of the human pathology, systemic EC might be deficient in this differentiation process and accumulate in the injured tissue. Interestingly, a recent article, using an ex vivo human precision-cut lung slice (hPCLS) model of lung fibrogenesis, also described a transcriptomic proximity between alveolar PCEC and a systemic venous EC state[61]. In this case, the authors presented a differentiation

trajectory in the opposite direction, proposing that PCEC in injured alveoli give rise to a $VWA1^{pos}/PLVAP^{pos}$ ectopic EC state with transcriptional similarities to the systemic vasculature. Additional analyses are thus needed to better characterize this ectopic EC state versus the $COL15A1^{pos}$ systemic EC and distinguish between these two hypotheses.

Altogether, our findings shed light into the molecular mechanisms associated with alveolar endothelial capillaries resolution in a mouse model of reversible lung fibrosis and how aging influences specific PCEC to delay this resolution process. In particular, they raise a number of key hypotheses regarding the functional importance of these specific PCEC populations in lung repair, which may lead to therapeutic strategies for the treatment of IPF.

## Methods

### Reagents and antibodies
BLM was obtained from Sigma Aldrich (B1141000). The following antibodies were used in immunofluorescence experiments: Mouse Anti-Human CD31 (MA5-13188 Thermofisher), Rabbit Anti-Human COL15A1 (53667 Invitrogen), Purified Mouse Anti- αSMA–FITC (F3777 Sigma-Aldrich), donkey DyLight 594 Anti-Rabbit IgG (A32754 Life Technologies), donkey DyLight 647 Anti-Mouse IgG (A31571 Life Technologies). The following antibodies were used in western blot experiments: Rabbit anti-Human pSmad2 (18338, Cell signaling), Rabbit anti-Human Smad2 (5339, Cell signaling), Rabbit anti-Human pSmad1/5 (9516, Cell signaling), Rabbit anti-Human Smad5 (9517, Cell signaling), Rabbit anti-Human LRG1 (PA5-96832, Thermofisher), Rabbit anti-Human HSP90 (4877, Cell signaling) and Goat anti-rabbit HRP (P0448, Dako).

### Experimental design and animal treatment
All animal care and experimental protocols were conducted according to European, national and institutional regulations (Protocol numbers: 00236.03 and APAFIS#31298, IPMC approval E0615252; Protocol Number APAFIS#12540, University of Lille; approval 00236.03 CNRS). Personnel from the laboratory performed all experimental protocols under strict guidelines to ensure careful and consistent handling of the mice. The animals were maintained under a 12-h light-dark cycle with free access to food and water. Seven-week ("young") and 18-month ("old")-old C57BL/6 male mice (Charles River) were divided randomly into two groups: (A) saline-only (PBS, $n = 3$), or (B) BLM ($n = 3$). To induce fibrotic changes, 50 μl BLM (1 U/kg) or PBS was aerosolized in mouse lungs using a MicroSprayer Aerosolizer (Penn-Century, Inc.) as previously described[62]. Mice were sacrificed at designated time points (days 14, 28 and 60) after instillation.

### IPF lung tissue sections
Lung tissue sections were obtained from the UGMLC Giessen Biobank affiliated to the European IPF Registry as well as from the Nice Hospital-Integrated Biobank (transfer authorization from the French ministry of research n° AC-2021-4655).

### Histopathology
Lungs were fixed overnight with neutral buffered formalin and then embedded in paraffin. 5-micrometer-thick sections were mounted and stained with hematoxylin and eosin (HE) as well as Sirius Red to assess the degree of fibrosis. HE sections were scanned using Axio Scan Z1 scanner (Zeiss) at x10 magnification. Data are processed using the Zen blue imaging software (Zeiss). Hematoxylin and eosin zoomed in pictures were performed using Axiophot 2 light microscope.

### Hydroxyproline quantification of mouse lungs
Lung hydroxyproline content was assayed as previously described[62]. Briefly, lungs were disrupted in liquid nitrogen and 10 mg were homogenized in 100 μl of water. Samples were hydrolyzed for 3 hours at 120 °C in a temperature-controlled heating block after adding 100 μl

of HCl 12 M and then dried. Hydroxyproline quantification was finally performed in microplate as described by the manufacturer (Sigma).

### Visium™ spatial transcriptomics protocol
**Tissue preparation.** Tissue preparation and slides processing were performed according to the Visium Tissue Preparation Guide (CG000238 Rev A; CG000239 Rev A; CG000240 Rev A, 10× Genomics). Briefly, lungs of BLM-challenged mice (Young.D14/D28; Old.D14/D28, $n = 2$ for a total of 8 samples) were inflated with diluted Optimal Cutting Temperature (OCT), gently removed from mice and then embedded in OCT on dry ice and stored at -80 °C. OCT blocks were cryosectioned at 10 μm thickness on capture areas of Visium 10 Genomics slide. Before performing the complete protocol, Visium Spatial Tissue Optimization was performed according to manufacturer's procedure and using the Axioscan 7 scanner (Zeiss). Thirty minutes was selected as optimal permeabilization time. The experimental samples were fixed, stained with H&E, and imaged using Axioscan 7 scanner at 5x magnification.

**Generation of spatial transcriptomic libraries.** Library preparation was performed according to the Visium Spatial Gene Expression User Guide. Libraries were loaded and sequenced on a NextSeq2000 System (Illumina) as paired-end-dual-indexed, at a sequencing depth of approximately 25 M read-pairs per capture area.

**Analysis of spatial transcriptomics data.** The 8 space ranger pipeline outputs containing both the spot-level gene counts and the histological image of the tissue slice were read as seurat objects. After inspection of the space ranger (v1.3.1) reports, 2 tissue slices (Young.D14_#2 and Young.D28_#2) appeared unusable because of a very low level of detected molecules per spot (<800 detected genes and <1200 detected UMI compared to a mean of 2290 and 4688 in the other 6 slices). The remaining 6 samples (Young.D14, Old.D14.a, Old.D14.b, Young.D28, Old.D28.a, Old.D28.b) were jointly processed using the SpatialExperiment[63] (v1.10.0) and STDeconvolve[29] (v1.3.1) R packages. Firstly, the genes x spots count matrices of the 6 samples were stripped of mitochondrial, ribosomal, unannotated non-coding and hemoglobin subunit coding genes. The preprocess() function was run a first time on each sample to identify overdispersed genes, using the following arguments: nTopGenes = 5, removeAbove = 0.95, min.reads = 10, min.lib.size = 1, min.detected = 1, ODgenes = TRUE, od.genes.alpha = 0.05, gam.k = 5, nTopOD = 1000. We took the union of the 6 lists of overdispersed genes, representing 2482 unique genes. The preprocess() function was run a second time on the concatenated count matrix of those 2482 genes in all spots, specifying to discard spots containing less than 200 UMIs, and genes detected in less than 10 spots, resulting in a corpus of 2479 genes x 10813 spots count matrix and the associated spatial coordinates. The fitLDA() function was run on the count matrix in order to help selecting an optimal number K of predicted topics. Following the recommendations of the authors, we chose a value of K = 14, i.e., the value that produces the smallest perplexity without exceeding 5 topics with an average proportion of less than 5%. The topics proportions across spots (theta) and topics gene probabilities (beta) matrices from the optimal fitted LDA model were obtained with the getBetaTheta() function with a betaScale factor = 1000 to scale the predicted topics gene expression profiles. To annotate each of the 14 topics, the $\log_2FC$ of all expressed genes were calculated by dividing the expression in the topic versus the average expression in the rest of the topics. Genes with a $\log_2FC > 0.4$ were used to perform a canonical pathway enrichment in each topic using IPA (Ingenuity® Systems, www.ingenuity.com).

### Chromium™ single-cell transcriptomics protocol
**Lung dissociation and cell preparation.** Lung single cell suspensions were generated as previously described[64]. Briefly, after euthanasia,

lung tissue was perfused with sterile saline through the heart and were inflated through the trachea with dissociation cocktail containing dispase (50 caseinolytic U/ml), collagenase A (2 mg/ml), elastase (1 mg/ml), and DNase (30 µg/ml). Lungs were immediately removed, minced to small pieces ~1mm3), and transferred for mild enzymatic digestion for 20–30 min at 37 °C in 4 ml of the dissociation cocktail. Enzymatic activity was inhibited by adding 5 mL of PBS supplemented with 10% fetal calf serum (FCS). Single cells suspensions were passed through a 40-micron mesh, and then harvested by centrifugation at 300 $g$ for 5 min (4 °C). Reb blood cell lysis was performed using RBC lysis buffer (Thermo fisher) for 2 min at 4 °C and stopped using PBS 10% FCS. After another centrifugation at 300 g for 5 min (4 °C) cells were counted and critically assessed for single cell separation and overall cell viability using the Countess 3 FL (Fisher Scientific). Samples were then stained for multiplexing using cell hashing [3], using the Cell Hashing Total-Seq-ATM protocol (Biolegend) following the protocol provided by the supplier, using 6 distinct Hash Tag Oligonucleotides-conjugated mAbs (3 PBS and 3 BLM). Briefly $1.10^6$ cells were resuspended in 100 µl of PBS, 2% BSA, 0.01% Tween and incubated with 10 µl Fc Blocking reagent for 10 min 4 °C then stained with 0.5 µg of cell hashing antibody for 20 minutes at 4 °C. After washing with PBS, 2% BSA, 0.01% Tween samples were counted and merged at the same proportion, spun 5 minutes 350 x $g$ at 4 °C, resuspended in PBS supplemented with 0.04% of bovine serum albumin at final concentration of 500 cells/µl and pooled sample were immediately loaded on Chromium.

**Generation of single-cell libraries, 10X Genomics scRNA-seq and data processing.** Single-cell capture was performed using the 10X Genomics Chromium device (3′ V3). Single-cell libraries were sequenced on the Illumina NextSeq 500. Alignment of reads from the single cell RNA-seq libraries and unique molecular identifiers (UMIs) counting were performed with 10X Genomics Cell Ranger tool (v3.0.2). Reads of oligonucleotides tags (HTOs) used for Cell Hashing were counted with CITE-seq-Count (v1.4.2). Counts matrices of total RNA and HTOs were thus obtained for the 6 sequencing run, respectively named YoungD14, YoungD28, YoungD60, OldD14, OldD28 and OldD60.

**Single cell data secondary analysis.** All downstream analyses were carried out with Seurat R package (v4.1.0).

**HTOs demultiplexing.** For each 6 sequencing run, RNA and HTOs counts matrices were integrated into a Seurat object. HTOs counts were demultiplexed with HTODemux function to assign mice-of-origin for each cell (3 PBS-treated and 3 BLM-treated). Only the cells identified as "Singlet" and passing quality control metrics (more than 200 detected genes, less than 0.25% of mitochondrial content) were kept for each Seurat object. The 6 Seurat objects containing the raw counts of filtered cells and their metadata (age, day of lung collection post-injection, treatment, mice-of-origin) were merged.

**Samples integration.** The computational integration of the different samples was performed using reciprocal PCA, as described in the dedicated vignette from Satija lab website (https://satijalab.org/seurat/articles/integration_rpca.html). Briefly, a "detailed samples" metadata column was created by aggregating age, day of lung collection post injection and treatment (e.g.,: Young.D14.BLM, Young.D14.PBS, etc). According to this "detailed samples" metadata, cells were split into 12 seurat objects. For each of 12 datasets, raw counts were normalized using SCTransform with Gamma-Poisson generalized linear model to estimate regression parameters. 3000 features were selected for being repeatedly variable across the 12 datasets. The sctransform residuals corresponding to the selected 3000 features were computed if they were missing in a particular object. Principal Component Analysis (PCA) was run individually for each object on the sctransform

normalized counts of the 3000 selected features. A set of common anchors was found among the selected features, in dimensionally reduced data by reciprocal PCA, using FindIntegrationAnchors function with the following arguments: normalization.method = "SCT", dims = 1:50, k.anchor = 5. This set of anchors was used to create a combined object containing integrated data using 50 dimensions for the anchor weighting procedure.

**Clustering and annotation.** First, PCA was run on the integrated data. UMAP and kNN clustering were computed on the first 80 computed principal components (PCs). Clusters corresponding to aggregates of cells sharing a low-UMIs content were discarded. The entire integration and clustering processes were rerun on the cleared dataset, following the exact steps described above. The finally obtained clusters were annotated on the basis of the expression of their specific markers listed in the output of the FindAllMarkers function. Cell type–specific marker genes were established using the Wilcoxon rank sum test with p-values adjusted for multiple comparisons using the Bonferroni method. Adjusted p-values < 0.05 were considered significant.

**Differential expression analyses between fibrotic and PBS conditions in all cell populations.** Differential expression analyses between cells from BLM- and PBS-treated mice were carried out with DESeq2 (v1.30.1). First, all counts of each cell population from the same mice were aggregated to create a pseudobulk sample. For the PBS condition, we thus systematically obtain 18 pseudobulks samples corresponding to the 3 young and 3 old PBS-treated lungs collected at day 14, at day 28, and at day 60. For the fibrotic condition, we excluded the samples from -treated lungs collected at day 60, which correspond to an almost complete fibrosis resolution. We thus obtained 12 pseudobulks corresponding to the 3 young and 3 old BLM-treated lungs collected at day 14 and at day 28. For the alveolar macrophages pseudobulks, all counts from the 3 subpopulations (AM1, AM2, AM3) were aggregated. Genes considered as differentially expressed were selected according to an adjusted p-value threshold of 0.05, obtained with the Wald test and the Benjamini-Hochberg method for multiple tests correction implemented in DESeq2.

**Detailed analysis of endothelial cells (EC)**
**Identification of EC subpopulations.** The clusters corresponding to general capillary cells (gCap) and aerocytes (aCap) were extracted and subclustered using the 80 PCs already computed from the integrated assay of the complete dataset. Clusters corresponding to Lrg1$^{pos}$ aCap, Lrg1$^{pos}$ gCap and sCap subpopulations were isolated and added to the annotation. Following the same procedure, the cluster of venous EC was split into SV EC and PV EC. UMAP was rerun on the subset of cells corresponding to gCap, aCap, proliferating EC, Lrg1$^{pos}$ aCap, Lrg1$^{pos}$ gCap, sCap, SV EC, PV EC, and Arterial EC, using the initial 80 PCs. The raw counts were finally log-normalized with NormalizeData for all data exploration and visualization.

**Relative subpopulations frequencies.** All PBS-treated mice of the same age and sacrificed at D14, D28, or D60 were considered as replicates ($n = 9$), while BLM-treated mice were separated according to age and time of sacrifice ($n = 3$ for each time point). The relative subpopulations frequencies at the 4 time points (PBS, D14, D28, D60) were obtained by calculating the ratio between the number of cells in each subpopulation and the total number of cells in each mouse replicate, either mixing or distinguishing young and old mice. Differential abundance analyses were carried out using edgeR (v3.32.1) to model overdispersed numbers of cells per group with the NB GLM method and then to test for differences in abundance between sample groups with the glmQLFTest function, which performs empirical Bayes quasi-likelihood F-tests and FDR controlling.

**Differential expression analyses between fibrotic and PBS conditions in EC subpopulations.** Differential expression analyses between fibrotic and PBS conditions in EC subpopulations were performed following the pseudobulk approach described above. To build gCap and aCap pseudobulks, all counts of gCap plus Lrg1$^{pos}$ gCap or aCap plus Lrg1$^{pos}$ aCap from the same mice were aggregated. For sCap and SV EC, which were almost absent in physiological condition, their PBS pseudobulks were respectively built by aggregating the counts of gCap and PV EC from each PBS-treated mice. Mice from BLM-treated lungs collected at day 60 were also considered for the differential expression analysis for these two particular populations. Genes considered as differentially expressed were selected according to an adjusted p-value threshold of 0.05, obtained with the Wald test and the Benjamini-Hochberg method for multiple tests correction implemented in DESeq2.

**BLM-induced fibrosis score.** To compute a BLM-induced fibrosis score in sCap, gCap, aCap, SV EC, and PV EC, only the most significant upregulated genes from the differential expression analyses described above were selected according to adjusted p-value (<0.05), log2FoldChange (>0.5), and base mean expression (>10 for gCap, >5 for aCap, sCap, and PV EC, >1 for SV EC). Lists of selected genes were used to compute a score at the single-cell level using the AddModuleScore function. Scores were finally compared between the 4 time points (PBS, D14, D28, D60) and between young or old animals, using the Wilcoxon rank sum test implemented in the FindMarker function.

**Comparison between old vs young gCap in fibrotic and physiological conditions.** To compare the transcriptomic profiles of gCap from old and young mice in fibrotic condition, counts from BLM-treated gCap and Lrg1$^{pos}$gCap were aggregated within the same mouse for day 14 and day 28 time points, in order to obtain 6 young and 6 old pseudobulks samples. The same process has been applied to build pseudobulk samples of physiological gCap but including day 60 samples, resulting in 9 pseudobulks for either the young of old groups. Differential analysis between old and young pseudobulks samples was carried out with DESeq2. DEGs were selected according to adjusted $p$-value (<0.05), absolute log2FoldChange (>0.5) and base mean expression (>5).

**RNA velocity analyses.** The RNA velocity analyses were carried out following the kallisto | bus (https://bustools.github.io/BUS_notebooks_R/velocity.html) and scVelo workflows (https://scvelo.readthedocs.io/).

**Generation of spliced and unspliced matrices.** Intronic sequences of the mouse genome assembly GRCm38 were first identified using the get_velocity_files function of the BUSpaRse R package (v1.5.2) and the gencode annotation (v.M25). In order to allow the pseudoalignment of reads on both cDNA and identified intronic sequences, indexes for both cDNA and intronic sequences were built using kallisto (v0.46.2). The spliced and unpliced matrices were then generated for each sequencing run using the wrapper combining kallisto and bustools (v0.39.2) functions, typically with the following command line: kb count -i mm_cDNA_introns_index.idx -g tr2g.tsv -x 10xv3 -o kb \

```
    -c1 cDNA_tx_to_capture.txt  -c2 introns_tx_to_capture.txt  --work-
flow lamanno \
    sample_x_L001_R1_001.fastq.gz sample_x_L001_R2_001.fastq.gz \
    sample_x_L002_R1_001.fastq.gz sample_x_L002_R2_001.fastq.gz \
    sample_x_L003_R1_001.fastq.gz sample_x_L003_R2_001.fastq.gz \
    sample_x_L004_R1_001.fastq.gz sample_x_L004_R2_001.fastq.gz.
```

**Preprocessing.** Generated spliced and unspliced matrices of each sequencing run were then preprocessed to filter out empty droplets, to remove undetected genes and to replace ensembl IDs by gene symbols. Spliced and unspliced matrices corresponding to barcodes of a subset of cells of interest, for example, sCap EC, Lrg1$^{pos}$ gCap and gCap from BLM

and PBS-treated mice at day 14, were integrated in a Seurat object. The spliced raw counts were normalized using SCTransform from Seurat R package, and PCA was run on these normalized data, as well as UMAP with the first 30 PCs. The Seurat object was then converted to.h5ad file, keeping only the two raw spliced and unspliced assays.

**RNA velocity using scVelo.** The.h5ad file was read to an AnnData object using scVelo python package (v0.2.4). Again, data were preprocessed: the counts of the 3000 most variable genes with at list 20 reads were log-normalized using the filter_and_normalize function. PCA and nearest neighbors were computed before the estimation of velocity. The full transcriptional dynamics of splicing kinetics were resolved using the generalized dynamical model and represented as latent time in each cell. To visualize cells in 2 dimensions, diffusion maps were computed using the scanpy (v1.9.1) implementation with default parameters.

**NicheNet analysis.** We used the nichenet R package (v1.0.0)[40] to infer ligand-receptor interaction candidates able to induce the expression of pro-angiogenic genes in sCap, Lrg1$^{pos}$ gCap, and Lrg1$^{pos}$ aCap cells. For each of these 3 subpopulations, alternatively considered as receiver cells, we computed the activities of ligand sent by all the detected populations of the whole dataset. As gene sets, we alternatively used the pro-angiogenic genes expressed in at least one third of cells with an average expression above 1 in the particular subpopulation. For each ligand activities computed for each subpopulation, we selected the top 14 most likely to induce the geneset of interest by ranking them according to Pearson's correlation coefficient. The union of the 3 lists of prioritized ligands resulted in the 17 ligands used for the visualization. The selected receptors of those 17 ligands were only those considered as "bona fide", i.e., those whose interaction with their ligand is not only predicted in silico. The only exception is for Apoe, which did not have a bona fide receptor according to the NicheNet model, for which its best predicted receptor Scarb1 was added. The circos plot linking prioritized ligands and their receptors was realized following the steps described in the dedicated nichenet's vignette (https://github.com/saeyslab/nichenetr/blob/master/vignettes/circos.md).

**Ingenuity Pathway Analysis (IPA).** We used the core analysis of the IPA tool (Ingenuity® Systems, www.ingenuity.com) to investigate the enriched pathways and functions associated to the following 3 kind of DEGs lists used as inputs: (1) DEGs obtained by comparing fibrotic vs physiological cells in all EC subpopulations (see dedicated section above); (2) DEGs obtained by comparing old vs young gCap in fibrotic and physiological conditions (see dedicated section above); (3) DEGs obtained by comparing the pool of sCap, Lrg1+ gCap and gCap from young D14 mice positioned at the beginning of the differentiation trajectory inferred by RNA velocity (latent time <0.5), with the pool of cells positioned at the end of the trajectory (latent time >0.5).

**RNA-FISH**
RNAscope probes against mmu-Lrg1 (ID: 423381), mmu-Aplnr (ID: 436171),mmu-Ednrb (ID: 473801), mmu-Col15a1 (1092391), mmu-Hif1a1 (313821), mmu-Cd74 (437501), mmu-Sox17 (433151), mmu-Vegfa (405131) were prepared and obtained from Advance Cell Diagnostics. FISH assays were performed on FFPE lung sections, as previously described[56] using Multiplex Fluorescent Reagent Kit V2 (Advanced cell Diagnostics) and TSA Plus Cyanine 3 and Cyanine 5 (Perkin Elmer) as fluorophores according to manufacturer's recommendations. Acquisition was performed using an inverted confocal microscope LSM 710 (Zeiss), and a 40x oil immersion lens.

**Immunohistochemistry and Immunofluorescence Microscopy**
Five-μm paraffin-embedded sections were deparaffinized/rehydrated and H&E stained as previously described[62]. After washing, the sections were antigen-retrieved using citrate buffer (pH 6.0; Sigma-Aldrich).

Autofluorescence was quenched for 20-30 minutes in NH4Cl buffer (0.1 M) and passed in Photobleacher for 1h30. Slides were incubated with the primary antibodies (anti-CD31 (1:50), anti-COL15A1 (1:100), Anti-SMA–FITC (1:500)) diluted in a blocking solution BSA (1%) overnight at 4 °C. After washing in PBS-T (0.05%), slides were incubated for 1h with secondary antibodies (1:400) at room temperature. Nuclei were counterstained with DAPI (1:5000, D1306 Life Technologies). Slides were mounted using Immu-Mount (9990414 Fisher Scientific). Representative regions of stained slices were digitalized on an epifluorescent Zeiss microscope and analyzed using the software FIJI.

### Cell culture and LRG1 transduction
Human lungs primary microvascular endothelial cells (HMVEC-L) were obtained from Lonza (CC-2527, Basel, Switzerland) and HMEC1 (CRL-3243) were a gift from Christian Dani (Université Côte d'Azur, CNRS, INSERM, iBV, Nice, France). Cells were grown in EBM-2 Endothelial Cell Growth Medium containing SingleQuots kit EGM$^{TM}$-2 in an incubator (20% $O_2$, 5%$CO_2$) at 37 °C. Cells were transduced by a lentivirus expression vector containing the Ensembl sequence of LRG1 (VectorBuilder (pLV[Exp]-Puro-EF1A > hLRG1[NM_052972.3]) and were selected for resistance to puromycin (1 μg/mL). Cells were used directly after transduction.

### Western blot
Cells were lysed with RIPA cell lysis buffer (Thermofisher) containing phosphatase and protease inhibitors (Invitrogen). Proteins were separated on commercial gradient (4-12%) acrylamide gels and transferred onto PVDF membranes. Proteins were detected by overnight incubation at 4 °C with the indicated primary antibodies followed by 1h incubation at room temperature with horseradish peroxidase-coupled secondary antibody and enhanced chemiluminescent substrate (PerkinElmer, Waltham, USA) using a LasImager (GE HealthCare, Chicago, Illinois, USA). Blots were run on parallel gels when proteins of similar molecular weight were analyzed. The western blot quantification was performed with the help of the ImageJ software.

### Taqman Real-time PCR
Total HMEC1 RNA was isolated using trizol (Thermo Fisher Scientific) according to the manufacturer's instructions. RNAs (1 μg) were reverse-transcribed with High-Capacity cDNA Reverse Transcription Kit (Thermo Fisher Scientific), and PCR was performed using Taqman Real-time PCR technology (Applied Biosciences). All reactions were done using a Real-Time PCR system (Thermo Fisher Scientific), and expression levels were calculated using comparative CT method (2 DDCT). Gene expression levels were normalized to the RPLP0 level. The following probes from Taqman (Applied Bioscience) were used: Hs-RPLP0 (99999902-m1), Hs-VEGFA (00900055-m1), and Hs-SMAD2 (00183425-m1).

### 3D-migration assay (Spheroid sprouting assay)
Briefly, HMVEC-L were seeded in 96-well Cellstar U-bottom plates in suspension culture in EBM-2 complete medium. A day after spheroid formation, spheroids were individually embedded in interstitial type I collagen gel (2 μg/μL, Corning) on 35 mm glass bottom dishes (ibidi GmbH, Gräfelfing, Germany). Collagen gel was polymerized at 20 °C for 30 min and at 37 °C for 30 additional min. EBM-2 complete medium with or without TGF-β (10 ng/mL) was added on the spheroid for 24 h. Spheroid were fixed with PFA 4% durant 30 min at room temperature, and actin was labeled with Phalloïdine-594 (93042, 1/500, Sigma-Aldricht, Germany). Spheroids were mounted in 80% glycerol with DAPI (1:5000, D1306 Life Technologies) for nuclei staining. Z-stack images of spheroids were captured using a fluorescent confocal microscope (LSM780; Zeiss, Oberkochen, Germany; 10X magnification). Image processing spheroids measurements (number of migrating cells, maximum distance to spheroid surface, and spheroid volume) for each spheroid were performed by ImageJ software with the 3D Objects Counter plugin (Threshold: 20).

### Re-analysis of public scRNA-seq datasets
Human lung fibrosis[36] and mouse lung public datasets[37] were re-analyzed using reciprocal PCA, in order to integrate samples from different patients or different mice. To reduce the complexity of the human lung fibrosis dataset, we only kept cells coming from either IPF-diagnosed or normal samples, and we randomly sampled 10,000 immune cells and 5000 epithelial cells to reduce the dataset dimensionality. Adding the total number of endothelial and mesenchymal cells, we obtained a dataset of 25,816 cells. Differential expression and differential abundance analyses were carried out on this dataset in the same way as for our BLM dataset as described above. To be consistent with the analyses run on the mouse model, control pseudobulks of sCap and SV EC were respectively built by aggregating the counts of physiological gCap and PV EC from each donor.

### Statistical analysis
Statistical analyses were performed using GraphPad Prism and R Studio (v.4.1.2). Boxplots are represented with the median in the center, the whiskers correspond to the interquartile ranges, and the bounds correspond to the minimum and maximum values. Histogram and scatter plot are given as mean ± SD or ±SEM. Two-tailed Mann–Whitney test or multiple paired t test were used for single comparisons; Two-way ANOVA test followed by a multiple comparisons test with Holm-Šídák or Tukey correction was used for multiple comparisons. $P$-value < 0.05 was considered statistically significant.

For DESeq2-based differential expression analysis, $p$-values were obtained with the Wald test and adjusted following Benjamini-Hochberg correction.

For edgeR-based differential abundance analysis, $p$-values were obtained with the empirical Bayes quasi-likelihood F-test and adjusted following Benjamini-Hochberg correction.

For Seurat-based differential expression analysis, $p$-values were obtained with the Wilcoxon rank test and adjusted following Bonferroni correction.

Enrichment p-values obtained with IPA (Ingenuity® Systems, www.ingenuity.com) are calculated by right-tailed Fisher's Exact Test and adjusted following Benjamini-Hochberg correction.

### Reporting summary
Further information on research design is available in the Nature Portfolio Reporting Summary linked to this article.

## Data availability
The scRNA-seq and spatial transcriptomic data sets have been deposited in the Gene Expression Omnibus (GEO) database under accession record SuperSerie GSE234199 "Lung injury shifts pulmonary capillary endothelial cells towards regeneration-associated Lrg1+ sub-populations with delayed dynamics in aged mice". This SuperSerie contains 2 distinct datasets under the following accession codes: GSE234197: Gene expression profile at single cell level of whole lungs of mice collected from BLM or PBS-treated young and old mice at day 14, day 28, and day 60 post-injection. GSE234198: "Spatial RNA sequencing of BLM-treated young and old mice lungs at day 14 and day 28 post-injection". The two publically available datasets used in this study are available at GEO under the following accession codes: GSE135893 (Habermann et al.). GSE141259 (Strunz et al.). All the other data generated in this study are provided in the Supplementary information/Source Data file. Source data are provided with this paper.

## Code availability
The scripts used for the analysis of scRNA-seq and spatial transcriptomic data are available on github: https://github.com/

marintruchi/Aging_affects_reprogramming_of_PCEC. The corresponding archived repository can be downloaded and cited via the following https://doi.org/10.5281/zenodo.15706476, 2025.

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

## Acknowledgements

The authors thank the technical support of the UCA GenomiX, the CoBioDA bioinformatics hub of the IPMC, and the microscopy facility from the IPMC, part of the « Microscopie Imagerie Cytométrie Azur» GIS IBiSA labeled platform. We also thank the staffs from the Nice Hospital-Integrated Biobank (BB-0033-00025) and the Giessen PNEUMObank, as well as from the animal care facilities institutions at Sophia Antipolis (IPMC Animal Care Facility) and Lille (High Technology Animal Care Facility, University of Lille 2). We are also grateful to A. Monteil and C. Lemmers from the Vectorology facility, PVM, Biocampus Montpellier, CNRS UMS3426. We acknowledge the support from the Centre National de la Recherche Scientifique (CNRS), Institut National de la Santé et de la Recherche Médicale (Inserm), Université Côte d'Azur, Université de Lille, the French Government, through the UCA J.E.D.I. Investments in the Future project managed by the National Research Agency (ANR) with the reference number ANR-15-IDEX-01 (to BM and MGI), the "Investments for the Future" LABEX program reference # ANR-11-LABX-0028-0, the ANR programs ANR-PRCI-18-CE92-0009-01 "FIBROMIR", ANR-22-CE17-0046-01 "MIR-ASO" (to BM and NP), ANR-24-CE18-2311 "STARNASH" (to NP), National Infrastructure France Génomique [ANR-10-INBS-09-03, ANR-10-INBS-09-02] (to PB), IHU Respirera [ANR-23-IAHU-0007] (to PH), 3IA@coted'azur [ANR-19-P3IA-0002] (to PB); PPIA 4D-OMICS [21-ESRE-0052] (to PB) and the Programme Contrat Plan État Région (CPER) - CTRL (Centre Transdisciplinaire de Recherche sur la Longévité, projet FISSURE) (to CC) as well as Canceropôle PACA (to BM and PB) and Conseil Départemental des Alpes Maritimes (2019-390DGADSH). SB was supported by grants from the DFG (BE4443/18-1, BE4443/1-1, BE4443/4-1, BE4443/6-1, KFO309 284237345 P7 and SFB CRC1213 268555672 projects A02 and A04), UKGM, the Universities of Giessen and Marburg Lung Center (UGMLC) and DZL.

## Author contributions

B.M., N.P., G.V., and C.C. conceived and designed the study and supervised the entire work. M.T. and B.M. wrote the manuscript. G.S., M.G.I., H.C., and A.B. performed in vivo studies. G.S., M.G.I., H.C., J.F., N.B., C.S., and A.L. performed biochemical and cellular biology experiments. G.S., Cd.S., N.M., and M.J.A. performed the Visium spatial transcriptomic experiment. V.M. and C.G.R. generated scRNA-seq data and primary analysis. MT performed bioinformatic analysis of scRNA-seq and spatial transcriptomic data. K.L. co-supervised the computational analysis. G.S., M.G.I., R.L., and H.C. performed and analyzed immunostainings. M.T., G.S., N.R., G.V., C.C., S.B., N.P., and B.M. performed biological interpretation. V.H., P.H., A.G., C.H.M., and S.L. provided IPF-derived biological materials and contributed to the biological interpretation of data. R.R., M.P., N.R., P.B., and S.B. gave conceptual advices. O.P. and S.B. provided resources. All authors read and corrected the final manuscript.

## Competing interests

The authors declare no competing interests.

## Additional information

[1]Université Côte d'Azur, UMR CNRS 7275 Inserm U1323, IPMC, Valbonne, France. [2]IHU RespirERA, Université Côte d'Azur, Nice, France. [3]Univ. Lille, CNRS, Inserm, CHU Lille, Institut Pasteur de Lille, UMR9020 CNRS - U1277 Inserm - CANTHER, Lille, France. [4]Cardio-Pulmonary Institute (CPI) and Department of Pulmonary and Critical Care Medicine and Infectious Diseases, Universities of Giessen and Marburg Lung Center (UGMLC), member of the German Center for Lung Research (DZL), Justus-Liebig-University (JLU), Giessen, Germany. [5]Institute for Lung Health (ILH), Giessen, Germany. [6]Laboratory of Clinical and Experimental Pathology and Biobank Côte d'Azur BB-0033-00025, Centre Hospitalier Universitaire de Nice, Nice, France. [7]Université Côte d'Azur, UMR 7413 CNRS Inserm U1081, IRCAN, Nice, France. [8]Department of Pulmonary Medicine and Thoracic Oncology, Centre Hospitalier Universitaire de Nice, Nice, France. [9]European IPF Registry and Biobank, Giessen, Germany. [10]Lung Clinic, Agaplesion Evangelisches Krankenhaus Mittelhessen, Giessen, Germany. [11]BioSanté unit U1292, Grenoble Alpes University, INSERM, CEA, Grenoble, France. [12]3IA Côte d'Azur, Université Côte d'Azur, Sophia Antipolis, France. [13]These authors contributed equally: Marin Truchi, Marine Gautier-Isola, Grégoire Savary, Nicolas Pottier, Bernard Mari. ✉e-mail: mari@unice.fr

