## [Transparent Peer Review file · Nature Communications]

Aging affects reprogramming of murine pulmonary capillary endothelial cells after lung injury

Corresponding Author: Dr Bernard Mari

Version 0:

Reviewer comments:

Reviewer #1

(Remarks to the Author)

The roles of the vascular compartments and their changes in human pulmonary fibrosis are critical in understanding the pathogenesis of pulmonary fibrosis. Several recent publications that described a decline in lung capillary cells and heterogeneity in pulmonary endothelial cells in lung injury and fibrosis (PMID: 32832599/ 32832598/34030460/37137915/38529320/36712020). The current manuscript by Truchi and colleagues generated a single cell RNA-seq (scRNA-seq) database on young vs old mouse lungs at fibrotic and resolution stages, and annotated the endothelial cell subpopulations with several subpopulations positive for Lrg1 (Lrg1+ gCap and Lrg1+ aCap). The frequencies of the endothelial cell subpopulations are dynamically changed at different fibrosis and resolution stages, and those of the Lrg1+ endothelial cells, although overall low in percentages, peaked on D14 in young and on D28 in old mouse lungs. However, this manuscript lacks novelty compared to the aforementioned publications on pulmonary endothelial cells in fibrotic lungs, and is overly descriptive with a lack of rigorous verifications on several key points of interest, thus it significantly dampened the reviewer's enthusiasm.

1. The heterogeneities of pulmonary cells, especially pulmonary endothelial cells, in fibrotic lungs have been well described, which significantly diminished the novelty (and necessity) of Figure 1, 2, and S1. Figure 1 described a delay of lung fibrosis resolution and Figure 2 described a cellular heterogeneity in aged mice lungs.
2. One of the major concerns is the less rigorous presentation of the data: One of the primary obstacles in single-cell analysis is cell contamination, particularly during the process of cluster subsetting. The authors presented most of the gene expression by either heatmaps or violin plots. Some Feature plots are necessary when cell populations/subpopulations are defined and when transcription of some genes are shown, e.g., Lrg1 in endothelial cell subpopulations. Another reason why cell purity needs to be confirmed stands out is because that top DE genes of the old gCap cells are pro-inflammatory gene, such as S100a8, S100a9, Ms4a1, Ly6d, Cd74 or Cd52. Almost all these genes are specific to the myeloid cell lineages. It is critical to eliminate the effect of cell contamination in this analysis, as well as in the following analysis regarding function prediction of specific subpopulation.
3. Gene expression and some other endothelial cell phenotypes related to aging and fibrosis need better verifications. Alveolar niche regeneration and pro-inflammatory phenotype by Lrg1+ PCEC subpopulations are highly descriptive via the changes in some representative gene transcription and bioinformatic pathway/interaction analysis. Some protein-level-based verifications (on the pro-inflammatory/Hypoxia-regulated/pro-angiogenic genes) or functional studies (on the alveolar niche regeneration) are highly suggested.
4. The aCap and gCap in fibrotic lungs have been discussed in several previous publications, but the concept of the Lrg1+ PCECs is novel. However, some critical questions this study should have addressed: what signal is driving the emergence of this specific subpopulation and what cell types do they give rise to in the lungs of resolution stage. Without additional functional and lineage tracing studies, its role in lung fibrosis (and in aging) is largely speculative.
5. Another concern regarding the Lrg1+ PCECs is their frequency in each subpopulation. Actually, the Lrg1+ PCEC are highest in PV EC at all time points (Figure S2B-C), are there cells distinct from the main PV EC population? It seems that the percentages of Lrg1+ gCap and Lrg1+ aCap are extremely low in the scRNA-seq analysis (Figure 3D and 4A), however in the in-situ hybridization staining (Figure 4E), their percentages are much higher in both young and aged lungs. It is necessary to verify the dynamic change of Lrg1 in lungs of different timepoints in others' dataset (Figure S2F), but it would be more helpful to see if the distinct Lrg1+ gCap or Lrg1+ aCap subpopulations can be identified in this dataset.
6. Figure 5 presented the pathological changes in endothelial cells in IPF and mouse fibrotic lungs, which is not directly related to "aging". The data in this figure did not show any advances compared to previous publications (e.g., PMID:

34030460). The relationship to Lrg1 is not clear. The conclusion with COL15A1 is speculative.

7. One of the main disappointments is the failure to identify LRG1+ EC in IPF, which significantly diminishes the importance of LRG1+ EC in human disease.

8. Figure 6 needs a more rigorous and comprehensive analysis. It includes only two cell (sub)types. An unbiased analysis including ALL endothelial cells and all timepoints is needed. It seems that the Lrg1+ gCap are derived from SV EC and differentiate into gCap in fibrosis models of both young and old mice, when only two cell types were included.

9. There are many over-speculative statements, for example, "in the human pathology, SV EC are likely not able to differentiate and accumulate, contributing to persistent fibrosis". There is no experimental evidence presented.

Minor

10. Cell number of the scRNA-seq: 4,4541 (line 133) or 4,5311 (Figure 2)?

11. Lymphatic ECs were just mentioned in a few panels in supplemental figures, but did not show up in any figures.

12. Please define the "bleomycin-induced signature score" in Figure 4B.

13. Many quantification analyses are in lack of statistics (Figure 2D, 4A-B, 5B, and many in sup-Figures).

(Remarks on code availability)

Reviewer #2

(Remarks to the Author)

This is an interesting manuscript that describes a potential role for aberrant differentiation of pulmonary capillary endothelial cells (PCECs) in the impaired (and delayed) regeneration of the lung following fibrotic lung injury. Specifically, the expression of Lrg1+ gCap/aCap within the alveolar niche is delayed in aged mice, suggesting that this may impair the formation of a new mature capillary network. They also describe an accumulation of COL15A1+ systemic venous ECs (SV-ECs) in aged mice and hypothesize that these cells (also found in IPF) may contribute to delayed regeneration and persistent fibrosis.

The main concern with the current manuscript is its descriptive nature without actual experimental evidence for how these EC populations either support or impair alveolar regeneration. If in-vivo studies with gain and/or loss of function studies are "beyond the scope", then at the very least ex-vivo (surrogate) studies with angiogenesis assays could provide more insights into the specific role of these EC subtypes and their reprogramming during the repair/regenerative process.

(Remarks on code availability)

Reviewer #3

(Remarks to the Author)

Comments for "Aging affects reprogramming of murine pulmonary capillary endothelial cells after lung injury"

The authors utilize scRNA-Seq, spatial transcriptomics and in situ hybridizations to analyse gene expression changes upon bleomycin induced injury in young and old mice. A strength of eth work is that they collect data during the establishment and resolution of lung fibrosis (days 14, 28 and 60) of lung fibrosis and its resolution. Previous studies on lung fibrosis using similar methods have described changes in alveolar macrophages and the appearance of Lrg1pos endothelial cells. The authors have additionally utilized bioinformatic tools to predict cell communications that are affected in the injured and/or old samples may drive the gene expression changes. They also compared their mouse data to previous data from lung fibrosis patients providing suggestions about species related differences that result to fibrosis resolution only in the mouse but not human. In general, the study is a systematic effort to analyse the age-related differences in the lung response to bleomycin induced injury but they mainly focus on endothelium. It would have been interesting to the reader to see an initial analysis of all detected cells states since a major strength of the study is the tight sampling of time points. The inclusion of such an analysis and the publication of all data before focusing on the endothelium would increase the general interest on the paper and may provide a better basis for the cell-interaction analyses.

Main Comments:

1) Even if the results are presented clearly in the figures, the analysis methods are not explained enough. For example, in lines 119-120 the authors state that they used "an unsupervised assignation of each spot." but they do not describe how they did it in the methods.

2) In Figure 1 and Suppl. Figure 1, the authors present histology panels from both PBS and Bleo treated lungs but they do not show any Visium results from PBS treated lungs. This is a drawback because it is unclear how they have determined the "normal" gene expression conditions in alveolar, airway, interstitial, vessel and SMC layer. They also do not describe any differences between PBS young and old mice.

3) In Figure 1D, the barplot shows a significantly increased percent of "low counts alveoli" in the old D28 lungs. Is that because of emphysematous or fibrotic/scarred areas or just stems from a technical issue? The normal, PBS treated sample would address this.

4) The authors used a mild tissue dissociation approach that might have resulted to incomplete cell release and introduction

of bias. This could explain the small number of epithelial cells compared to the abundant immune cells. They should at least comment on this limitation and its impact in the described cell-type frequencies, especially for the fibrotic lungs, which should be more difficult to dissociate. Cell type frequencies would be best addressed by the Visium experiment. The authors have not assigned cell types to their spatial reads, as it is commonly done in this type of experiments. What is the reason for this?

5) In the lines 140-141 the authors describe an expected shift in several populations following bleomycin. But they do not specify if that refers to changes in the frequencies of cells (lines 142-144) or gene -expression changes in a given cell-type or both (Fig. 2D).

6) In Figure 2 a better representation of the changes in cell-type frequencies in all conditions, that will also account for the different library sizes should be used. Also, it might be informative to examine if the differences in alveolar macrophages states upon bleomycin treatment have any specific spatial distribution. Are they found in severely damaged or healthy areas or at their borders?

7) The authors indicate that “transcriptional remodelling of gCap into Lrg1+ transitory subpopulations may affect their abilities to differentiate and replenish the pool of functional aCap” (lines 250-252). Importantly they have observed that this transition in both young and old lungs but only the old showed impaired healing. Also, this conclusion comes in contrast to their discussion point in lines 398-401 (“Lrg1-associated pro-angiogenic activity appears to be necessary to promote alveolar vascular repair...”). Thus, their findings might better reflect a compromised reparatory capacity of old cells.

8) In Figure 4, the authors present convincing results about the expression of Lrg1, Aplnr and Ednrb on tissue sections but it will be informative to examine the localization of these cells relatively to the injured areas and specify it (as in comment 6).

9) In Figure 4G, two of the significantly changed genes between the OLD vs YOUNG gCap in PBS are the “Scgb1a1” and “Sftpc”. These markers are expressed in airway and alveolar secretory cells and not vascular endothelium. Thus, some of these cells are likely doublets or mRNAs of these highly expressed epithelial genes were released from lysed/destroyed epithelial cells during tissue dissociation. This can happen but more stringent filtering should be applied regarding the expression levels of the processed genes to avoid technical noise.

10) In Figure 6, the authors performed pseudotime analysis to analyse the transition of systemic vascular endothelial cells to gCap using UMAP dimension reduction. It would be better to use Diffusion Maps, because it performs more efficiently for that type of analysis. It would also be helpful to include aCap cells as the end state, these are destroyed by bleomycin and are expected to become repaired after the injury.

11) In lines 139-140 and 361 the authors mentioned that they have identified a population of proliferating endothelial cells but they do not provide any further information about possible differences in the number and gene expression profiles of these cells between young and old mice. The only UMAP showing endothelial cells is in Figure 2 and there, the assigned proliferating cells are very few.

12) My final comment relates to the analysis approach of considering day60 post bleo administration as healthy (In Suppl. Fig. 1 A, it is obvious that both young and old lung tissue condition is improved but differs from PBS). Also, I am sceptical of the merging young and old day14 and 28 datasets of PBS and bleomycin- treated mice (lines 739-750) and sometimes including day60 (lines 783-784) because this approach might dilute differences that are temporally restricted to one timepoint and/or age. Differential expression between samples of the same timepoint and age should be the basis for identifying the induced or suppressed genes in response to bleomycin treatment and/or aging in all cell-types.

(Remarks on code availability)

Reviewer #4

(Remarks to the Author)

In the submitted manuscript, researchers tried to find potential role of endothelial cells in age related fibrosis formation and resolution context. For this purpose, they used multi-omics approach to verify their histology and in vivo results via Single Cell RNA sequencing and Spatial Transcriptomics. They first evaluated the changes induced by aging and fibrosis and figured out that the endothelial cells are the second most affected cell type after macrophages. They defined novel pulmonary capillary endothelial subclusters in their mouse model (Lrg1+ PCEC) that contributes to aging associated angiogenesis and alveolar niche regeneration. Furthermore, they discovered aging induced potential pro-fibrotic role of gCaps, which may influence the impairment of alveolar regeneration. Also, they showed that aging induces shift in the existence of Lrg1+gCap and Lrg1+aCap in young and aged mice model where they prove their contribution to remodelling of PCEC and potential role in fibrosis resolution. To identify the role of these subclusters, they performed RNA Velocity to indicate the transition process of Lrg1+gCap and potential progenitor systemic venous vessels (SV EC) towards gCaps. They were able to observed similar subcluster of Col15a1 SV EC in human single cell data that is upregulated in IPF patients, yet couldn't able to identify Lrg1+ subclusters.

Main problem:

1- Mouse data findings especially related with Lrg1+ PCEC subclusters do not exist in the human IPF single cell RNA sequencing data

Open questions:

1- Researchers performed spatial transcriptomics to validate their histological findings of „delayed fibrosis resolution“ in aged mice model. However even they have n=3 for histological findings in old and young mice model, they performed Visium spatial transcriptomics by using n=1 for young mice and n=2 for old mice. The differential gene expression is not valid to calculate p-adj method. Furthermore, the number of cells detected will be less from n=1 young group which gives misleading UMAP plot. Also in reanalysis of spatial data for Lrg1+ cells in Fig4.C, it is clear that old D28a and old D28b samples do not match with each other. In the method part they explained why they had to use n=1 for young mice, yet the

problems are still valid.

2- Researchers used PBS as negative control for histological analysis but no control group used in spatial transcriptomics.

3- In Figure1 description, damaged alveolar regions are characterized by low UMI content yet the heatmap in Supp.Fig1D shows that it is not specific to age related differences. Therefore, it would be better to include DEG Volcano plot that compares young vs old mice tissue in "low counts Alveoli" subset to support their aim in the first part of the result as "fibrosis resolution is delayed lungs from aged mice". Furthermore, it would be better to use cluster annotation age to cluster the rows in heatmap in Supp.Fig1D.

4- In line 123, regarding the comment above, the comparisons are not age related but damaged associated.

5- In Figure3, researchers labelled aCap cells that has higher Lrg1 expression as Lrg1+aCap. Is the difference between two subclusters in terms of Lrg1 expression is significant to assign them as Lrg1 overexpressed (Line 157) ? Difference in between gCap is more clear both in heatmap (Fig3.B) and Violin Plot (Fig.3C).

6- In between lines 159-161, researcher claimed that there is a subpopulation called Prolif.EC, which is visible in the heatmap Fig3B with Mki67 expression. The figure that supports the difference between day14 and day 28 following bleomycin treatment is Fig3D not C.

7- Since single cell RNA Seq results indicates the significant downregulation of certain genes (Supp.Fig3B-C) it would be a better approach to verify these findings in spatial transcriptomics.

8- The expression of Smad family members' difference in between Pbs vs Bleo model are not significant for Smad2 and Smad5 (Supp.Fig3C). In the discussion, it is claimed that Smad2 and Smad5 works controversially with the influence of Lrg1 and Tgfb1 signalling. Is there any functional experiment related with the activity of Smad Tgfb and Lrg1? Or is there any transcriptomic data shows that Age+Bleo related differences among these genes that supports the claim in line 400?

9- Legend is missing indicating colors of different Bleo treatment days (Fig.4E)

10- Due to the experimental single cell sequencing results of the researchers Lrg1 expression mostly depending on increase in age, how do they define no expression in IPF human data? (except lines 327-328) Did the researchers checked other available data sets? (ex. lpfcellatlas.com)

11- Heatmap indicating D28 old mice transcriptional dynamics is missing (Fig6) supports the claim in line 320

(Remarks on code availability)

Version 1:

Reviewer comments:

Reviewer #2

(Remarks to the Author)

The authors have not directly addressed my concerns regarding a lack of functional interrogation of potentially "pathogenic" EC subpopulations (difficulty with available antibodies for flow sorting). An attempt at showing functional significance of LRG1 in a HMEC1 human endothelial cell line is not very convincing (Supplemental Fig. 4).

(Remarks on code availability)

Reviewer #3

(Remarks to the Author)

The authors have addressed satisfactory most of my comments. I still believe that the discussion is too long and in places repetitive.

(Remarks on code availability)

Reviewer #5

(Remarks to the Author)

Major concerns of Reviewer#1 were related to:

1) Novelty of the study

Reviewer noted that other publications, including a recent study by Raslan et al. reporting similar findings, may impact the novelty of this research. In their rebuttal letter, Truchi et al. differentiated their study from Raslan et al. by highlighting the addition of time points and spatial transcriptomic analysis. Although the tools and experimental designs differ between the two studies, the findings remain for the most part incremental and descriptive, as reviewer#1 pointed out. However, considering the timing and growing interest in this pulmonary vascular field, I believe that generating independent findings from different groups is still valuable and contributes meaningfully to the field.

Nevertheless, to strengthen their manuscript, Truchi et al. should provide a more detailed comparison of their findings with those reported by Raslan et al., highlighting both the similarities and differences between the two studies. For example,

Raslan et al. reported a time course analysis in young animal using scRNA-seq and lineage tracing which provides deeper insights into capillary endothelial cells (ECs) changes resulting from bleomycin injury. In addition, Raslan et. al. showed that Ntrk2+ gCap ECs (this population has the Lrg1+ EC transcriptional signature) peaks during fibrosis (day 14) and declines during the resolution phase (day 30) in injured young, while persist in injured aged mice. This indicates that this EC accumulation in aged mice may contribute to fibrosis progression, a conclusion that differs from the one proposed by Truchi et.al. Clarifying these distinctions would provide a more accurate understanding of the impact of aging on lung capillary heterogeneity and fibrosis development. In addition, Truchi et al. reports spatial transcriptomic data, which could be leveraged to provide additional context and insights and better differentiate the contribution of their study with respect to previous ones.

2) Concerns related to the capillary nature of Lrg1+ ECs.

The study's conclusion regarding the specific population of lung endothelial cells (ECs) expressing Lrg1 remains unclear. The scRNA-seq data suggest that Lrg1 is broadly expressed across ECs from all pulmonary vascular beds, with notably high expression levels in venous ECs. Thus, it remains unclear whether Lrg1+ capillary ECs are uniquely susceptible to aging-related effects compared to ECs from other vascular beds.

Truchi et al. have revised their manuscript to include RNA-scope analysis showing the expression of Col15, a venous EC marker previously implicated in human lung fibrosis. Although the discovery of Col15a1-expressing ECs in mice following bleomycin injury is intriguing, its relevance to capillary Lrg1 remains unclear, as it is not clearly established whether Lrg1 is co-expressed in Col15 EC population. In addition, in human lungs, COL15a1+ ECs were primarily associated with bronchial venous-derived ECs, rather than capillary ECs, which highlights the need for further clarification on the relationship between COL15a1 and Lrg1 expression in mouse lung ECs.

Reviewer suggests using protein-level-based verification for several inflammatory, hypoxia-regulated, and pro-angiogenic genes, particularly regarding their functional roles in alveolar regeneration. The authors did not fully address this concern as RNAscope was utilized in the reported experiments (Fig. 2-4). It is unclear why this method was chosen over immunohistochemical analysis, especially since some proteins, such as HIF1A, are regulated primarily at the protein level.

3) Notable absence of Lrg1+ ECs in human lungs with pulmonary fibrosis.

Another issue raised by the reviewer#1 is the lack of clarity and depth in addressing on LRG1+ capillary EC absence in human lungs with pulmonary fibrosis. The authors linked their findings to COL15-expressing ECs previously reported in the literature. This may create ambiguity and undermines the focus of the study. The authors should provide a rationale for the absence of LRG1+ capillary ECs in human lungs and distinguish their findings from those previously reported implicating COL15 ECs in human lung fibrosis. One possible interpretation is that the discrepancy may be attributed to the end-stage nature of human diseased lungs, which differs significantly from the acute injury model used in mice. This emphasize that mouse models do not fully capture the disease pathogenesis, particularly in its advanced stage.

4) Lack of mechanism/s involving Lrg1 capillary ECs

Reviewer#1 suggested to provide a possible mechanism explaining the emergence of the Lrg1+ ECs during lung injury. Truchi et. al. did not fully address this point. While this may be experimentally challenging and time consuming, the authors should use bioinformatic tools to get insights into the molecular events leading to Lrg1+ capillary EC activation, based on gene signatures emerging during the onset of lung fibrosis.

Major concerns of Reviewer#4 were related to:

1) Absence of Lrg1+ ECs in human lungs.

The absence of Lrg1 ECs in human lungs remains a weakness of the study. Thus, the relevance of Lrg1 ECs in the context of human pulmonary disease remains inconclusive. The discrepancy between mouse and human data may be attributed to the end-stage nature of human diseased lungs, which differs from available mouse models. This aspect does not seem to be adequately discussed in the manuscript.

2) Lack of statistical power and absence of healthy lungs in the spatial transcriptomic study.

The authors have sufficiently addressed this concern.

3) In Figure1 description, damaged alveolar regions are characterized by low UMI content yet the heatmap in Supp.Fig1D shows that it is not specific to age related differences.

The authors have sufficiently addressed this concern.

4) Since single cell RNA Seq results indicates the significant downregulation of certain genes (Supp.Fig3B-C) it would be a

better approach to verify these findings in spatial transcriptomics

The authors were unable to address this point due to the low resolution of the spatial transcriptomic analysis.

5) The expression of Smad family members' difference in between Pbs vs Bleo model are not significant for Smad2 and Smad5 (Supp.Fig3C). In the discussion, it is claimed that Smad2 and Smad5 works controversially with the influence of Lrg1 and Tgfb1 signalling. Is there any functional experiment related with the activity of Smad Tgfb and Lrg1? Or is there any transcriptomic data shows that Age+Bleo related differences among these genes that supports the claim in line 400?

The authors have sufficiently addressed this weakness and provided additional data.

(Remarks on code availability)

Version 2:

Reviewer comments:

Reviewer #2

(Remarks to the Author)

The authors have adequately addressed my concerns.

(Remarks on code availability)

Reviewer #5

(Remarks to the Author)

The authors have addressed the reviewer's concerns. However, the discussion needs to be revised as follow:

In the discussion the author stated: "Their data derived in part from scRNA-seq combined with lineage tracing of Aplnr+ cells indicated that this activation resolves in young mice but persisted at day 30 in aged animals confirming that aging delays the dynamics of these endothelial populations, as we demonstrated using robust statistical frameworks. However, their conclusions, stating that this population is dysfunctional and accumulates with aging, differ from our".

It is unclear why the authors stated that " Their conclusions....differ from our". After carefully reading both manuscripts, this reviewer finds that both groups reached same conclusions about the impact of aging on the persistency of gCap EC activation following injury. Thus, the statement above should be revised to better reconcile the findings and provide a clearer understanding of the similarities

(Remarks on code availability)

Response to Reviewers

We are grateful to the reviewers for taking the time to evaluate our manuscript. We appreciate their constructive comments. We have responded (in blue) to each comment. We have also added supporting additional figures for Reviewers at the end of the document (**Figs. Rev.1-8**) as well as the complete list of references cited in comments and answers (referred to by PMID).

REVIEWER COMMENTS

Reviewer #1 (Remarks to the Author):

The roles of the vascular compartments and their changes in human pulmonary fibrosis are critical in understanding the pathogenesis of pulmonary fibrosis. Several recent publications that described a decline in lung capillary cells and heterogeneity in pulmonary endothelial cells in lung injury and fibrosis (PMID: 32832599/ 32832598/34030460/37137915/38529320/36712020). The current manuscript by Truchi and colleagues generated a single cell RNA-seq (scRNA-seq) database on young vs old mouse lungs at fibrotic and resolution stages, and annotated the endothelial cell subpopulations with several subpopulations positive for Lrg1 (Lrg1+ gCap and Lrg1+ aCap). The frequencies of the endothelial cell subpopulations are dynamically changed at different fibrosis and resolution stages, and those of the Lrg1+ endothelial cells, although overall low in percentages, peaked on D14 in young and on D28 in old mouse lungs. However, this manuscript lacks novelty compared to the aforementioned publications on pulmonary endothelial cells in fibrotic lungs, and is overly descriptive with a lack of rigorous verifications on several key points of interest, thus it significantly dampened the reviewer's enthusiasm.

We respectfully disagree with the reviewer regarding the lack of novelty of our study. While several scRNA-seq studies have been already performed in pulmonary fibrosis mouse models, no accepted study had previously characterized Pulmonary Capillary Endothelial Cells (PCEC) heterogeneity in this context, in particular with respect to fibrosis resolution and aging. The preprint from Raslan et al (PMID: 36712020), which has been recently accepted in "Nature Communications" (PMID: 38937456) after having received this first round of review, represents the only study that addresses this issue but using a rather different design with distinct results and conclusions, as mentioned in the discussion (lines 514-525). We are notably providing a temporal scRNA-seq dataset on young and aged mice at 3 different time points (with n= 3 for each condition), complemented by a spatial transcriptomic analysis, leading to the characterization of new injury associated states of PCEC as well as the emergence of a new subpopulation of systemic EC only characterized in IPF patients to date. We also believe that our findings also provide important new insights regarding the molecular and cellular mechanisms associated with the altered vascular response to injury in aged lungs.

1. The heterogeneities of pulmonary cells, especially pulmonary endothelial cells, in fibrotic lungs have been well described, which significantly diminished the novelty (and necessity) of Figure 1, 2, and S1. Figure 1 described a delay of lung fibrosis resolution and Figure 2 described a cellular heterogeneity in aged mice lungs.

To our knowledge, a direct longitudinal comparison of young and aged mice in a pulmonary fibrosis model using scRNA-seq had not been carried out at the time of submission. We have therefore retained the essential data showing experimental design and delayed lung resolution only in aged mice using separate but complementary approaches. As recommended, we have also extensively reorganized the manuscript and the figures. Accordingly, we have notably revised and improved our analysis of the visium dataset (**new Fig. 1D-F**), providing a more precise characterization of the cellular and molecular alterations occurring in lungs from BLM-treated aged *versus* young mice. We have also extensively modified **Fig.2**, focusing on the characterization of novel mouse PCEC populations and comparing them with the EC populations observed in IPF.

2. One of the major concerns is the less rigorous presentation of the data: One of the primary obstacles in single-cell analysis is cell contamination, particularly during the process of cluster subsetting. The authors presented most of the gene expression by either heatmaps or violin plots. Some Feature plots are necessary when cell populations/subpopulations are defined and when transcription of some genes are shown, e.g., *Lrg1* in endothelial cell subpopulations. Another reason why cell purity needs to be confirmed stands out is because that top DE genes of the old gCap cells are pro-inflammatory gene, such as *S100a8*, *S100a9*, *Ms4a1*, *Ly6d*, *Cd74* or *Cd52*. Almost all these genes are specific to the myeloid cell lineages. It is critical to eliminate the effect of cell contamination in this analysis, as well as in the following analysis regarding function prediction of specific subpopulation.

There is no perfect representation of gene expression at the single-cell level. We have chosen to represent individual gene expression using Violin plots, as we believe they provide a better quantitative overview of the number distribution per group and per sample than Feature plots, which can be misleading when cells overlap in the UMAP embedding. To complement the Violin plots, we have also provided a graph showing *Lrg1* expression levels and percentages in all endothelial cell subpopulations in each experimental condition (**Supplementary Figure S3D**). Nevertheless, for obvious reasons, we cannot do this for all other genes.

We agree with the reviewer regarding the risk of potential cell contamination, and this was one of our major concerns for our analysis. Indeed, we first tried to prevent this issue using standard tools to detect both doublets and ambient RNA. Concerning doublets, such contamination was reduced given that we multiplexed all samples from the same age and time point using Cell Hashing (6 samples by run). Such design allowed the identification of doublets corresponding to droplets containing 2 barcoded cells with 2 different HTOs. As a complement to remove the remaining doublets (cells barcoded with the same HTO), we used “Scrublet” (Single-Cell Remover of Doublets) tool on RNA counts. However, as most of cells classified as doublets corresponded to real cell populations with a non-ambiguous profile, we preferred to not apply the correction, assuming that the few remaining true doublets wouldn't interfere with the interpretation of our results.

We also chose against applying an ambient RNA correction with a tool such as SoupX, as the resulting data transformation tended to confuse expression profiles of endothelial and immune cells. We believed that this phenomenon was due to the over-representation of these low-UMI cells at the boundary between cell-containing droplets and empty droplets from which ambient RNA is estimated. Moreover, some of the suspicious genes mentioned in the reviewer's comments have been previously shown to be associated with specific subpopulations of endothelial cells. For example, *CD74* has been shown to increase in endothelial cells from PAH patients, contributing to the abnormal pro-inflammatory phenotype of pulmonary ECs in PAH (**PMID: 26203495**). Large single-cell projects (**PMID: 32183954**) also indicate that lung capillary ECs expressed a signature of genes involved in MHC-II-mediated antigen presentation and processing at higher levels than other phenotypes, suggesting a function as semi-professional antigen presentation cells. Interestingly, the authors describe a novel capillary phenotype that might be induced by tumor-derived cytokines, called scavenging capillaries expressing significant levels of *CD52* as well as several genes associated with macrophages and antigen processing.

However, we agree that obvious non-endothelial markers, such as *Sftpc*, *Scgb1a1*, or *S100a9*, were still found in our differential analyses between young and old mice. As those genes were all characterized by a very high and specific expression (see **Fig. Rev.1**), we used the supplemental table S2 to detect those with a $\log_2FC > 4$ in non-endothelial populations. In doing so, we have eliminated contaminating genes and modified the figures associated with the differential analyses between old and young gCap, as well as the **Supplementary tables S4 and S5**. Finally, we also checked experimentally the expression of one of the markers mentioned by the reviewer using RNAScope ISH. The result demonstrates a stronger signal of *Cd74* in lungs from BLM-treated aged mice as well as partial co-localization of *CD74* with *Lrg1* (**new Fig. 5C**). Overall, we now believe that we are proposing a list of reliable markers that are specifically deregulated in old gCap.

3. Gene expression and some other endothelial cell phenotypes related to aging and fibrosis need better verifications. Alveolar niche regeneration and pro-inflammatory phenotype by Lrg1+ PCEC subpopulations are highly descriptive via the changes in some representative gene transcription and bioinformatic pathway/interaction analysis. Some protein-level-based verifications (on the pro-inflammatory/Hypoxia-regulated/pro-angiogenic genes) or functional studies (on the alveolar niche regeneration) are highly suggested.

We confirmed the expression of several important hypoxia pro-regenerative-related genes (Vegfa, Sox17, Hif1a1) in Lrg1^{pos} gCap from both young and old animals using RNAscope on an independent serie of samples (**Fig. Rev.2 and new Fig. 3E**). Quantification of single, double- and triple-positive cells indicated that the source of these pro-angiogenic genes comes predominantly from Lrg1^{pos} gCap during the resolution process (red versus green histograms) in both young and aged mice (**Fig. Rev.2**).

While *in vivo* functional studies were not feasible within a reasonable time frame, we also provide *in vitro* functional experiments on human lung microvascular endothelial cells (HMEC) to show that LRG1 can promote activation of the pro-angiogenic Smad 1 /5 signaling, stimulating VEGFA expression (**new Supplemental fig. S4**), confirming the pro-angiogenic action of LRG1, as previously proposed (**PMID: 23868260**). We recognize that these experiments do not fully recapitulate the complex *in vivo* settings but we believe that our data now strongly suggest that Lrg1-associated pro-angiogenic activity promotes alveolar vascular repair and that delayed expression of Lrg1 in aged mice may likely slow down the formation of a new mature capillary network thus causing a delay in the regeneration of the alveolar niche.

4. The aCap and gCap in fibrotic lungs have been discussed in several previous publications, but the concept of the Lrg1+ PCECs is novel. However, some critical questions this study should have addressed: what signal is driving the emergence of this specific subpopulation and what cell types do they give rise to in the lungs of resolution stage. Without additional functional and lineage tracing studies, its role in lung fibrosis (and in aging) is largely speculative.

The main hypotheses regarding the signaling pathways involved in the activation of Lrg1^{pos} PCEC are presented in the circosplot (**Fig.3G**). As already shown above (point #3), our *in vitro* data indicate that Lrg1 overexpression is associated with a proangiogenic phenotype (**new Supplemental fig. S4**). However, final validation of the role of this population *in vivo* (lineage tracing experiments) requires additional mouse models and cannot be achieved within a reasonable time frame.

5. Another concern regarding the Lrg1+ PCECs is their frequency in each subpopulation. Actually, the Lrg1+ PCEC are highest in PV EC at all time points (Figure S2B-C), are there cells distinct from the main PV EC population? It seems that the percentages of Lrg1+ gCap and Lrg1+ aCap are extremely low in the scRNA-seq analysis (Figure 3D and 4A), however in the in-situ hybridization staining (Figure 4E), their percentages are much higher in both young and aged lungs. It is necessary to verify the dynamic change of Lrg1 in lungs of different timepoints in others' dataset (Figure S2F), but it would be more helpful to see if the distinct Lrg1+ gCap or Lrg1+ aCap subpopulations can be identified in this dataset. PV EC are the endothelial population with the most modulated gene signature between the fibrotic and physiological states following gCap (see **Fig. 2C**). However, this variation in expression profiles did not lead to the identification of a distinct cluster as it is the case for Lrg1^{pos} aCap, Lrg1^{pos} gCap and sCap. It also appears that the comparison of the fibrosis-associated gene signature in PV EC compared with aCap/gCap shows a limited pathological signature in PV EC (see **Fig. Rev.3**). Furthermore, this associated gene signature does not show strong enrichment for activation of angiogenesis or vasculogenesis, migration and proliferation of ECs, as for sCap, Lrg1^{pos} gCap and Lrg1^{pos} aCap (**Fig. 3A-B**).

There are indeed some differences between the percentages of Lrg1^{pos} subpopulations (mainly Lrg1+ aCap) calculated from the scRNA-seq experiment and ISH, but these differences are expected due to dissociation biases (aCap are probably more fragile) and differences in the sensitivity of the 2 techniques.

Regarding other datasets: In Strunz *et al.* (PMID: 32678092), Lrg1^{pos} gCap or Lrg1^{pos} aCap subpopulations could not be detected as distinct clusters from the physiological subpopulation, because vascular endothelial cells are far less abundant (4% in Strunz *et al.* compared to 19.5% in our dataset) and because the overall UMI content per cell is lower (781,4 in Strunz *et al.* compared to 5516 in our dataset). Unfortunately, tracking Lrg1^{pos} EC population dynamics is therefore not possible in Strunz's dataset.

6. Figure 5 presented the pathological changes in endothelial cells in IPF and mouse fibrotic lungs, which is not directly related to "aging". The data in this figure did not show any advances compared to previous publications (e.g., PMID: 34030460). The relationship to Lrg1 is not clear. The conclusion with COL15A1 is speculative.

The reviewer has misunderstood the aims and the conclusions of this paragraph. We felt it was important to perform a comparative analysis of gene expression and cellular composition of EC populations between the most studied pulmonary fibrosis model in which resolution occurred (even in old mice) and the human situation (in which no resolution is observed). As our study is the first to describe the emergence of a Col15a1 positive population in fibrotic mouse lungs, it was also important to compare the gene signature of this specific population with the human counterpart, which was indeed well characterized in the publication mentioned (PMID: 34030460). Three main original conclusions could be drawn from this analysis: i) a conserved pathological signature was found in col15a1 positive systemic EC from fibrotic mouse and IPF lungs, confirming their close functional state; ii) the absence of LRG1^{pos} PCEC in lungs from IPF patients; iii) however, a conserved pro-angiogenic signature between PCEC from mouse fibrotic lungs and human IPF systemic EC, suggesting that a dynamic process may link these subpopulations. These data not only highlight the complex and contrasting situation between the bleomycin mouse model and the human IPF, but also provide new hypotheses to better understand the origin of Lrg1^{pos} PCEC. We have edited this section and our conclusions to better clarify these points. We also provide a better characterization of the localization and dynamic of mouse systemic EC population with new ISH experiments (new Fig. 4B).

7. One of the main disappointments is the failure to identify LRG1+ EC in IPF, which significantly diminishes the importance of LRG1+ EC in human disease.

As discussed above, it is important to show the absence of LRG1^{pos} PCEC in lungs from IPF patients, even if this represents a negative result. As IPF corresponds to an advanced and irreversible stage of the disease, it may come as no surprise not to find a cell subpopulation / state associated with lung repair. We have extensively edited the results section in the new version of the manuscript in order to sequentially compare the data found in the BLM model with those from IPF (see lines 176-195 and 274-288).

8. Figure 6 needs a more rigorous and comprehensive analysis. It includes only two cell (sub)types. An unbiased analysis including ALL endothelial cells and all timepoints is needed. It seems that the Lrg1+ gCap are derived from SV EC and differentiate into gCap in fibrosis models of both young and old mice, when only two cell types were included.

We are aware that, by definition, it is impossible to produce an unbiased RNA velocity analysis due to the many limitations surrounding its modelling assumptions (PMID: 36094956). We therefore chose to apply the velocity analysis in a context that we considered as adding the least circumstantial bias to this set of inherent pitfalls, following the recommendations of the authors of scVelo (PMID: 34435732). First, we performed each analysis on simultaneously sequenced cells so that differences in depth observed between different time points (new Supplemental Fig. S2A) would not affect the estimation of model parameters. Furthermore, we know that trajectory inference becomes confusing in the presence of multiple cell lineages likely to contribute to contradictory kinetic regimes or in the absence of intermediate stages between terminally differentiated cell types. Indeed, when we performed a RNA velocity analysis including all vascular endothelial subpopulations (excepting the proliferative cell state) from either young D14 or old D28 mice, we observed that the Col15a1^{pos} systemic EC (now

referred as systemic Cap (sCap) still initiated the differentiation dynamics, which ended up in physiological aCap (**Fig. Rev.4**).

However, the expression pattern of genes supporting this trajectory in cells ordered by latent time confirmed that the aCap pool cannot be connected to the gCap pool by an intermediate cell state. In the meantime, representing Arterial EC and PV EC on this trajectory is confusing and suggests that their dynamics are independent, because they are placed in the middle of the trajectory, whereas their genes are expressed in a specific and discontinuous manner compared to those of the sCap, Lrg1^{POS} gCap and gCap continuum (**Fig. Rev.4 and Rev.5**).

We therefore chose to use the RNA velocity to infer a trajectory among cell states whose profiles form an expression gradient suggesting a common lineage, *i.e.*: sCap (formerly “SV EC”), Lrg1^{POS} gCap and gCap. Reproducing this trajectory on 2 independent sets of cells has not prevented us from being cautious about the value of this analysis, which provides only a hypothesis that would require a lineage tracing model to be demonstrated, as mentioned in the discussion section.

As recommended by **Reviewer 3 (point #10)**, we have replaced UMAP embedding with diffusion map embedding for the velocity analysis in the **new Fig. 6A-B**.

9. There are many over-speculative statements, for example, “in the human pathology, SV EC are likely not able to differentiate and accumulate, contributing to persistent fibrosis”. There is no experimental evidence presented.

We agree that this conclusion should be removed from the results section. These hypotheses are now addressed in the discussion section.

Minor

10. Cell number of the scRNA-seq: 4,4541 (line 133) or 4,5311 (Figure 2)?

We have corrected this error; the correct number is 45,311.

11. Lymphatic ECs were just mentioned in a few panels in supplemental figures, but did not show up in any figures.

Lymphatic ECs are also shown in **Fig. 2B**, but their numbers are too negligible to be analyzed further (44 cells).

12. Please define the “bleomycin-induced signature score” in Figure 4B.

The bleomycin-induced signature score is already defined in the Materials and Methods section “Detailed analysis of endothelial cells (EC)”, lines 718-725. We have also added a p-value from the Wilcoxon rank sum test and adjusted for multiple comparison to compare the expression of this score between cells grouped by age and time points (**Fig. 4C**).

13. Many quantification analyses are in lack of statistics (Figure 2D, 4A-B, 5B, and many in sup-Figures).

We have performed statistics for both differential abundance analyses and differential expression of the bleomycin-induced signature score (see new Materials and Methods section, lines 713-717 and 724-725).

Reviewer #2 (Remarks to the Author):

This is an interesting manuscript that describes a potential role for aberrant differentiation of pulmonary capillary endothelial cells (PCECs) in the impaired (and delayed) regeneration of the lung following fibrotic lung injury. Specifically, the expression of Lrg1^{POS} gCap/aCap within the alveolar niche is delayed in aged mice, suggesting that this may impair the formation of a new mature capillary network. They also describe an accumulation of COL15A1+ systemic venous ECs (SV-ECs) in aged mice and hypothesize that these cells (also found in IPF) may contribute to delayed regeneration and persistent fibrosis.

The main concern with the current manuscript is its descriptive nature without actual experimental evidence for how these EC populations either support or impair alveolar regeneration. If in-vivo studies with gain and/or loss of function studies are “beyond the scope”, then at the very least ex-vivo (surrogate) studies with angiogenesis assays could provide more insights into the specific role of these EC subtypes and their reprogramming during the repair/regenerative process.

We thank the reviewer for his/her positive comments and we understand his/her point regarding the lack of functional experiments. Unfortunately, *ex vivo* experiments on sorted gCap have been hampered by the lack of valid antibodies against mouse *Aplnr* as well as *Lrg1* (and other valid markers such as *Ackr3* or *Cd94*). We have performed several attempts using commercially available antibodies but none were of high enough quality for robust FACS applications, requiring the use of *Aplnr* (or *Plvap*) Cre-ERT2 transgenic mice, which was not feasible within a reasonable time frame. In this context, we now provide *in vitro* functional experiments on human lung microvascular endothelial cells (HMEC) to show that LRG1 can promote activation of the pro-angiogenic Smad 1 /5 signaling, stimulating VEGFA expression (new Supplemental fig. S4), confirming the pro-angiogenic action of LRG1, as previously proposed (PMID: 23868260). We recognize that these experiments do not fully recapitulate the complex *in vivo* settings but we believe that our data now strongly suggest that *Lrg1*-associated pro-angiogenic activity promotes alveolar vascular repair and that delayed expression of *Lrg1* in aged mice may likely slow down the formation of a new mature capillary network thus causing a delay in the regeneration of the alveolar niche.

Reviewer #3 (Remarks to the Author):

Comments for “Aging affects reprogramming of murine pulmonary capillary endothelial cells after lung injury”

The authors utilize scRNA-Seq, spatial transcriptomics and in situ hybridizations to analyse gene expression changes upon bleomycin induced injury in young and old mice. A strength of eth work is that they collect data during the establishment and resolution of lung fibrosis (days 14, 28 and 60) of lung fibrosis and its resolution. Previous studies on lung fibrosis using similar methods have described changes in alveolar macrophages and the appearance of *Lrg1*pos endothelial cells. The authors have additionally utilized bioinformatic tools to predict cell communications that are affected in the injured and/or old samples may drive the gene expression changes. They also compared their mouse data to previous data from lung fibrosis patients providing suggestions about species related differences that result to fibrosis resolution only in the mouse but not human. In general, the study is a systematic effort to analyse the age-related differences in the lung response to bleomycin induced injury but they mainly focus on endothelium. It would have been interesting to the reader to see an initial analysis of all detected cells states since a major strength of the study is the tight sampling of time points. The inclusion of such an analysis and the publication of all data before focusing on the endothelium would increase the general interest on the paper and may provide a better basis for the cell-interaction analyses.

We thank the reviewer for his/her positive and constructive comments. We also agree that it would have been interesting to follow cells states for additional cell types during the course of fibrosis and resolution. Unfortunately, the number of captured cells was too low in the case of epithelial and fibroblasts to perform a complete cell state analysis. We did, however, carry out a parallel analysis of the macrophage population, which plays a central role in disease initiation and progression, whose dynamics interestingly did not show significant differences associated with aging at the different time points (see Supplemental fig. S2D-F).

Main Comments:

1) Even if the results are presented clearly in the figures, the analysis methods are not explained enough. For example, in lines 119-120 the authors state that they used “an unsupervised assignment of each spot.” but they do not describe how they did it in the methods.

We have revised and improved our analysis of the visium dataset, using a more ad hoc tool to process such spatial transcriptomic profiling with a multi-cellular pixel-resolution (new Fig. 1D-F). The new analysis process is described in the dedicated section of the Materials & Methods (lines 594-619). Briefly, we have chosen to use STdeconvolve [PMID: 35487922] to disentangle the spatial patterns within each spot without leveraging on scRNA-seq reference profiles. Indeed, our first analysis revealed some discrepancies between spatial and dissociated data in terms of cell type coverage in the lung, which strongly limited the use of a scRNA-seq reference to annotate spatial spots, even if both types of data come from the same tissue type. Among these differences, we noted in particular the presence of expression modules in spatial data not captured in our dissociated data, such as areas enriched in immunoglobulins, connective tissue, adipose tissue or respiratory epithelium. Coupled with differences in the expression distribution of marker genes from cell populations identified by single-cell transcriptomics, we opted for an approach less sensitive to the deconvolution biases potentially generated by using a scRNA-seq reference.

2) In Figure 1 and Suppl. Figure 1, the authors present histology panels from both PBS and Bleo treated lungs but they do not show any Visium results from PBS treated lungs. This is a drawback because it is unclear how they have determined the “normal” gene expression conditions in alveolar, airway, interstitial, vessel and SMC layer. They also do not describe any differences between PBS young and old mice.

In designing the Visium experiment, it became clear to us that the structure of healthy lung tissue was not optimal for Visium technology, as the majority of spots would have been located under empty tissue areas. This intuition was recently verified in a study using the same spatial technology to characterize human or bleomycin-induced pulmonary fibrosis (PMID: 38951642), in which slides corresponding to healthy tissue showed a much lower number of genes detected per spot than slides of pathological tissue. We therefore opted to perform the experiment on denser bleomycin-remodelled tissue, and to focus the analysis on potential differences in fibrosis formation and resolution between young and old mice.

3) In Figure 1D, the barplot shows a significantly increased percent of “low counts alveoli” in the old D28 lungs. Is that because of emphysematous or fibrotic/scarred areas or just stems from a technical issue? The normal, PBS treated sample would address this.

As we were unable to answer this question, we decided to impose minimum thresholds for the expression of mRNA and annotated non-coding genes in the reanalysis of the Visium data (see new Materials & Methods, lines 594-619). In this way, 502 spots with low UMI content were eliminated from the analysis. Nevertheless, we found that lung sections from mice aged at D28 still contained a number of spots associated with a topic characterized by overexpression of *Malat1* and *Lars2*, ahead of alveolar epithelial cell markers. As these two genes are known to be highly expressed and localized in the nucleus, it is likely that this overexpression is indicative of the prevalence of cytosol-free nuclear debris in these alveolar areas. However, it is impossible to determine whether this is a technical problem or a result of treatment-induced damage. We have therefore not considered this topic in the comparison of lung sections from young and old mice.

4) The authors used a mild tissue dissociation approach that might have resulted to incomplete cell release and introduction of bias. This could explain the small number of epithelial cells compared to the abundant immune cells. They should at least comment on this limitation and its impact in the described cell-type frequencies, especially for the fibrotic lungs, which should be more difficult to dissociate. Cell type frequencies would be best addressed by the Visium experiment. The authors have

not assigned cell types to their spatial reads, as it is commonly done in this type of experiments. What is the reason for this?

These are interesting and complex questions. Below are our answers to these 2 points:

- Cell-type frequencies and dissociation protocol bias: we have included a comment in the discussion regarding cell preparation for Chromium and its impact on cell type repartition (lines 473-482). Depletion of fibroblasts was mainly due to the Cell Hashing process which resulted in around 70% loss, probably due to additional labelling and washing steps. Overall, our protocol resulted in the depletion of epithelial cells and fibroblasts and to an enrichment of endothelial cells, allowing an in-depth analysis of the endothelial cell sub-populations.

- Cell-type frequencies and Visium experiment: although Visium technology skips the dissociation step, its resolution is too low to perform single-cell characterization, as each 55µm diameter spot contains several cells. Moreover, as the alveolar niche is formed by the association of cell types of different lineages (epithelial, mesenchymal, endothelial, immune), this implies that each spot is a combination of the transcriptome of heterogeneous populations. We are aware that numerous tools designed to estimate the contribution of each population by deconvolution based on reference expression profiles of each cell type exist (PMID: 36941264). It should be noted, however, that these tools are mostly developed and compared from brain tissue sections, with much more homogeneous histological compartments making their annotation from dissociated data more obvious. As indicated above, we chose to use a deconvolution tool that identified spatial topics only based on spatial gene expression profiles, as the lung is a more complex model to rely on scRNA-seq reference for deconvolution (see **new Fig. 1D-F**).

We have calculated correlations between the profiles of the scRNA-seq data and the topics identified in the spatial data (see **Fig. Rev.6**). While the profiles of some cell types match well with the signatures of specific spatial topics, in particular for epithelial cells and some immune cells, other profiles are not or poorly represented. Moreover, each slice has its own histological particularities and therefore its own tissue characteristics. It thus appears that each approach has its own intrinsic biases and only gives an approximate idea of the real distribution of cell types.

5) In the lines 140-141 the authors describe an expected shift in several populations following bleomycin. But they do not specify if that refers to changes in the frequencies of cells (lines 142-144) or gene-expression changes in a given cell-type or both (Fig. 2D).

It is difficult to differentiate gene-expression changes from cell population changes as both phenomenon may occur simultaneously. For example, the shift induced by bleomycin in alveolar macrophages is at the same time a consequence of recruitment of monocyte-derived macrophages (AM3) or a consequence of resident macrophages polarization to a M2 phenotype, which both contribute to modulation of cell populations frequencies. In other cases, where cellular dynamics are less well known, it is difficult to determine whether the phenotypical shift occurs solely within a pool of resident cell types through differentiation, or whether it requires the contribution of an ectopic population. Moreover, in scRNA-seq data, a strong shift in gene expression can lead to the identification of a distinct cell cluster from the physiological population, which will change the overall repartition if considered as a particular cell state. In our manuscript, we both present Lrg1^{POS} aCap/gCap as cell states or gene expression signatures induced in aCap/gCap upon bleomycin treatment.

6) In Figure 2 a better representation of the changes in cell-type frequencies in all conditions, that will also account for the different library sizes should be used. Also, it might be informative to examine if the differences in alveolar macrophages states upon bleomycin treatment have any specific spatial distribution. Are they found in severely damaged or healthy areas or at their borders?

The heterogeneous size of the library may have an impact on the identification of clusters associated with the different cell states and thus on their frequencies, but this problem is largely minimized by the integration process. Indeed, the cell states frequencies found in PBS samples are highly similar, while they are coming from 3 time points (D14, D28, D60) with various library size (**new Supplemental fig. S2A**).

The Visium data indicated that we cannot discriminate between AM2 and AM3 signatures, which are both highly correlated with the topic 1, a topic corresponding to fibrotic areas (cf Supplemental Figure S1C + figure rev. 6). As expected, AM1 were more widely distributed and found also correlated with topics corresponding to both fibrotic (topics 1 and 13) as well as healthy or repaired areas (topics 2 and 5).

7) The authors indicate that “transcriptional remodelling of gCap into Lrg1+ transitory subpopulations may affect their abilities to differentiate and replenish the pool of functional aCap” (lines 250-252). Importantly they have observed that this transition in both young and old lungs but only the old showed impaired healing. Also, this conclusion comes in contrast to their discussion point in lines 398-401 (“Lrg1-associated pro-angiogenic activity appears to be necessary to promote alveolar vascular repair...”). Thus, their findings might better reflect a compromised reparatory capacity of old cells.

We agree with the reviewer's comment. The old gCap also expresses a profibrotic signature, but this conclusion is premature at this stage of the analysis. We have suppressed the text accordingly.

8) In Figure 4, the authors present convincing results about the expression of Lrg1, Aplnr and Ednrb on tissue sections but it will be informative to examine the localization of these cells relatively to the injured areas and specify it (as in comment 6).

It was not possible to show histological analyses on the same slides but sections were selected to focus on fibrotic areas. As the **new Fig. 3E** and **Fig. 4D** clearly show, the Lrg1 signal is localized in the areas of high cell density typical of fibrosis, compared with the PBS control.

9) In Figure 4G, two of the significantly changed genes between the OLD vs YOUNG gCap in PBS are the “Scgb1a1” and “Sftpc”. These markers are expressed in airway and alveolar secretory cells and not vascular endothelium. Thus, some of these cells are likely doublets or mRNAs of these highly expressed epithelial genes were released from lysed/destroyed epithelial cells during tissue dissociation. This can happen but more stringent filtering should be applied regarding the expression levels of the processed genes to avoid technical noise.

We agree with the reviewer regarding the risk of potential cell contamination, and it was one of our major concern in our analysis. This point was also raised by **Reviewer #1**, please see **comments #2** and **new Fig. 5**. We indeed applied a stringent filter but a few genes still emerged. We agree that obvious non-endothelial markers, such as Sftpc, Scgb1a1, or S100a9, were still found in our differential analyses between young and old mice. As those genes were all characterized by a very high and specific expression (see **Fig. Rev.1**), we used the **supplemental table S2** to detect those with a $\log_2FC > 4$ in non-endothelial populations. In doing so, we have eliminated contaminating genes and modified the figures associated with the differential analyses between old and young gCap, as well as the **supplementary tables S4 and S5**. Overall, we now believe that we are proposing a list of reliable markers that are specifically deregulated in old gCap.

10) In Figure 6, the authors performed pseudotime analysis to analyse the transition of systemic vascular endothelial cells to gCap using UMAP dimension reduction. It would be better to use Diffusion Maps, because it performs more efficiently for that type of analysis. It would also be helpful to include aCap cells as the end state, these are destroyed by bleomycin and are expected to become repaired after the injury.

We have followed the recommendation of the Reviewer and replaced umap embedding to diffusion map embedding for the velocity analysis in the **new Fig. 6A-B**. We also tried several times to include aCap to the trajectory, as they are supposed to be the product of gCap differentiation in physiological condition and following lung injury [PMID: 33057196] (see **Fig. Rev.4 and Rev.5** as well as **point #8 of Reviewer 1 comments**). However, it seems that we were unable to capture any transitional state between gCap and aCap by scRNA-seq. This is firstly seen in the RNA counts space, where gCap and aCap are two distinct clusters of terminally differentiated cells without any expression gradient between them. This is also the case in the velocity analysis, where there is no dynamic linking gCap to

aCap. Indeed, the Lrg1^{pos} aCap population is predicted to initiate 2 distinct trajectories, the first leading to physiological aCap, and the second leading to physiological gCap through systemic EC (now referred as systemic Cap / sCap) and Lrg1^{pos} gCap. However, this last hypothetical trajectory is not supported by transitory gene expression pattern when the cells are ordered by latent pseudotime.

11) In lines 139-140 and 361 the authors mentioned that they have identified a population of proliferating endothelial cells but they do not provide any further information about possible differences in the number and gene expression profiles of these cells between young and old mice. The only UMAP showing endothelial cells is in Figure 2 and there, the assigned proliferating cells are very few.

Indeed, we identified this population in our initial clustering on the full dataset, as the cell cycle gene signature it carries is easily detectable in the scRNA-seq data despite the small number of cells (180 cells). We have included this population in the description of the heterogeneity of pulmonary endothelial cells (**Fig. 2D-F**), where their expression profile appears to be close to that of gCap, although their UMI content is mainly predominated by cycle genes. We also indicated their average relative proportion at different time points (**Fig. 2E**), which we then detailed by age and time point in **Fig. 4A**. We found that this cellular state is transiently activated following bleomycin-induced injury, then disappears upon resolution, and that this dynamic is similar between young and old mice, although the percentage appears lower for aged mice at day 14. As this proliferative state is specific to the bleomycin treatment, which we also observed by IHC of the MKI67 marker (**Fig. Rev.7**), it supports the hypothesis of an angiogenic response to lung injury. However, the small number of cells and their cell-cycle-oriented signature profile did not enable us to quantify the potential differences between young and old mice.

12) My final comment relates to the analysis approach of considering day60 post bleo administration as healthy (In Suppl. Fig. 1 A, it is obvious that both young and old lung tissue condition is improved but differs from PBS). Also, I am sceptical of the merging young and old day14 and 28 datasets of PBS and bleomycin- treated mice (lines 739-750) and sometimes including day60 (lines 783-784) because this approach might dilute differences that are temporally restricted to one timepoint and/or age. Differential expression between samples of the same timepoint and age should be the basis for identifying the induced or suppressed genes in response to bleomycin treatment and/or aging in all cell-types.

We would like to thank the reviewer for raising this important issue, which has been a subject of discussion during the analysis and requires clarification.

First, with regard to differential expression analysis using the pseudobulk approach, samples collected at d60 post-bleomycin administration were never included to compare BLM with PBS-treated mice or to evaluate the impact of aging on the bleomycin response. They are not considered as healthy and confounded with PBS samples, as it is clear that these lungs have been subjected to a process of remodeling. Our control set of samples is thus composed by all PBS-treated lungs collected at D14, D28 and D60, i.e. 9 samples for young and 9 samples for old, as we consider those samples as “biological replicates” despite the few weeks delay between their collection.

Secondly, our set of samples representing the injured lungs condition is indeed composed of BLM-treated samples collected at D14 and D28. This choice to merge both time points rather than performing a separated analysis was motivated by both biological and technical considerations to perform a fair pseudobulk comparison between samples from young and aged mice following bleomycin treatment. Two factors were taken into consideration:

- the variability of the cellular and molecular signals of bleomycin-induced fibrosis that is captured between individuals by scRNA-seq: as BLM-treated individuals showed some heterogeneity, both in terms of cell states proportions and gene expression changes compared to other replicates, we included all samples from BLM-treated D14 and D28 mice in order to double the number of samples taken into account for estimating dispersion within the BLM condition (6 samples rather than 3 in a

pseudobulk approach with DESeq2), even though the signal was probably attenuated by the inclusion of samples with a low injury response signature.

- the resolution time shift between young and aged mice: indeed, comparing young and old only at D14 or only at D28 would not have been informative to describe age-specific gene-level response to injury, as both time points represent different stages of the resolution kinetic. Considering that the most representative time point of bleomycin-induced response is at D14 in young and at D28 in old in our scRNA-seq dataset, we only found few differentially expressed genes when comparing BLM-treated old and young gCap (**Fig. 5A**). Moreover, we observed a Pearson's correlation coefficient of 0.881 when comparing Log2 FC (BLM/PBS) at D14 vs Log2 FC (BLM/PBS) at D28, confirming that aging affects the resolution dynamics rather than a strong modulation of gene expression (**Fig. Rev.8**).

Reviewer #4 (Remarks to the Author):

In the submitted manuscript, researchers tried to find potential role of endothelial cells in age related fibrosis formation and resolution context. For this purpose, they used multi-omics approach to verify their histology and in vivo results via Single Cell RNA sequencing and Spatial Transcriptomics. They first evaluated the changes induced by aging and fibrosis and figured out that the endothelial cells are the second most affected cell type after macrophages. They defined novel pulmonary capillary endothelial subclusters in their mouse model (Lrg1+ PCEC) that contributes to aging associated angiogenesis and alveolar niche regeneration. Furthermore, they discovered aging induced potential pro-fibrotic role of gCaps, which may influence the impairment of alveolar regeneration. Also, they showed that aging induces shift in the existence of Lrg1+gCap and Lrg1+aCap in young and aged mice model where they prove their contribution to remodelling of PCEC and potential role in fibrosis resolution. To identify the role of these subclusters, they performed RNA Velocity to indicate the transition process of Lrg1+gCap and potential progenitor systemic venous vessels (SV EC) towards gCaps. They were able to observed similar subcluster of Col15a1 SV EC in human single cell data that is upregulated in IPF patients, yet couldn't able to identify Lrg1+ subclusters.

Main problem:

1- Mouse data findings especially related with Lrg1+ PCEC subclusters do not exist in the human IPF single cell RNA sequencing data

We understand the necessity to discuss this point. Comparing IPF with mouse pulmonary fibrosis models is a complex issue. We felt that it was important to compare gene expression and cellular composition of EC populations in the most studied pulmonary fibrosis model in which resolution occurred (even in old mice) and in the human situation (in which no resolution is observed). Importantly, our study is the first to describe the emergence of a Col15a1 positive population in fibrotic mouse lungs and it was thus important to compare the gene signature of this specific population with the human counterpart. Three main conclusions could be drawn from this analysis: i) a conserved pathological signature was found in Col15a1 positive systemic EC from fibrotic mouse and IPF lungs, confirming their close functional state; ii) the absence of LRG1^{POS} PCEC in lungs from IPF patients; iii) however, a conserved pro-angiogenic signature between PCEC from mouse fibrotic lungs and human IPF systemic EC, suggesting that a dynamic process may link these different subpopulations. Overall, our data highlight the complex and contrasting situation in mouse bleomycin model and in human IPF, but also provide new hypotheses to better understand the origin of Lrg1^{POS} PCEC. We have extensively edited the results section in the new version of the manuscript in order to sequentially compare the data found in the BLM model with those from IPF (see lines 176-195 and 274-288).

Open questions:

1- Researchers performed spatial transcriptomics to validate their histological findings of „delayed fibrosis resolution“ in aged mice model. However even they have n=3 for histological findings in old and young mice model, they performed Visium spatial transcriptomics by using n=1 for young mice

and n=2 for old mice. The differential gene expression is not valid to calculate p-adj method. Furthermore, the number of cells detected will be less from n=1 young group which gives misleading UMAP plot. Also in reanalysis of spatial data for Lrg1+ cells in Fig4.C, it is clear that old D28a and old D28b samples do not match with each other. In the method part they explained why they had to use n=1 for young mice, yet the problems are still valid.

Our response is grouped with point 2, see below.

2- Researchers used PBS as negative control for histological analysis but no control group used in spatial transcriptomics.

We decided to perform Visium experiments in order to draw hypotheses regarding the potential molecular and cellular responses discriminating the shift in the resolution process in aged versus young mice. Indeed, our design did not allow to perform statistics but rather to complete our histological analyses with both cellular and molecular information. Moreover, in designing the Visium experiment, it became clear to us that the structure of healthy lung tissue was not optimal for Visium technology, as the majority of spots would have been located under empty tissue areas. This intuition was recently verified in a study using the same spatial technology to characterize human or bleomycin-induced pulmonary fibrosis (PMID: 38951642), in which slides corresponding to healthy tissue showed a much lower number of genes detected per spot than slides of pathological tissue. We therefore opted to perform the experiment on denser bleomycin-remodelled tissue, and to focus the analysis on potential differences in fibrosis formation and resolution between young and old mice.

We have significantly revised and improved our analysis of the visium dataset (new Fig. 1D-F), providing a more precise characterization of the cellular and molecular alterations occurring in lungs from bleomycin aged mice. The new analysis process is described in the dedicated section of the Materials & Methods (lines x-y). Briefly, we have chosen to use STdeconvolve [PMID: 35487922] to disentangle the spatial patterns within each spot without leveraging on scRNA-seq reference profiles. Indeed, our first analysis revealed some discrepancies between spatial and dissociated data in terms of cell type coverage in the lung, which strongly limited the use of a scRNA-seq reference to annotate spatial spots, even if it is coming from the same tissue type. Among these differences, we noted the presence of expression modules in spatial data not captured in our dissociated data, such as areas enriched in immunoglobulins, connective tissue, adipose tissue or respiratory epithelium. Coupled with differences in the expression distribution of marker genes from cell populations identified by single-cell transcriptomics, we opted for an approach less sensitive to the deconvolution biases potentially generated by using a scRNA-seq reference.

3- In Figure1 description, damaged alveolar regions are characterized by low UMI content yet the heatmap in Supp.Fig1D shows that it is not specific to age related differences. Therefore, it would be better to include DEG Volcano plot that compares young vs old mice tissue in “low counts Alveoli” subset to support their aim in the first part of the result as “fibrosis resolution is delayed lungs from aged mice”. Furthermore, it would be better to use cluster annotation age to cluster the rows in heatmap in Supp.Fig1D.

We have revised the analysis of Visium experiment (new Fig. 1D-F) and now provide new information illustrating the delay of resolution in aged mice (see revised results section lines 116-133). Regarding the specific point about low UMI content, we have decided to impose minimum thresholds for the expression of mRNAs and annotated non-coding genes in the reanalysis of the Visium data (see new Materials & Methods). In this way, 502 spots with low UMI content were excluded from the analysis. Nevertheless, we found that lung sections from aged mice at D28 still contain a number of spots associated with a topic characterized by overexpression of Malat1 and Lars2, ahead of alveolar epithelial cell markers. As these two genes are known to be highly expressed and localized in the nucleus, it is likely that this overexpression is indicative of the prevalence of cytosol-free nuclear debris in these alveolar areas. However, we cannot say whether this is a technical issue or the result of treatment-induced damages. We have therefore excluded this topic when comparing lung sections from young and old mice.

4- In line 123, regarding the comment above, the comparisons are not age related but damaged associated.

As mentioned above, we have edited this paragraph with a new analysis.

5- In Figure3, researchers labelled aCap cells that has higher Lrg1 expression as Lrg1+aCap. Is the difference between two subclusters in terms of Lrg1 expression is significant to assign them as Lrg1 overexpressed (Line 157)? Difference in between gCap is more clear both in heatmap (Fig3.B) and Violin Plot (Fig.3C).

Quantitative values of Lrg1 overexpression in aCap following bleomycin treatment are shown in **Supplementary Table S3** ($\log_2FC=2.93$, $padj = 6.02956E-13$). Moreover, our "Lrg1^{POS}" nomenclature don't rely on the Lrg1 expression only, but rather refers to a signature of pro angiogenic genes such as Sparc, Col4a1, Col4a2, Cd34, Tgfbr2, Hif1a which are also all differentially expressed in BLM aCap.

6- In between lines 159-161, researcher claimed that there is a subpopulation called Prolif.EC, which is visible in the heatmap Fig3B with Mki67 expression. The figure that supports the difference between day14 and day 28 following bleomycin treatment is Fig3D not C.

We have modified the sentence accordingly.

7- Since single cell RNA Seq results indicates the significant downregulation of certain genes (Supp.Fig3B-C) it would be a better approach to verify these findings in spatial transcriptomics.

The Visium resolution is too low to perform single-cell characterization, as each 55 μ m diameter spot contains several cells. Moreover, as the alveolar niche is formed by the tight association of cell types of different lineages (epithelial, mesenchymal, endothelial, immune), this implies that each spot is a combination of the transcriptome of heterogeneous populations. We have calculated correlations between the profiles of the scRNA-seq data and the topics identified in the spatial data (see **Fig. Rev.6**). While the profiles of some cell types match well with the signatures of specific spatial topics, in particular for epithelial cells and some immune cells, other profiles, including PCEC are not correlated with a specific topic and appeared mixed among several.

8- The expression of Smad family members' difference in between Pbs vs Bleo model are not significant for Smad2 and Smad5 (Supp.Fig3C). In the discussion, it is claimed that Smad2 and Smad5 works controversially with the influence of Lrg1 and Tgfb1 signalling. Is there any functional experiment related with the activity of Smad Tgfb and Lrg1? Or is there any transcriptomic data shows that Age+Bleo related differences among these genes that supports the claim in line 400?

We have now provided *in vitro* functional experiments on human lung microvascular endothelial cells (HMEC) showing that LRG1 can promote activation of the pro-angiogenic Smad 1/5 signaling, stimulating VEGFA expression (new **Supplemental fig. S4**), confirming the pro-angiogenic action of LRG1, as previously proposed (**PMID: 23868260**). These data are also in agreement with a strong induction of markers associated with EC proliferation and migration after lung injury to promote angiogenesis and sprouting (**Fig. 3D**).

9- Legend is missing indicating colors of different Bleo treatment days (Fig.4E)

The graph was modified accordingly.

10- Due to the experimental single cell sequencing results of the researchers Lrg1 expression mostly depending on increase in age, how do they define no expression in IPF human data? (except lines 327-328) Did the researchers checked other available data sets? (ex. Ipfcellatlas.com)

We indeed looked at LRG1 expression in 3 scRNA-seq datasets from the IPF Cell Atlas (Banovich/Kropski, Kaminski/Rosas and Lafyatis), and the transcript was almost not expressed.

11- Heatmap indicating D28 old mice transcriptional dynamics is missing (Fig6) supports the claim in line 320

We have added the corresponding heat map, as requested. The two heatmaps are highly similar.

References:

PMID: 32832599. Adams TS, Schupp JC, Poli S, Ayaub EA, Neumark N, Ahangari F, Chu SG, Raby BA, Deluliis G, Januszyk M, Duan Q, Arnett HA, Siddiqui A, Washko GR, Homer R, Yan X, Rosas IO, Kaminski N. Single-cell RNA-seq reveals ectopic and aberrant lung-resident cell populations in idiopathic pulmonary fibrosis. *Sci Adv.* 2020 Jul 8;6(28):eaba1983. doi: 10.1126/sciadv.aba1983.

PMID: 32832598. Habermann AC, Gutierrez AJ, Bui LT, Yahn SL, Winters NI, Calvi CL, Peter L, Chung MI, Taylor CJ, Jetter C, Raju L, Roberson J, Ding G, Wood L, Sucre JMS, Richmond BW, Serezani AP, McDonnell WJ, Mallal SB, Bacchetta MJ, Loyd JE, Shaver CM, Ware LB, Bremner R, Walia R, Blackwell TS, Banovich NE, Kropski JA. Single-cell RNA sequencing reveals profibrotic roles of distinct epithelial and mesenchymal lineages in pulmonary fibrosis. *Sci Adv.* 2020 Jul 8;6(28):eaba1972. doi: 10.1126/sciadv.aba1972.

PMID: 34030460. Schupp JC, Adams TS, Cosme C Jr, Raredon MSB, Yuan Y, Omote N, Poli S, Chioccioli M, Rose KA, Manning EP, Sauler M, Deluliis G, Ahangari F, Neumark N, Habermann AC, Gutierrez AJ, Bui LT, Lafyatis R, Pierce RW, Meyer KB, Nawijn MC, Teichmann SA, Banovich NE, Kropski JA, Niklason LE, Pe'er D, Yan X, Homer RJ, Rosas IO, Kaminski N. Integrated Single-Cell Atlas of Endothelial Cells of the Human Lung. *Circulation.* 2021 Jul 27;144(4):286-302. doi: 10.1161/CIRCULATIONAHA.120.052318.

PMID: 37137915. Bian F, Lan YW, Zhao S, Deng Z, Shukla S, Acharya A, Donovan J, Le T, Milewski D, Bacchetta M, Hozain AE, Tipograf Y, Chen YW, Xu Y, Shi D, Kalinichenko VV, Kalin TV. Lung endothelial cells regulate pulmonary fibrosis through FOXF1/R-Ras signaling. *Nat Commun.* 2023 May 4;14(1):2560. doi: 10.1038/s41467-023-38177-2.

PMID: 38529320. Ge J, Shao H, Ding H, Huang Y, Wu X, Sun J, Que J. Single Cell Analysis of Lung Lymphatic Endothelial Cells and Lymphatic Responses during Influenza Infection. *J Respir Biol Transl Med.* 2024 Mar;1(1):10003. doi: 10.35534/jrbtm.2024.10003.

PMID: 36712020 (preprint). Raslan AA, Pham TX, Lee J, Hong J, Schmottlach J, Nicolas K, Dinc T, Bujor AM, Caporarello N, Thiriot A, von Andrian UH, Huang SK, Nicosia RF, Trojanowska M, Varelas X, Ligresti G. Single Cell Transcriptomics of Fibrotic Lungs Unveils Aging-associated Alterations in Endothelial and Epithelial Cell Regeneration. *bioRxiv [Preprint].* 2023 Jan 20:2023.01.17.523179. doi: 10.1101/2023.01.17.523179. **Article published in Nat. Commun. (Jun 27 2024).** **PMID: 38937456.** Raslan AA, Pham TX, Lee J, Kontodimas K, Tilston-Lunel A, Schmottlach J, Hong J, Dinc T, Bujor AM, Caporarello N, Thiriot A, von Andrian UH, Huang SK, Nicosia RF, Trojanowska M, Varelas X, Ligresti G. Lung injury-induced activated endothelial cell states persist in aging-associated progressive fibrosis. *Nat Commun.* 2024 Jun 27;15(1):5449. doi: 10.1038/s41467-024-49545-x.;

PMID: 26203495. Le Hiress M, Tu L, Ricard N, Phan C, Thuillet R, Fadel E, Dorfmüller P, Montani D, de Man F, Humbert M, Huertas A, Guignabert C. Proinflammatory Signature of the Dysfunctional Endothelium in Pulmonary Hypertension. Role of the Macrophage Migration Inhibitory Factor/CD74 Complex. *Am J Respir Crit Care Med.* 2015 Oct 15;192(8):983-97. doi: 10.1164/rccm.201402-0322OC..

PMID: 32183954. Goveia J, Rohlenova K, Taverna F, Treps L, Conradi LC, Pircher A, Geldhof V, de Rooij LPMH, Kalucka J, Sokol L, García-Caballero M, Zheng Y, Qian J, Teuwen LA, Khan S, Boeckx B, Wauters E, Decaluwé H, De Leyn P, Vansteenkiste J, Weynand B, Sagaert X, Verbeken E, Wolthuis A, Topal B, Everaerts W, Bohnenberger H, Emmert A, Panovska D, De Smet F, Staal FJT, McLaughlin RJ, Impens F, Lagani V, Vinckier S, Mazzone M, Schoonjans L, Dewerchin M, Eelen G, Karakach TK, Yang H, Wang J, Bolund L, Lin L, Thienpont B, Li X, Lambrechts D, Luo Y, Carmeliet P. An Integrated Gene Expression Landscape Profiling Approach to Identify Lung Tumor

Endothelial Cell Heterogeneity and Angiogenic Candidates. *Cancer Cell*. 2020 Mar 16;37(3):421. doi: 10.1016/j.ccell.2020.03.002.

PMID: 23868260. Wang X, Abraham S, McKenzie JAG, Jeffs N, Swire M, Tripathi VB, Luhmann UFO, Lange CAK, Zhai Z, Arthur HM, Bainbridge J, Moss SE, Greenwood J. LRG1 promotes angiogenesis by modulating endothelial TGF- β signalling. *Nature*. 2013 Jul 18;499(7458):306-11. doi: 10.1038/nature12345.

PMID: 32678092. Strunz M, Simon LM, Ansari M, Kathiriya JJ, Angelidis I, Mayr CH, Tsidiridis G, Lange M, Mattner LF, Yee M, Ogar P, Sengupta A, Kukhtevich I, Schneider R, Zhao Z, Voss C, Stoeger T, Neumann JHL, Hilgendorff A, Behr J, O'Reilly M, Lehmann M, Burgstaller G, Königshoff M, Chapman HA, Theis FJ, Schiller HB. Alveolar regeneration through a Krt8+ transitional stem cell state that persists in human lung fibrosis. *Nat Commun*. 2020 Jul 16;11(1):3559. doi: 10.1038/s41467-020-17358-3.

PMID: 36094956. Gorin G, Fang M, Chari T, Pachter L. RNA velocity unraveled. *PLoS Comput Biol*. 2022 Sep 12;18(9):e1010492. doi: 10.1371/journal.pcbi.1010492.

PMID: 34435732. Bergen V, Soldatov RA, Kharchenko PV, Theis FJ. RNA velocity-current challenges and future perspectives. *Mol Syst Biol*. 2021 Aug;17(8):e10282. doi: 10.15252/msb.202110282.

PMID: 35487922. Miller BF, Huang F, Atta L, Sahoo A, Fan J. Reference-free cell type deconvolution of multi-cellular pixel-resolution spatially resolved transcriptomics data. *Nat Commun*. 2022 Apr 29;13(1):2339. doi: 10.1038/s41467-022-30033-z.

PMID: 38951642. Franzén L, Olsson Lindvall M, Hühn M, Ptasiński V, Setyo L, Keith BP, Collin A, Oag S, Volckaert T, Borde A, Lundberg J, Lindgren J, Belfield G, Jackson S, Ollerstam A, Stamou M, Ståhl PL, Hornberg JJ. Mapping spatially resolved transcriptomes in human and mouse pulmonary fibrosis. *Nat Genet*. 2024 Aug;56(8):1725-1736. doi: 10.1038/s41588-024-01819-2.

PMID: 33057196. Gillich A, Zhang F, Farmer CG, Travaglini KJ, Tan SY, Gu M, Zhou B, Feinstein JA, Krasnow MA, Metzger RJ. Capillary cell-type specialization in the alveolus. *Nature*. 2020 Oct;586(7831):785-789. doi: 10.1038/s41586-020-2822-7. Epub 2020 Oct 14.

PMID: 36941264. Li H, Zhou J, Li Z, Chen S, Liao X, Zhang B, Zhang R, Wang Y, Sun S, Gao X. A comprehensive benchmarking with practical guidelines for cellular deconvolution of spatial transcriptomics. *Nat Commun*. 2023 Mar 21;14(1):1548. doi: 10.1038/s41467-023-37168-7.

Additional Figures for Reviewers

Figure Rev.1. Expression of selected genes from the DEG signature between gCap from young and old BLM-treated mice in various cell types. The seven genes on the left have been eliminated from the signature based on their very high expression in specific cell types, thus potentially inducing contamination in the gCap.

Figure Rev.2. Co-expression of Vegfa and Sox17 with Lrg1 and Aplnr in mouse lungs. A. RNA scope assay performed with Vegfa or Sox17 (white), Lrg1 (red) and Aplnr (green) probes at different times points for young and old mice following PBS or BLM treatment. **B.** Quantification of the percentage of Vegfa + or Sox17+ cells co-expressing Lrg1, Aplnr or both markers (3 fields / condition).

Figure Rev.3. Dot plot of top common differentially expressed genes in aCap, gCap and PV EC for the contrast BLM vs PBS in young and old mice.

Figure Rev.4. RNA velocity analysis including all endothelial cells. Analysis was performed using the samples from young mice at day 14 (top) and aged mice at day 28 (bottom). (Top left panel) UMAP plot computed on spliced RNA data (3 Bleomycin and 3 PBS-treated mice) with cells colored according to PCEC subpopulation including in addition to aCap, gCap and sCap (formerly “SV EC”), also PV EC and Arterial EC. (Bottom left panel) RNA velocity latent time. (Right) Heatmap of the top contributing genes to the transcriptional dynamic, with cells ordered by latent time.

Figure Rev.5. RNA velocity analysis including aCaps and gCaps. Analysis was performed using the samples from young mice at day 14. (Up left panel) UMAP plot computed on spliced RNA data (3 Bleomycin and 3 PBS-treated mice) with cells colored according to PCEC subpopulation (sCap: formerly “SV EC”). (Bottom left panel) RNA velocity latent time. (Right) Heatmap of the top contributing genes to the transcriptional dynamic, with cells ordered by latent time.

Figure Rev.6. Heatmap of correlation between topics (Visium, numbered from 1-14) and main cell types from single cell RNA seq (Chromium) annotations.

Figure Rev.7. *In situ* evaluation of proliferating cells in the lungs from BLM-treated mice. Immunofluorescence using antibodies directed against Ki67, Cd31 and α SMA in lungs from PBS- or BLM-treated mice (7 weeks-old) at 2 time points following instillation (D14, D28), showing a partial co-localization of Ki67 and Cd31 in fibrotic areas.

Figure Rev.8. Correlation of expression between samples from young mice at day 14 and old mice at day 28. Pearson's correlation between differential expressed genes from young mice (3 PBS vs 3 BLM at day 14) and old mice (3 PBS vs 3 BLM at day 28).

REVIEWER COMMENTS

We thank the reviewers for taking the time to evaluate our manuscript. We appreciate their constructive comments. We have responded (in blue) to each comment. We have modified and inserted some new figures and tables, as follows:

- Changes in Fig. 2D, E, F, G, I, J, K, Fig. 3A-D, F, H, Fig. 4A and Fig6C-D.
- Changes in Tables 1 and 2.
- Changes in Supplemental Fig. S3A-B, D (new panel), E, Fig. S4A (new panel), D-F (new panels), Fig. S5A, Fig. S6C-F (new panels), G-H and Fig. S7A-B (new panels)
- Changes in Supplemental tables S2, S4-S6 and new Supplemental table S7.

Reviewer #2 (Remarks to the Author):

The authors have not directly addressed my concerns regarding a lack of functional interrogation of potentially "pathogenic" EC subpopulations (difficulty with available antibodies for flow sorting). An attempt at showing functional significance of LRG1 in a HMEC1 human endothelial cell line is not very convincing (Supplemental Fig. 4).

We understand the reviewer's comment but addressing this specific point *in vivo* would require the use of transgenic knock-in mice, which is not realistic within a reasonable time frame. To confirm our data obtained on the human endothelial cell line HMEC1, we have now overexpressed LRG1 in primary human pulmonary microvascular EC (HMVEC-L) and again show that LRG1 expression preferentially promotes SMAD1/5 phosphorylation following TGF- β 1 stimulation (new Supplemental Fig. S4A). We also show that this switch is accompanied by a significant increase in spheroid sprouting (new supplemental Fig. S4D-E), confirming that Lrg1^{POS}PCEC signaling contributes to induction of angiogenesis.

Reviewer #3 (Remarks to the Author):

The authors have addressed satisfactory most of my comments. I still believe that the discussion is too long and in places repetitive.

We thank the reviewer for his/her positive comments. As suggested, we have revised the discussion section and removed the remaining redundancies.

Reviewer #5 (Remarks to the Author):

Major concerns of Reviewer#1 were related to:

1) Novelty of the study

Reviewer noted that other publications, including a recent study by Raslan et al. reporting similar findings, may impact the novelty of this research. In their rebuttal letter, Truchi et al. differentiated their study from Raslan et al. by highlighting the addition of time points and spatial transcriptomic analysis. Although the tools and experimental designs differ between the two studies, the findings remain for the most part incremental and descriptive, as reviewer#1 pointed out. However, considering the timing and growing interest in this pulmonary vascular field, I believe that generating independent findings from different groups is still valuable and contributes meaningfully to the field. Nevertheless, to strengthen their manuscript, Truchi et al. should provide a more detailed comparison of their findings with those reported by Raslan et al., highlighting both the similarities and differences between the two studies. For example, Raslan et al. reported a time course analysis in young animal using scRNA-seq and lineage tracing which provides deeper insights into capillary endothelial cells (ECs) changes resulting from bleomycin injury. In addition, Raslan et al. showed that Ntrk2+ gCap ECs (this population has the Lrg1+ EC transcriptional signature) peaks during fibrosis (day 14) and declines during the resolution phase (day 30) in injured young, while persist in injured aged mice. This indicates that this EC accumulation in aged mice may contribute to fibrosis progression, a conclusion that differs from the one proposed by Truchi et.al. Clarifying these distinctions would provide a more accurate

understanding of the impact of aging on lung capillary heterogeneity and fibrosis development. In addition, Truchi et al. reports spatial transcriptomic data, which could be leveraged to provide additional context and insights and better differentiate the contribution of their study with respect to previous ones.

We truly appreciate the time the reviewer took to review our manuscript and suggest improvements. We have done our best to address most reviewer's concerns and now believe that the manuscript is substantially improved.

We thank the reviewer for his/her positive comments regarding the scientific merit of our manuscript. We have taken his/her recommendation into account concerning the comparison with the study by Raslan *et al.* by considerably revising the paragraph already present in the discussion (**see lines 487-507**). Nevertheless, we would like to clarify one point concerning the novelty of our study, which should not be compromised by this article, as clearly stated by Nature Communications policy (<https://doi.org/10.1038/s41467-020-17817-x>), given that the paper by Raslan *et al.* has been accepted at Nat. Commun. during the first round of review of our manuscript.

We believe that the study authored by Raslan *et al.* complements our own, as we independently demonstrate the activation of the same signatures in response to bleomycin-induced injuries in PCECs. Furthermore, their dataset derived from lineage tracing of Aplnr cells in young mice offers particularly interesting insights in light of our findings. Firstly, these data confirm that the signatures observed at D14 in young mice are similar to those observed in aged mice at D30 in their initial dataset, but have resolved in young mice by this time. This confirms that aging delays the dynamics of these endothelial populations, as we demonstrate using a robust statistical framework.

Overall, our data strongly suggests that the transient emergence of Lrg1^{POS} PCEC subpopulations during the early phase of injury is involved in the regeneration of the alveolar niche (young mice). Conversely, a delayed time of appearance and late accumulation of these pro-angiogenic EC are likely to be associated with a defect in resolution. Additionally, the presence of EGFP in cells expressing both macro and microvascular markers suggests that bleomycin induces the recruitment of cells identified as post-capillary venules. However, it remains unclear whether this population corresponds to our sCap, as Raslan *et al.* do not detail the markers associated with this sub-population. To answer this question, we downloaded the corresponding processed data available on GEO but we were unable to find any EGFP transgene quantification to conclude on this point.

In addition to the characterization of the Ackr1^{POS} population (related to SV ECs according to the nomenclature of de Rooij *et al.*, Cardiovasc Res 2023) by Raslan *et al.*, we propose here a more precise characterization by highlighting in murine lung in response to bleomycin-induced injury, an additional capillary population named systemic capillary (sCap), expressing systemic markers, and homologous to the systemic population described in scRNA-seq data from lungs of IPF/COVID-19 patients (**see Fig. 2D-H and Supplemental Fig. S3A**). More broadly, by comparing fibrosis-induced modulations in the bleomycin model and in IPF for each endothelial population, we systematically establish the commonalities and divergences found between IPF patient lungs and the bleomycin model (**see Fig. 3H**).

Finally, our dataset appears best suited to provide reliable information on cellular and molecular differences induced by aging, a key mechanism of pulmonary fibrosis. Thanks to our longitudinal design, which includes three replicates for each of three time points after PBS/bleomycin injection for young or aged mice, we were able to develop statistical approaches for abundance and differential expression analyses that take into account variability between samples of the same condition, thus limiting the description of sample-specific events. On this basis, we have shown that aging not only impacts the activation dynamics of endothelial populations recruited in response to injury (**see Fig. 4A & 4D**), but also modulates the expression profile of gCap towards a pro-inflammatory phenotype in both fibrotic and physiological conditions (**see Fig. 5**). We have also shown that physiological gCap in aged mice displays certain characteristics of the injured endothelium.

The contribution of spatial transcriptomics data further enhances the originality of our study. However, the analysis of these data cannot elucidate the spatial organization of PCEC populations activated in

response to injuries, due to the low resolution of Visium technology, which hinders the clear observation of their signatures.

2) Concerns related to the capillary nature of Lrg1+ ECs.

The study's conclusion regarding the specific population of lung endothelial cells (ECs) expressing Lrg1 remains unclear. The scRNA-seq data suggest that Lrg1 is broadly expressed across ECs from all pulmonary vascular beds, with notably high expression levels in venous ECs. Thus, it remains unclear whether Lrg1+ capillary ECs are uniquely susceptible to aging-related effects compared to ECs from other vascular beds.

We acknowledge that this crucial point requires clarification. The misunderstanding may stem from the annotation of the "Lrg1^{POS} PCEC" populations that emerge in response to bleomycin-induced injuries (*i.e.* Lrg1^{POS} aCap, Lrg1^{POS} gCap, and sCap). This choice was initially motivated by the fact that Lrg1 is the gene most strongly up-regulated in PCECs from bleomycin-treated mice compared with their physiological counterparts, where the transcript is weakly expressed. Consequently, this modulation differs from that observed in venous cells, where Lrg1 was also found upregulated but from an already high basal level (see **Suppl. Fig. S3B & S3E**). We also selected Lrg1 for its functions described in the literature, which are in line with the pro-angiogenic signature detected in these subpopulations of PCECs.

With regard to the effect of aging on different endothelial populations, our two methods of comparison between young and old mice (*i.e.*, differential abundance comparison and differential expression comparison) are limited by the number of cells captured. Indeed, for differential abundance analysis, the proportion of each endothelial population per individual generally ranges between 0 and 3%, except for gCap, which limits the observation of stable dynamics. Despite this low number, we observed a delayed activation peak at day 28 in aged mice for Lrg1^{POS} gCap, Lrg1^{POS} aCap and SV EC, as well as a trend for sCap (see **Fig. 4A**). Thus, Lrg1^{POS} capillary EC are not the only endothelial population to exhibit activation dynamics modulated by aging. At the molecular level however, we were only able to observe this delay by comparing the activation signature score between young and aged mice at each time point in gCap (see **Fig. 4C**). Furthermore, for direct comparison of expression profiles between young and old mice under fibrotic or physiological conditions, only gCap had sufficient cells numbers to construct pseudobulks used for differential expression under physiological and fibrotic conditions (see **Fig. 5**).

Truchi et al. have revised their manuscript to include RNA-scope analysis showing the expression of Col15, a venous EC marker previously implicated in human lung fibrosis. Although the discovery of Col15a1-expressing ECs in mice following bleomycin injury is intriguing, its relevance to capillary Lrg1 remains unclear, as it is not clearly established whether Lrg1 is co-expressed in Col15 EC population. In addition, in human lungs, COL15a1+ ECs were primarily associated with bronchial venous-derived ECs, rather than capillary ECs, which highlights the need for further clarification on the relationship between COL15a1 and Lrg1 expression in mouse lung ECs.

The reviewer is right to point out that this is an important point to clarify. We believe that the comparative study from Peter Carmeliet's team (de Rooij *et al.*, Cardiovasc Res 2023) currently provides the best characterization of the human fibrotic pulmonary endothelium at the scRNA-seq level. This study demonstrates the increased abundance in fibrotic lungs of two populations both displaying systemic markers including COL15A1: one with venous cell characteristics (previously defined as bronchial venous EC) and a capillary population that had not been identified in the IPF cell atlas studies (Adams *et al.*, Habermann *et al.*). Based on this nomenclature, we updated the annotations in our bleomycin dataset and the IPF dataset used for comparison (Habermann *et al.*) to clearly distinguish between venous ECs (SV EC) and this novel systemic capillary population (sCap).

Our updated **Fig. 2D-H** clearly shows that our annotated sCap population, expressing systemic markers such as Col15a1, also expresses capillary markers like Aplinr, similar to what is found in IPF. Furthermore, if we directly quantify the proportion of cells co-expressing Lrg1 and Col15a1 among ECs, we find an equivalent percentage of double-positive cells in both sCap and SV ECs. However, adding

co-expression of *Aplnr* or *Ackr1* as a criterion, we observed that the majority of *Lrg1*^{pos}/*Col15a1*^{pos} sCap are *Aplnr*^{pos}/*Ackr1*^{neg}, whereas *Lrg1*^{pos}/*Col15a1*^{pos}/*Ackr1*^{pos} cells are almost exclusively SV EC (see figure below). These results demonstrate that *Col15a1* and *Lrg1* are co-expressed in the two systemic subpopulations displaying either a capillary (*Aplnr*^{pos} sCap) or a venous (*Ackr1*^{pos} SV EC) phenotype. Finally, we validated the co-expression of *Col15a1*, *Lrg1* and *Aplnr* by RNA FISH to clearly demonstrate the emergence of sCap in mouse lungs following bleomycin-induced injury (**new Supplemental Fig. 3D**).

Reviewer suggests using protein-level-based verification for several inflammatory, hypoxia-regulated, and pro-angiogenic genes, particularly regarding their functional roles in alveolar regeneration. The authors did not fully address this concern as RNAscope was utilized in the reported experiments (Fig. 2-4). It is unclear why this method was chosen over immunohistochemical analysis, especially since some proteins, such as HIF1A, are regulated primarily at the protein level.

The reviewer is right to point out that a validation at the protein-level would be ideal to limit the potential bias between RNA and protein expression levels. However, we were unfortunately still limited by the quality of antibodies (notably against mouse *Lrg1*), and we have not been able to set-up a sensitive approach for co-expression analysis combining RNAscope (for *Lrg1*) and immunofluorescence (for *Hif1a* and other inflammatory or proangiogenic factors), required to specifically localize the increased expression in *Lrg1*^{pos} PCECs. It is important to note, however, that RNAscope offers unique sensitivity and specificity thanks to multiple “double Z target probes” making this platform a gold standard for scRNA-seq marker validation.

3) Notable absence of *Lrg1*+ ECs in human lungs with pulmonary fibrosis.

*Another issue raised by the reviewer#1 is the lack of clarity and depth in addressing on *LRG1*+ capillary EC absence in human lungs with pulmonary fibrosis. The authors linked their findings to *COL15*-expressing ECs previously reported in the literature. This may create ambiguity and undermines the focus of the study. The authors should provide a rationale for the absence of *LRG1*+ capillary ECs in human lungs and distinguish their findings from those previously reported implicating *COL15* ECs in human lung fibrosis. One possible interpretation is that the discrepancy may be attributed to the end-stage nature of human diseased lungs, which differs significantly from the acute injury model used in mice. This emphasize that mouse models do not fully capture the disease pathogenesis, particularly in its advanced stage.*

We agree that the absence of *LRG1*^{pos} capillary ECs in human IPF lungs is an important point to discuss. Firstly, it is obvious that *LRG1* is not a relevant marker of IPF in scRNA-seq data. This can be initially explained by a significantly different basal expression pattern compared to that observed in murine data, suggesting variable physiological regulation between the two species. In addition, the difference between the acute injury bleomycin model and the advanced stage of fibrosis found in human samples is another plausible explanation. The transient dynamics of *Lrg1* induction in response to injuries,

which returns to a basal state upon resolution, supports this notion. These hypotheses are now included in the discussion section (see line 519-536).

More globally, it is also surprising to find that gCap and aCap exhibit relatively few transcriptomic modulations between IPF patients and donors (see Fig. 3H and Supplemental Table S4) compared to what is observed in the bleomycin-induced fibrosis, while the signature of sCap is well-conserved between the human pathology and the mouse model. Given that the sCap population relevant to IPF in mice expresses *Lrg1* and appears capable of differentiating into *Lrg1*^{POS} gCap cells, we believe it is important not to ignore this potential regenerative mechanism in mice from its potential implications in IPF, through a process that remains to be elucidated.

4) Lack of mechanism/s involving *Lrg1* capillary ECs

Reviewer#1 suggested to provide a possible mechanism explaining the emergence of the Lrg1+ ECs during lung injury. Truchi et. al. did not fully address this point. While this may be experimentally challenging and time consuming, the authors should use bioinformatic tools to get insights into the molecular events leading to Lrg1+ capillary EC activation, based on gene signatures emerging during the onset of lung fibrosis.

Thank you for pointing out this proposition to complete our observation and hypotheses on the molecular events involved in the emergence of *Lrg1*^{POS} ECs. Our first approach has been to investigate potential ligand-receptor interactions that could induce the *Lrg1*^{POS} capillary EC signatures using NicheNet (see Fig. 3F-G and Supplemental Fig. S5A). This analysis suggested the involvement of pro-angiogenic ligands and/or those associated with alveolar niche regeneration, such as *Vegfa*, *Cxcl12*, *Fgf2*, *Tgfb1*, or *Il1b*, supported by the significant upregulation of their receptors in *Lrg1*^{POS} gCap (Supplemental Fig. S5B).

First, we have first complemented this analysis by providing a list of predicted upstream regulators in Supplemental Fig. S4F based on the comparative Ingenuity Pathway enrichment analysis of the different pathological signatures of *Lrg1*^{POS} aCap, *Lrg1*^{POS} gCap, and sCap, to which we added those of SV EC and PV EC (see Fig. 3A-B for the previous analyses showing canonical pathways and functions associated). This new analysis confirmed the potential activation of hypoxia- and inflammatory-associated pathways and proposed additional candidates such as *FoxM1*, an important mediator of endothelial regeneration following chronic injury, whose expression is impaired in aged mice (Huang et al Science Translational Medicine 2023).

Next, we explored a second way to explain the mechanisms underlying the emergence of *Lrg1*^{POS} gCap, namely finding a potential differentiation trajectory originating from sCap based on RNA velocity (see Fig. 6). This trajectory scheme, obtained in two independent subsets (Young D14 and Old D28) sharing a large subset of genes (see new Supplemental Fig. 7B), is supported by the dynamics of genes suggesting other regulatory hypotheses, such as the transcription factor *Sox17*, the growth factor *Pdgfb*, and genes associated with apoptotic mechanisms like *Emp1*, *Cdk19*, and *Scarb1*. Again the main predicted upstream regulators candidates correspond to hypoxia-induced genes such as *Sox17*, *Cxcl8*, *Yap1* and *Gata6* with pro-survival, pro-angiogenic and/or pro-inflammatory activities (see new Supplemental Fig. 7C). The activation of the YAP/TAZ is in agreement with the data of Raslan *et al.*, pointing out YAP as a putative upstream regulator implicated in the transcriptional program leading to EC activation following bleomycin injury. It is particularly interesting to note that this model also predicted the switch of the TGF- β signaling from SMAD2/3 angiostatic signaling to SMAD1/5 angiogenic signaling associated with *Lrg1* function (Supplemental Fig. 4).

Overall, these various inferential analyses raise several hypotheses about the molecular events that could lead to the emergence of *Lrg1*^{POS} capillary EC, suggesting that these are potentially synergistic mechanisms involving activation of a hypoxic response and interaction of PCEC with several cell types within the alveolar niche. Activation hypoxic-mediated signaling pathways appears central and its duration and intensity may likely explain the efficacy of the vascular repair (see discussion line 459-463).

Major concerns of Reviewer#4 were related to:

1) Absence of Lrg1+ ECs in human lungs.

The absence of Lrg1 ECs in human lungs remains a weakness of the study. Thus, the relevance of Lrg1 ECs in the context of human pulmonary disease remains inconclusive. The discrepancy between mouse and human data may be attributed to the end-stage nature of human diseased lungs, which differs from available mouse models. This aspect does not seem to be adequately discussed in the manuscript. We agree that the absence of LRG1^{POS} capillary ECs in human IPF lungs is an important point to discuss. This can be initially explained by a significantly different basal expression pattern compared to that observed in murine data, suggesting variable physiological regulation between the two species. In addition, the difference between the acute injury bleomycin model and the advanced stage of fibrosis found in human samples is another plausible explanation. The transient dynamics of Lrg1 induction in response to injury, which returns to a basal state after resolution, supports this notion. These hypotheses have been now included in the discussion section (see line 517-536).

More globally, it is also surprising to find that gCap and aCap exhibit relatively few transcriptomic modulations between IPF patients and donors (see Fig. 3H and Supplemental Table S4) compared to what is observed in the bleomycin-induced fibrosis, while the signature of sCap is well-conserved between the human pathology and the mouse model. Given that the sCap population relevant to IPF in mice expresses Lrg1 and appears capable of differentiating into Lrg1^{POS} gCap cells, we believe it is important not to ignore this potential regenerative mechanism in mice from its potential implications in IPF, through a process that remains to be elucidated.

2) Lack of statistical power and absence of healthy lungs in the spatial transcriptomic study.
The authors have sufficiently addressed this concern.

3) In Figure1 description, damaged alveolar regions are characterized by low UMI content yet the heatmap in Supp.Fig1D shows that it is not specific to age related differences.
The authors have sufficiently addressed this concern.

4) Since single cell RNA Seq results indicates the significant downregulation of certain genes (Supp.Fig3B-C) it would be a better approach to verify these findings in spatial transcriptomics.
The authors were unable to address this point due to the low resolution of the spatial transcriptomic analysis.

5) The expression of Smad family members' difference in between Pbs vs Bleo model are not significant for Smad2 and Smad5 (Supp.Fig3C). In the discussion, it is claimed that Smad2 and Smad5 works controversially with the influence of Lrg1 and Tgfb1 signalling. Is there any functional experiment related with the activity of Smad Tgfb and Lrg1? Or is there any transcriptomic data shows that Age+Bleo related differences among these genes that supports the claim in line 400?
The authors have sufficiently addressed this weakness and provided additional data.

We thank the reviewer for his positive feedback on these different points. As mentioned above, we have now confirmed these data in primary human pulmonary microvascular EC and also showed that this TGF- β switch is accompanied by a significant increase in spheroid sprouting (new supplemental Fig. S4D-E). It is also interesting to note that the gene signature supporting the inferred trajectory in the RNA velocity approach also predicted this switch (Supplemental Fig. 7C).